# Knowledgeable Language Models as Black-Box Optimizers for Personalized Medicine

**Michael S Yao**[*]
University of Pennsylvania
myao2199@upenn.edu

**Osbert Bastani**
University of Pennsylvania

**Alma Andersson**
Genentech

**Tommaso Biancalani**
Genentech

**Aïcha Bentaieb**
Genentech

**Claudia Iriondo**
Genentech
iriondoc@gene.com

## Abstract

The goal of personalized medicine is to discover a treatment regimen that optimizes a patient's clinical outcome based on their personal genetic and environmental factors. However, candidate treatments cannot be arbitrarily administered to the patient to assess their efficacy; we often instead have access to an *in silico* surrogate model that approximates the true fitness of a proposed treatment. Unfortunately, such surrogate models have been shown to fail to generalize to previously unseen patient-treatment combinations. We hypothesize that domain-specific prior knowledge—such as medical textbooks and biomedical knowledge graphs—can provide a meaningful alternative signal of the fitness of proposed treatments. To this end, we introduce **LLM**-based **E**ntropy-guided **O**ptimization with k**N**owledgeable priors (**LEON**), a mathematically principled approach to leverage large language models (LLMs) as black-box optimizers without any task-specific fine-tuning, taking advantage of their ability to contextualize unstructured domain knowledge to propose personalized treatment plans in natural language. In practice, we implement LEON via 'optimization by prompting,' which uses LLMs as stochastic engines for proposing treatment designs. Experiments on real-world optimization tasks show LEON outperforms both traditional and LLM-based methods in proposing individualized treatments for patients.

## 1 Introduction

**Personalized medicine** is a clinical strategy that seeks to individualize treatment strategies based on a patient's unique genetic and environmental features (Gutowski et al., 2023; Mizan & Taghipour, 2022; Adams et al., 2022; Consortium, 2009). Through reasoning about a patient, previously observed patients, and existing medical knowledge and literature, clinicians seek to determine an optimal treatment (Mehandru et al., 2025). Such a task can be framed as a *conditional optimization problem*, where the goal is to design a treatment regimen—conditioned on the patient's unique features—that optimizes their clinical outcome. However, applying traditional optimization methods in this setting presents significant challenges. First, **ground-truth design evaluations are costly**; it is infeasible to assess the efficacy of novel treatment options directly in human subjects. Evaluating newly proposed treatment regimens may therefore be difficult or even impossible. To overcome this limitation, a common approach is to instead leverage feedback from a *surrogate* for the ground-truth objective, such as a machine learning model or digital twin, to estimate the quality of a proposed treatment for the patient (Kuang et al., 2024; Katsoulakis et al., 2024). However, **such surrogate models are frequently imperfect and used on out-of-distribution patients** (De Domenico et al., 2025). Certain populations are systematically under-enrolled in clinical studies (Costa et al., 2023; Walker et al., 2022; Islam et al., 2024; Casey et al., 2022; Pittell et al., 2023), and so black-box surrogate functions often fail to accurately predict design fitness for different patient populations (Taylor et al., 2022; Kirtane & Lee, 2017; Smith-Graziani & Flowers, 2021; Larkin et al., 2022).

---

[*]Work done during an internship at Genentech.

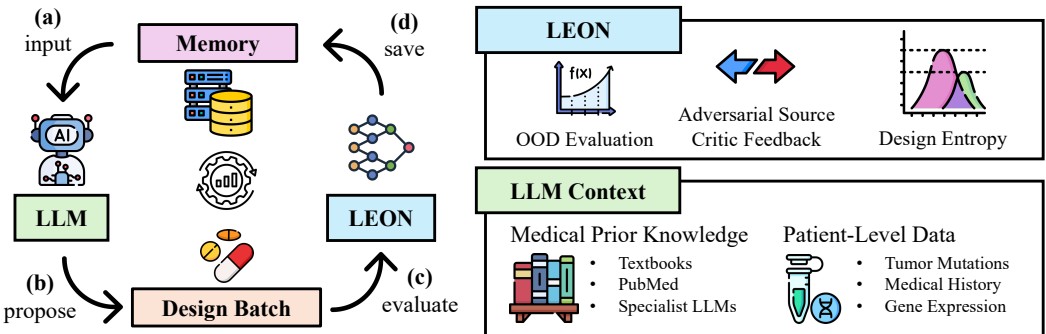

Figure 1: **LLM-based Entropy-guided Optimization with kNowledgeable priors (LEON).** We use LLMs as zero-shot conditional optimizers to propose personalized treatment designs at the patient level. **(a)** The LLM is prompted with prior knowledge and the history of previously proposed designs and their predicted scores to **(b)** propose a new batch of designs. **(c)** These candidates are evaluated using LEON, and then **(d)** cached as context to the LLM in subsequent iterations.

A natural question is whether designing better surrogate models can overcome these limitations, with the hope that more accurate proxies of the ground-truth objective will yield better optimized therapeutic proposals. For example, Kuang et al. (2024) enforce a physics-based prior on a learned digital twin model, and Yu et al. (2021) assume the objective is locally smooth over the design space. However, in many real-world applications we cannot specify the surrogate model—the underlying mechanisms behind patient responses to treatment may be under-specified or even entirely unknown (Kuang et al., 2024), and building accurate digital twin models is often limited by the availability of real patient data and patient privacy concerns (De Domenico et al., 2025). In these settings, we may only think of both the surrogate and underlying ground-truth functions as **black-box models**.

Recent work (Reinhart & Statt, 2024; Yang et al., 2024a; Chen et al., 2025a; Ma et al., 2024a; Nasir et al., 2024; Song et al., 2024) on large language models (LLMs) have explored their emerging capabilities in *black-box optimization*, where the goal is to propose a design $x$ that maximizes a black-box objective function $f(x)$. In particular, modern LLMs have been shown to solve zero-shot optimization problems in domains such as mathematics (Nadkarni et al., 2021; Novikov et al., 2025) and computer science (Garg et al., 2022; Guo et al., 2024c; Fu et al., 2024; Ma et al., 2024b). However, in these settings the objective function is almost always evaluable with minimal cost (i.e., using a code interpreter or formal verification tools)—unlike in clinical patient-centered tasks.

In this work, we introduce a method to leverage LLMs as black-box optimizers in clinical medicine. Our core hypothesis is that by leveraging domain-specific prior knowledge, LLM-based optimizers can overcome the limitations associated with out-of-distribution surrogate model predictions during optimization. More explicitly, our contributions are as follows:

1. **Formulating personalized medicine as a black-box optimization problem.** We propose a mathematically principled approach to formulate personalized medicine as a conditional black-box optimization problem, where the goal is to find an optimal treatment strategy conditioned on a set of input patient covariates that optimizes a target outcome metric.

2. **Constraining the optimization problem.** Existing surrogate models are often imperfect proxies of costly ground-truth objectives in personalized medicine. To overcome this, we introduce a set of intuitive constraints to our initial black-box optimization problem. Our constraints limit the optimization trajectory to treatment designs that (1) are likely to have reliable predictions from the surrogate model; and (2) are consistently proposed by the LLM as high-quality treatments according to relevant domain knowledge.

3. **Deriving a solution to the constrained optimization problem.** We derive a computationally tractable solution to our proposed re-formulation of personalized medicine as a constrained black-box optimization problem. Our approach fundamentally relies on the statistical analysis of the distribution of proposed designs and the use of an adversarial source critic model during optimization. We refer to our method as **LLM-based Entropy-guided Optimization with kNowledgeable priors** (**LEON**) (**Fig. 1**). We implement LEON via

'optimization-by-prompting' (Yang et al., 2024a), using the LEON-defined objective to score candidate designs. In this setting, the LLM functions as a stochastic treatment recommendation system that seeks to iteratively propose higher scoring designs.

4. **Using LEON to solve real-world personalized medicine tasks.** We demonstrate how LEON can be used to solve conditional optimization problems over both discrete and continuous search spaces. Comparing with 10 other baseline methods, we find that LEON achieves an average rank of **1.2** on 5 representative treatment design problems.

## 2 Related Work

**LLMs as optimizers.** A growing body of work has explored the ability of generalist LLMs to solve domain-specific optimization tasks (Reinhart & Statt, 2024; Liu et al., 2024b; Yang et al., 2024a; Ma et al., 2024a; Song et al., 2024). For instance, Chen et al. (2025a) fine-tune language models for antibody protein design. However, unlike in medicine, accurate molecular dynamic simulations (Albaugh & Gingrich, 2022; Ho et al., 2024), state-of-the-art foundation models (Novikov et al., 2025; Wang et al., 2024b), and scalable experimental setups (Yang et al., 2025; Rix et al., 2020; Johnston et al., 2024) have made online optimization methods empirically effective in this domain. Similarly, methods like those introduced by Shojaee et al. (2025); Shypula et al. (2024); Ma et al. (2024b) introduce methods to use LLMs for code optimization according to an objective that is easily verifiable by modern computer systems. Novikov et al. (2025) and Romera-Paredes et al. (2024) use LLMs to propose new solutions to resource allocation and extremal combinatoric problems. Chen et al. (2023a); Nasir et al. (2024); Ji et al. (2025); Chiquier et al. (2024) separately use LLMs to design new machine learning models for neural architecture search, Zeng et al. (2024); Hong et al. (2025) for tensor network design, and Akioyamen et al. (2024) for database query optimization without any model fine-tuning. However, unlike the problem of personalized treatment design, these applications are all examples of *unconditional* optimization tasks.

**Optimization under distribution shift.** A separate body of work has considered the problem of leveraging traditional (i.e., non-LLM-based) optimizers under distribution shift. Trabucco et al. (2021) leverages gradient data to update the surrogate model over the course of an optimization experiment to act as a conservative lower bound of the ground-truth function, Yu et al. (2021) imposes a smoothness prior on the surrogate model over the design space, and Chen et al. (2023b) leverages a retrieval-based approach to build a more robust surrogate model. However, such methods assume control over the design of the surrogate model—an assumption that fails in black-box optimization. Other techniques forgo learning a surrogate model altogether Mashkaria et al. (2023); Krishnamoorthy et al. (2023); however, such methods rely on learning from multiple design observations in a single context, which do not exist in *conditional* optimization tasks like those in personalized medicine. Finally, Angermueller et al. (2020) and Williams (1992) formulate optimization as a reinforcement learning problem; however, we do not consider sequential decision-making tasks in our work.

## 3 Background and Preliminaries

**Entropy and equivalence classes.** Entropy is a fundamental concept in information theory that quantifies the relative uncertainty associated with the variables of a system. Prior work from Qiu et al. (2025); Farquhar et al. (2024); Kuhn et al. (2023) has examined how the entropy of the distribution of an LLM's outputs can be used to estimate the model's epistemic uncertainty. Briefly, if a non-deterministic model consistently returns equivalent responses to the same prompt—corresponding to a low-entropy distribution of outputs—we can be more confident in its own certainty of its response. Such methods have been shown to improve question answering (Nikitin et al., 2024; Kuhn et al., 2023), hallucination detection (Farquhar et al., 2024), and document retrieval (Qiu et al., 2025).

Crucially, the definition of what constitutes a set of 'equivalent' responses is predicated on the existence of an **equivalence relation** $\sim$: two outputs $x, x'$ are equivalent iff $x \sim x'$. Any valid equivalence relation partitions the input space into a set of $N$ disjoint *equivalence classes* $[x]_i$, where $[x]_i = \{x' \in \mathcal{X} : x' \sim x\}$ and $\bigcup_{i=1}^{N}[x]_i = \mathcal{X}$.[1] We denote this *quotient set* of these $N$

---

[1]In this work, we assume that the number of equivalence classes $N$ is finite. This is a relatively weak assumption for most real-world, computable optimization problems (Kozen, 1997).

equivalence classes by $\mathcal{X}/\sim$. The entropy with respect to $\sim$ is given by $\mathcal{H}_{\sim} := -\sum_{i=1}^{N} p_i \log p_i$, where $p_i := |[x]_i|/|\mathcal{X}|$ is the fractional occupancy of equivalence class $[x]_i$.

**Adversarial supervision in optimization.** A central challenge in optimization under distribution shift lies in the absence of direct access to the ground-truth objective function during optimization. In practice, optimization can instead be performed against a surrogate model, which may be inaccurate in out-of-distribution (OOD) regions inevitably explored during the optimization process. Naïvely optimizing against a surrogate model can therefore produce candidate solutions that appear promising according to the surrogate, but perform poorly when ultimately evaluated using the ground-truth objective. To mitigate this, recent works bound the 1-Wasserstein distance between the distribution of real designs used to train the surrogate model and that of generated candidates (Yao et al., 2025b; 2024). Such a constraint provides theoretical guarantees on the generalization error of the surrogate model, effectively reducing the extent of extrapolation (**Supplementary Theorem D.1**).

However, directly computing the 1-Wasserstein distance between probability distributions poses significant computational challenges, as classical algorithms can scale as $\mathcal{O}(n^3)$ in the number of samples (Kuhn, 1955). To address this, one can exploit Kantorovich & Rubinstein (1958) to recast the 1-Wasserstein distance as a supremum over a class of Lipschitz functions:

$$W_1(p,q) := \mathbb{E}_{x \sim p(x)}[c^*(x)] - \mathbb{E}_{x \sim q(x)}[c^*(x)] \tag{1}$$

where $p(x)$ (resp., $q(x)$) is the (empirical) distribution of the real (resp., generated) designs and $c^*(x) := \arg\max_{||c||_L \leq 1} \left[ \mathbb{E}_{x \sim p(x)}[c^*(x)] - \mathbb{E}_{x \sim q(x)}[c^*(x)] \right]$. One can think of the function $c^*$ as an adversarial *source critic* that learns to discriminate the source distribution of an input $x$. The learned function $c^*(x)$ thereby assigns high (resp., low) value to inputs that are likely to have been sampled from $p(x)$ (resp., $q(x)$). Such an approach has been shown to reduce the extrapolation of learned models in generative adversarial learning (Arjovsky et al., 2017; Yao et al., 2025b; 2024).

# 4 ENTROPY-GUIDED OPTIMIZATION WITH KNOWLEDGEABLE PRIORS

## 4.1 PROBLEM FORMULATION

The task of finding an optimal patient treatment strategy can be formulated as an conditional black-box optimization problem, where the goal is to find an optimal distribution $q(x)$ according to

$$\arg\max_{q(x) \in p(\mathcal{X})} \quad \mathbb{E}_{x \sim q(x)}[f(x;z)] \tag{2}$$

where $\mathcal{X}$ is the set of all possible treatments, $p(\mathcal{X})$ the set of all valid probability measures over $\mathcal{X}$, $x$ any particular treatment *design*, and $z \in \mathcal{Z}$ the patient *conditioning vector*. For example, $\mathcal{X}$ might be a set of medications, $z$ a patient's personal health data sampled from a distribution $p(z)$, and $f$ the patient's therapeutic response to a medication. In this setting, the objective is not to find a universally optimal design, but rather to identify an optimal treatment $x$ given a specific context $z$.

In many real-world applications, the ground-truth objective function $f$ is inaccessible during optimization. We instead only have access to a *surrogate* function $\hat{f} : \mathcal{X} \times \mathcal{Z} \to \mathbb{R}$ trained on a distribution of observations whose $z$-marginal is not equal to $p(z)$. The surrogate $\hat{f}$ may be a patient simulator (Man et al., 2014; Evans et al., 2015), a digital twin (Katsoulakis et al., 2024; De Domenico et al., 2025; Kuang et al., 2024), or a machine learning model trained to approximate $f$. Importantly, **we highlight the mismatch between** $p(z)$ **and the source distribution of the training set of** $\hat{f}$; for example, $\hat{f}$ may only be learned from patients at one hospital, and our patient $z \sim p(z)$ is sampled from a different hospital. A natural strategy in this setting is to instead solve the related problem

$$\arg\max_{q(x) \in p(\mathcal{X})} \quad \mathbb{E}_{x \sim q(x)}[\hat{f}(x;z)] \tag{3}$$

with the hope that optimizing against $\hat{f}$ admits a distribution over $\mathcal{X}$ that also maximizes $f$ in expectation. Importantly, the *conditional* problem formulation in (3) diverges from the related unconditional problem commonly considered in prior work (Trabucco et al., 2022; 2021; Yu et al., 2021). In practice, solving (3) is ineffective (Trabucco et al., 2021; Yu et al., 2021). This is because the out-of-distribution surrogate function $\hat{f}$ frequently exhibits detrimental biases and performance degradations on a target distribution of patients, leading to worse outcomes according to the ground-truth objective function $f$ in clinical settings (Taylor et al., 2022; Kirtane & Lee, 2017; Larkin et al.,

2022). To overcome this limitation, we follow Yao et al. (2025b; 2024) and assume access to a dataset $\mathcal{D}_{\text{src}} \subseteq \mathcal{X}$ of previous treatment designs (e.g., the full treatment-patient dataset used to learn $\hat{f}$ projected onto $\mathcal{X}$); we show in our work how to use $\mathcal{D}_{\text{src}}$ to solve a modified instance of (3).

## 4.2 CONSTRAINED CONDITIONAL OPTIMIZATION

We first modify the original problem in (3) by introducing two constraints:

$$\arg\max_{q(x)\in p(\mathcal{X})} \quad \mathbb{E}_{x\sim q(x)}[\hat{f}(x;z)]$$
$$\text{s.t.} \quad \mathbb{E}_{x'\sim\mathcal{D}_{\text{src}}}[c^*(x')] - \mathbb{E}_{x\sim q(x)}[c^*(x)] \leq W_0, \quad \text{and} \quad \mathcal{H}_{\sim}(q(x)) \leq H_0 \tag{4}$$

Following (1), the first constraint imposes an upper bound $W_0$ on the 1-Wasserstein distance between the empirical distribution of proposed treatment designs $q(x)$ and a dataset $\mathcal{D}_{\text{src}}$ of previously proposed designs in the real world. Leveraging an auxiliary source critic model $c^* : \mathcal{X} \to \mathbb{R}$, this constraint ensures that the distribution of proposed designs is not too dissimilar to the distribution of historically reported designs, implicitly constraining the allowed degree of extrapolation against $\hat{f}$ during optimization (**Supplementary Theorem D.1**). This mitigates the risk of proposing spurious candidates that appear favorable under the surrogate objective $\hat{f}$ but are unlikely to perform well in practice (Yao et al., 2025b; 2024). Importantly, the domain of $c^*$ is restricted to $\mathcal{X}$; this means that no patient observations $z$ from the source dataset $\mathcal{D}_{\text{src}}$ are required. In other words, **patient privacy of individuals in $\mathcal{D}_{\text{src}}$ is explicitly preserved** in (2). The second constraint places an upper bound $H_0$ on the $\sim$-*coarse-grained entropy* of the distribution of designs $q(x)$, defined below:

**Definition 4.1** ($\sim$-Coarse-Grained Entropy). Let $\sim$ be an equivalence relation over the input space $\mathcal{X}$, and assume that the set of equivalence classes $\mathcal{X}/\sim$ imposed by $\sim$ is finite. Let $N := |\mathcal{X}/\sim|$ be the number of equivalence classes and $q(x)$ be a valid probability distribution over the input space $\mathcal{X}$. Denote $[x]_i$ as the $i$th equivalence class in $\mathcal{X}/\sim$, and $\bar{q}_i := \int_{[x]_i} dx\, q(x)$ to be the probability of drawing an element from the $i$th equivalence class. Then, the $\sim$-**coarse-grained entropy** $\mathcal{H}_{\sim} : p(\mathcal{X}) \to \mathbb{R}_+$ is defined as $\mathcal{H}_{\sim}(q(x)) := -\sum_{i=1}^{N} \bar{q}_i \log \bar{q}_i$.

The second constraint therefore enforces an upper bound on the entropy of the distribution of designs with respect to the equivalence relation $\sim$, encouraging sampling strategies (such as those that leverage domain-specific prior knowledge) that increase the certainty of the optimizer's proposals. In general, solving (4) exactly is highly intractable; both $\hat{f}$ and $c^*$ can be arbitrarily non-convex black-box functions, and the $\sim$-coarse-grained entropy may be highly sensitive to perturbations in the input space. To address this, we first show using **Lemma 4.2** how to derive an ansatz to the constrained problem in (4). We then show how to algorithmically solve for the free parameters of the solution class as applied to a suite of real-world optimization tasks for personalized medicine.

**Lemma 4.2** (Design Collapse Within Equivalence Classes). *Using the method of Lagrange multipliers, we can rewrite (4) as a function of the* partial Lagrangian $\mathcal{L}_{\lambda}(q)$ *for some constant* $\lambda \in \mathbb{R}_+$:

$$\arg\max_{q(x)\in p(\mathcal{X})} \quad \mathcal{L}_{\lambda}(q) := \mathbb{E}_{x\sim q(x)}[\hat{f}(x;z)] + \lambda(W_0 - \mathbb{E}_{x\sim\mathcal{D}_{\text{src}}}[c^*(x)] + \mathbb{E}_{x\sim q(x)}[c^*(x)])$$
$$\text{s.t.} \quad \mathcal{H}_{\sim}(q(x)) \leq H_0 \tag{5}$$

*Suppose there exists a distribution $q(x)$ that satisfies the remaining constraint in (5). Furthermore, assume that the function $\hat{f}(x;z) + \lambda c^*(x)$ is continuous everywhere and coercive in $\mathcal{X}$. For all $N$ equivalence classes, we can then define $x_i^*$ (not necessarily unique) according to*

$$x_i^*(\lambda) := \arg\max_{x\in[x]_i} \left( \hat{f}(x;z) + \lambda c^*(x) \right) \tag{6}$$

*Then, the alternative distribution $q^*(x) = \sum_{i=1}^{N} \bar{q}_i\, \delta(x - x_i^*)$, where $\bar{q}_i$ is as in **Definition 4.1**, also satisfies the constraint and simultaneously achieves a non-inferior value $\mathcal{L}_{\lambda}(q^*) \geq \mathcal{L}_{\lambda}(q)$.*

The proof for this result is shown in **Appendix A**. Intuitively, **Lemma 4.2** shows that any feasible distribution $q(x)$ cannot be superior to an alternative distribution $q^*(x)$ that is both feasible and places all of its probability mass on each of the optimal designs within each equivalence class. We remark that each $x_i^*$ need not be unique within the corresponding equivalence class $[x]_i$: it is easy to show that any solution to (6) within the same $\sim$-equivalence class (or an appropriately weighted

combination of multiple optimal solutions) still admits a feasible distribution $q^*(x)$. **Lemma 4.2** allows us to restrict the search space of optimal distributions within $p(\mathcal{X})$. In particular, note that for a given $\lambda$ and equivalence class $\sim$, the optimal policy is exactly specified by the choice of equivalence class probabilities $\bar{q}_i$. The original problem in (4) is therefore equivalent to

$$
\arg\max_{\bar{q} \in \Delta(N)} \quad \sum_{i=1}^{N} \bar{q}_i \hat{f}(x_i^*)
$$

$$
\text{s.t.} \quad \mathbb{E}_{x \sim \mathcal{D}_{\text{src}}}[c^*(x)] - \sum_{i=1}^{N} \bar{q}_i[c^*(x_i^*)] \leq W_0, \quad \text{and} \quad \mathcal{H}(\bar{q}) \leq H_0 \tag{7}
$$

where $\Delta(N)$ is the $N$-dimensional probability simplex and $\mathcal{H}(\cdot)$ is the standard Shannon entropy of the $N$-dimensional vector $\bar{q}$. This alternative problem formulation leads us to our main result:

**Lemma 4.3** (Probabilistic Sampling Over Equivalence Classes). *Consider the constrained optimization problem as in (7). The $i$th element of the $N$-dimensional vector $\bar{q}$ can be written as*

$$
\bar{q}_i = \exp\left[\mu\left(\hat{f}(x_i^*; z) + \lambda c^*(x_i^*)\right)\right] \Big/ \mathcal{Z}(\lambda) \text{ where } x_i^* := \arg\max_{x \in [x]_i}\left(\hat{f}(x; z) + \lambda c^*(x)\right) \tag{8}
$$

*where $\mathcal{Z}(\lambda)$ is a normalizing constant and $\lambda, \mu^{-1} \in \mathbb{R}_+$ are the Lagrange multipliers.*

The proof for **Lemma 4.3** is included in **Appendix A**. Intuitively, we can think of $\lambda$ and $\mu$ as 'certainty' parameters: increasing $\lambda$ upweights the importance of sampling a design $x_i^*$ associated with a high certainty of 'in-distribution-ness' according to the source critic function $c^*(x)$. Similarly, increasing $\mu$ upweights the importance of the probability vector $\bar{q}$ producing a 'collapsed' distribution of designs with low entropy (and therefore high certainty). Prior work has either explicitly fixed variables similar to $\lambda, \mu$ as hyperparameters (Yu et al., 2021; Trabucco et al., 2021) or imposed restrictive assumptions on the input space to solve for the Lagrange multipliers (Yao et al., 2024). In contrast, we introduce a principled and computationally tractable approach to dynamically solve for the optimal $\lambda$ and $\mu$ over the course of the optimization trajectory.

## 4.3 Empirically Fixing the LLM Certainty $\mu$

The $\sim$-coarse-grained entropy from **Definition 4.1** is an intrinsic property of a language model optimizer: given a single prompt at a particular optimization step, the LLM can return multiple possible designs because the autoregressive model is non-deterministic for positive temperature values. The values of $\bar{q}_i$ can therefore be empirically observed by sampling a batch of designs from the LLM optimizer, allowing us to estimate the $\mu$ degree of freedom in (8). More concretely, with each batched sampling of proposals from the LLM, we assign the treatment designs into their respective equivalence classes to arrive at an unbiased estimate of the fractional occupancies of each class $\hat{q}_i$, and compute the optimal values $\hat{f}(\hat{x}_i^*; z) + \lambda c^*(\hat{x}_i^*)$ according to (8). **Lemma 4.3** then gives

$$
\log \hat{p}_i \approx -\log \mathcal{Z}(\lambda) + \mu\left(\hat{f}(\hat{x}_i^*; z) + \lambda c^*(\hat{x}_i^*)\right)
$$

We can then estimate the value of $\mu$ by simple linear regression, treating each observation $(\hat{f}(\hat{x}_i^*; z) + \lambda c^*(\hat{x}_i^*), \log \hat{p}_i)$ over $N$ equivalence classes as an explanatory-dependent variable pair:

$$
\hat{\mu} = \frac{\sum_{i=1}^{N}\left[(\hat{f}(\hat{x}_i^*; z) - \bar{f}) + (\lambda c^*(\hat{x}_i^*) - \bar{c}^*)\right]\left[\log p_i - \frac{1}{N}\sum_{i'=1}^{N}\log p_{i'}\right]}{\sum_{i=1}^{N}\left[(\hat{f}(\hat{x}_i^*; z) - \bar{f}) + (\lambda c^*(\hat{x}_i^*) - \bar{c}^*)\right]^2} \tag{9}
$$

where the expectation values $\bar{f}, \bar{c}$ are defined over the $N$ equivalence classes. Intuitively, predictions with high entropy (i.e., low certainty) will be scaled to a lower reward, as $\log p_i$ will be constant over equivalence classes and so $\hat{\mu} \approx 0$. Conversely, a confident model with high certainty will lead to a greater estimate of $\hat{\mu} > 0$, increasing the reward associated with the proposed designs. In this framework, **the role of prior knowledge is to help 'overcome' the statistical randomness of the LLM's next-token generative process in proposing treatment designs** to improve LLM certainty.

### 4.4 SOLVING FOR THE SOURCE CRITIC CERTAINTY $\lambda$

We first solve for the Lagrangian dual function $g(\lambda, \mu) := \max_{\bar{q} \in \Delta(N)} \mathcal{L}(\bar{q}; \lambda, \mu)$.

**Corollary 4.4** (Dual Function of (7))**.** *The dual function of the constrained problem in (7) is*

$$g(\lambda, \mu) = \lambda(W_0 - \mathbb{E}_{x \sim \mathcal{D}_{src}}[c^*(x)]) + \mu^{-1} H_0 + \mu^{-1} \log \mathcal{Z}(\lambda)$$

*where $\mathcal{Z}(\lambda)$ is the normalizing constant from (8), and so*

$$\frac{\partial g(\lambda, \mu)}{\partial \lambda} = W_0 - \left( \mathbb{E}_{x \sim \mathcal{D}_{src}}[c^*(x)] - \sum_i \bar{q}_i c^*(x_i^*) \right) \tag{10}$$

The proof of this result is included in **Appendix A**. Importantly, (10) allows us to iteratively solve for the optimal value of the dual parameter $\lambda$ via gradient descent *without any explicit gradient information from the black-box functions $\hat{f}$ and $c^*$*:

$$\lambda_{t+1} = \lambda_t - \eta_\lambda \frac{\partial g(\lambda, \hat{\mu})}{\partial \lambda} = \lambda_t - \eta_\lambda \left[ W_0 - \mathbb{E}_{x \sim \mathcal{D}_{src}}[c^*(x)] + \sum_i \bar{q}_i c^*(x_i^*) \right] \tag{11}$$

Here, $\eta_\lambda > 0$ is a learning rate hyperparameter and $\bar{q}_i$ is as in (8). Intuitively, if the designs $x_i^*$ are in-distribution compared to $\mathcal{D}_{src}$, then the Wasserstein distance $\mathbb{E}_{x \sim \mathcal{D}_{src}}[c^*(x)] - \sum_{i=1}^{N} \bar{q}_i c^*(x_i^*) \leq W_0$, meaning $\partial g(\lambda, \mu)/\partial \lambda > 0$ and the value of $\lambda_{t+1}$ will decrease to allow for greater exploration of the design space. Conversely, sampling out-of-distribution designs $x_i^*$ will yield $\partial g(\lambda, \mu)/\partial \lambda < 0$, and $\lambda_{t+1}$ will increase to reduce the extrapolation against $\hat{f}$.

### 4.5 OVERALL ALGORITHM

Our overall method to solve (4) consists of four primary steps. (1) **Sampling.** We first query the LLM optimizer to propose a batch of **independently sampled** new treatment designs. The LLM is prompted with a description of the task and the patient $z$, any prior knowledge it produced prior to optimization, and a table of its previously proposed designs and their corresponding scores according to our algorithm. (2) **Clustering.** We then take as input the batch of designs and individually assign them to their corresponding equivalence classes. (3) **Certainty Estimation.** We estimate the fractional occupancy $\hat{q}_i$ and the optimal value of $\hat{f}(\hat{x}_i^*; z) + \lambda c^*(\hat{x}_i^*)$ observed in each equivalence class, and then estimate the certainty parameter $\mu$ according to (9). We also update the value of $\lambda$ following (11). (4) **Design Scoring.** Using our estimates of $\mu$ and $\lambda$, we score each sampled design $x$ according to $\mu[\hat{f}(x; z) + \lambda c^*(x)]$, and store the treatments and their scores to provide as context to subsequent LLM optimization prompting. These steps are repeated until the maximum query budget for $\hat{f}(x; z)$ is reached. We refer to our method as **LLM-based Entropy-guided Optimization with kNowledgeable priors** (**LEON**); the full pseudocode is in **Supplementary Algorithm 1**. Note that our method only updates the LLM prompt; **there is *no* fine-tuning of the weights of the LLM**.

**The role of prior knowledge in LEON.** Notably, LEON may be used with any generalist language model without any task-specific fine-tuning. Such consumer-grade LLMs may not be able to propose more optimal designs given only access to prior observations alone, adversely affecting the model certainty according to $\mu$. To overcome this limitation, we first provide the LLM access to a set of external knowledge repositories with domain-specific knowledge—such as medical textbooks, biomedical knowledge graphs, and publicly available clinical databases. Given a description of the optimization task and the patient features $z$, the LLM is allowed to choose which knowledge sources may be helpful, and then sequentially query the relevant textual corpora as *tools* to synthesize a prior knowledge statement in natural language. We then provide this prior knowledge as prompt context to the LLM in all subsequent optimization steps. In our work, we show how leveraging prior knowledge in this way can help the LLM propose higher-quality designs, thereby increasing the value of $\mu$ and improving the quality of individualized treatment regimens.

**Reflection on optimization steps.** After a batch of designs are scored and before a new batch of designs are acquired, we prompt the backbone language model to analyze the most recently sampled batch of designs and their corresponding scores following prior work (Ma et al., 2024b; Yao et al., 2023). The LLM is asked to reflect on the data and underlying sampling strategy in natural language; the output of this reflection is included into the LLM prompt in the next batch acquisition step.

## 5 EMPIRICAL EVALUATION

**Personalized medicine optimization tasks under distribution shift.** Recall that our core motivation for LEON is to overcome the effect of distribution shifts when personalizing treatment plans for previously unseen, potentially out-of-distribution patients. To this end, we constructed a set of 5 real-world optimization tasks to evaluate LEON and baseline methods. (1) **Warfarin** aims to propose an optimal dose of warfarin (a blood thinner medication) conditioned on the patient's pharmacogenetic variables (Consortium, 2009); (2) **HIV** an antiretroviral medication regiment based on the patient's HIV viral mutations (Rhee et al., 2003); (3) **Breast** and (4) **Lung** an optimal treatment strategy for patients diagnosed with breast or non-small cell lung cancer (NSCLC), respectively; and (5) **ADR** a prediction of a patient's risk of an adverse drug reaction (ADR) following the administration of a proprietary drug. We simulate a distribution shift between the observations used to learn the surrogate model $\hat{f}$ and the ground-truth objective $f$; see **Supplementary Table S1** for details.

**Prior knowledge generation.** We operationalize the task of prior knowledge synthesis as a tool-calling problem (Schick et al., 2023; Qin et al., 2024; Yao et al., 2023). We provide the LLM a set of external knowledge tools, including: (1) a corpus of medical textbooks (Xiong et al., 2024); prompting an auxiliary (2) MedGemma 27B LLM (Sellergren et al., 2025) fine-tuned on expert medical knowledge; querying structured biomedical knowledge graphs (3) HetioNet (Himmelstein et al., 2017) and (4) PrimeKG (Chandak et al., 2023); and other domain-specific knowledge repositories including (5) Cellosaurus (Bairoch, 2018) with cell-line data, (6) COSMIC (Tate et al., 2019) with cancer mutation data, (7) GDSC (Yang et al., 2013) with drug sensitivity data, and (8) DepMap (Tsherniak et al., 2017) with cancer cell dependencies. Using these tools, the LLM composes a prior knowledge statement, which is then included in the LLM prompts during optimization.

**Experiment implementation.** Each task includes two static datasets $\mathcal{D}_{\text{src}}^{\text{annotated}} = \left\{(x_j, z_j, y_j)\right\}_{j=1}^{n_{\text{src}}}$ and $\mathcal{D}_{\text{tgt}}^{\text{annotated}} = \left\{(x_i, z_i, y_i)\right\}_{i=1}^{n_{\text{tgt}}}$ from the observation space $\mathcal{X} \times \mathcal{Z} \times \mathbb{R}$. The source dataset $\mathcal{D}_{\text{src}}^{\text{annotated}}$ and target dataset $\mathcal{D}_{\text{tgt}}^{\text{annotated}}$ are constructed according to a distribution shift between the task-specific source and target distributions, and are non-overlapping at the patient level. We learn a task-specific surrogate model $\hat{f} : \mathcal{X} \times \mathcal{Z} \to \mathbb{R}$ on $\mathcal{D}_{\text{src}}^{\text{annotated}}$, and also a function $f : \mathcal{X} \times \mathcal{Z} \to \mathbb{R}$ on $\mathcal{D}_{\text{tgt}}^{\text{annotated}}$ taken to be the ground-truth objective for patients from the target population for the purposes of evaluation. Note that the full datasets $\mathcal{D}_{\text{src}}^{\text{annotated}}, \mathcal{D}_{\text{tgt}}^{\text{annotated}}$ are used only to learn $\hat{f}, f$ for our experimental setup; we project $\mathcal{D}_{\text{tgt}}^{\text{annotated}}$ onto $\mathcal{Z}$ and $\mathcal{D}_{\text{src}}^{\text{annotated}}$ onto $\mathcal{X}$ to construct $\mathcal{D}_{\text{tgt}} := \{z_i\}_{i=1}^{n_{\text{tgt}}}$ and $\mathcal{D}_{\text{src}} := \{x_j\}_{j=1}^{n_{\text{src}}}$ for our experiments. We embed proposed patient-design pairs, represented in natural language, using the `text-embedding-3-small` model from OpenAI, and perform $k$-means clustering (trained on the source dataset $\mathcal{D}_{\text{src}}$ using cosine similarity as the distance metric) in the embedding space to assign individual designs to equivalence classes—see **Appendix C** for additional details.

LEON also involves training a source critic model $c^* : \mathcal{X} \to \mathbb{R}$ as in (1). Consistent with prior work (Yao et al., 2025b; 2024), we implement $c^*$ as a fully-connected network with two hidden layers each with 2048 dimensions. We enforce the constraint on the critic's Lipschitz constant by clipping the model parameters to have an $\ell_\infty$-norm no greater than 0.01 after each weight update step, consistent with Arjovsky et al. (2017).[2] After each acquisition step during optimization, we re-train the source critic using gradient descent with a learning rate of $\eta = 0.001$ according to (1), and also perform a single-step update to the $\lambda$ certainty parameter according to (10) with $\eta_\lambda = 0.1$. We fix $W_0 = 1.0$ in (10) and the LLM temperature hyperparameter $\tau = 1.0$, and use a sampling batch size of 32 to avoid overfitting any of these hyperparameters to any particular task. All experiments were run on an internal cluster using only a single NVIDIA A100 80GB GPU. We report experimental results on a random sample of $n = 100$ unique patients from $\mathcal{D}_{\text{tgt}}$.

**Baselines.** We evaluate LEON against the LLM-based optimization methods (1) **L**arge **LA**nguage **M**odels to enhance **B**ayesian **O**ptimization (**LLAMBO**, Liu et al. (2024c)), which leverages LLMs to augment traditional Bayesian Optimization (BO); (2) **O**ptimization by **PRO**mpting (**OPRO**, Yang et al. (2024a)), which appends previously proposed solutions and their scores to subsequent LLM input prompts; and (3) **E**volution-driven **u**niversal **re**ward **k**it for **a**gent (**Eureka**, Ma et al. (2024b)),

---

[2]In general, computing the global Lipschitz constant of an arbitrary function is $\mathcal{NP}$-hard and still an open problem (Abdeen et al., 2025; Kim et al., 2021). While other methods than that proposed by Arjovsky et al. (2017) for enforcing the Lipschitz constraint exist (Scaman & Virmaux, 2018; Hu et al., 2024; Fazlyab et al., 2019; Song et al., 2023), we leave the investigation of how they might be used with LEON for future work.

Table 1: **Quality of patient-conditioned designs under distribution shift.** We report mean $\pm$ standard error of mean (SEM) ground-truth objective values achieved by the single proposed design for a given patient, averaged over $n = 100$ test patients from the target distribution. **Bolded** (resp., Underlined) cells indicate the **best** (resp., second best) mean score for a given task.

| Method | Warfarin $\downarrow$ RMSE Loss (*mg/week*) | HIV $\downarrow$ Viral Load (*copies/mL*) | Breast $\uparrow$ TTNTD (*months*) | Lung $\uparrow$ TTNTD (*months*) | ADR $\downarrow$ NLL Loss (*no units*) | Rank |
|---|---|---|---|---|---|---|
| Majority | $3.46 \pm 0.70$ | $4.55 \pm 0.07$ | $25.95 \pm 0.75$ | $20.13 \pm 0.13$ | $\mathbf{1.41 \pm 0.05}$ | 8.0 |
| Human | $2.68 \pm 0.86$ | $4.55 \pm 0.07$ | $29.65 \pm 1.14$ | $21.10 \pm 0.27$ | — | 8.5 |
| Grad. | $\underline{1.37 \pm 0.13}$ | $\underline{4.52 \pm 0.04}$ | $65.23 \pm 2.03$ | $24.09 \pm 0.44$ | $23.7 \pm 1.7$ | 5.2 |
| BO-qEI | $\mathbf{1.36 \pm 0.13}$ | $4.53 \pm 0.04$ | $67.05 \pm 1.87$ | $27.97 \pm 0.65$ | $23.2 \pm 1.7$ | $\underline{3.4}$ |
| Sim. Anneal | $1.38 \pm 0.12$ | $4.55 \pm 0.03$ | $66.62 \pm 2.62$ | $\underline{29.29 \pm 0.74}$ | $23.8 \pm 1.7$ | 5.0 |
| CMA-ES | $1.90 \pm 0.14$ | $4.53 \pm 0.04$ | $59.48 \pm 2.76$ | $27.43 \pm 0.68$ | $23.4 \pm 1.7$ | 6.2 |
| GA | $1.49 \pm 0.26$ | $4.62 \pm 0.05$ | $\underline{69.90 \pm 2.28}$ | $27.53 \pm 0.81$ | $20.0 \pm 2.3$ | 5.2 |
| LLAMBO | $3.28 \pm 0.10$ | $\underline{4.52 \pm 0.05}$ | $48.83 \pm 2.48$ | $20.60 \pm 0.31$ | $20.6 \pm 1.9$ | 7.0 |
| OPRO | $1.40 \pm 0.13$ | $4.55 \pm 0.04$ | $55.68 \pm 2.86$ | $24.35 \pm 0.43$ | $23.8 \pm 1.7$ | 7.0 |
| Eureka | $1.54 \pm 0.25$ | $4.58 \pm 0.04$ | $63.48 \pm 3.52$ | $25.10 \pm 0.69$ | $21.3 \pm 2.0$ | 6.8 |
| LEON | $\mathbf{1.36 \pm 0.13}$ | $\mathbf{4.50 \pm 0.04}$ | $\mathbf{72.43 \pm 2.86}$ | $\mathbf{32.71 \pm 0.32}$ | $\underline{12.4 \pm 1.6}$ | $\mathbf{1.2}$ |

which extends OPRO by intermittently prompting the LLM to reflect on the efficacy of previous optimization strategies in natural language. All LLM-based optimization strategies (including LEON) were evaluated using `gpt-4o-mini-2024-07-18` from OpenAI without any fine-tuning.

We also compare LEON to traditional, non-LLM-based optimization methods (4) Gradient Ascent (**Grad.**); (5) Simulated Annealing (**Sim. Anneal**, Tsallis (1988)); (6) Covariance Matrix Adaptation Evolution Strategy (**CMA-ES**, Hansen (2006)); (7) Genetic Algorithm (**GA**, Gad (2024)); and (8) Bayesian optimization with Expected Improvement (**BO-qEI**). Finally, we evaluate the (9) **Majority** baseline algorithm that always proposes the majority design (i.e., mean for continuous design dimensions and mode for discrete design dimensions) across all observations $x_j \in \mathcal{D}_{\text{src}}$; and the (10) **Human** baseline algorithm, which proposes the true design $x_i$ that the individual $z_i$ actually received according to the hidden annotated dataset $\mathcal{D}_{\text{tgt}}^{\text{annotated}}$. An optimization method that outperforms the Majority strategy suggests that the optimizer's personalized treatment strategies outperform the 'average' treatment in the source dataset $\mathcal{D}_{\text{src}}$ in expectation. An optimization method that outperforms the Human baseline means the treatment strategies proposed by the optimizer are better than the therapy the patient actually received from their clinician according to our oracle objective function.

**Evaluation metric.** For any particular input set of fixed patient covariates $z \sim \mathcal{D}_{\text{tgt}}$, an optimization method is allowed to sample and propose a total of 2048 possible designs $x$ and corresponding surrogate model evaluations $\hat{f}(x; z)$. After the surrogate model evaluation budget is exhausted, the optimizer proposes a *single* top design $x^*$ to evaluate using the ground-truth objective function $f$: for most optimization methods, $x^*$ is the proposed design that maximizes $\hat{f}$ (LEON follows the design proposal method in **Supplementary Algorithm 1**). We then evaluate this single proposed design $x^*$ using the ground-truth objective $f$, and report the corresponding score $f(x^*; z)$. Importantly, $f$ is only used for evaluation purposes and is not accessible to any method during optimization.

**Results.** LEON consistently outperforms all baseline methods, achieving an average rank of **1.2** across all tasks (**Table 1**). Notably, LEON proposes sets of personalized treatment designs that are superior to the treatments retrospectively received by the patients. It achieves the best performance on Warfarin dose prediction, HIV treatment, and both breast and lung cancer therapy design. Qualitatively, we found that the Majority algorithm was able to outperform LEON on the ADR task due to significant class imbalance in the population of patients in the dataset.

Additional empirical analysis of LEON is included in **Appendix D**. In **Appendix D.2**, we evaluate the monetary cost, token usage, and environmental impact of our method. We then interrogate the construction of prior knowledge used by LEON in **Appendices D.3-D.4**, and evaluate how the certainty parameters $\mu$ and $\lambda$ vary over the course of optimization in **Appendices D.5-D.6**. Finally, we investigate the fairness of LEON with respect to patient gender and race in **Appendix D.10**.

**Ablation studies.** In our main experiments, we use `gpt-4o-mini-2024-07-18` as the LLM optimizer; we ablate the choice of language model in **Supplementary Table S5**. We also ablate

the values of the certainty parameters $\lambda, \mu$ in **Supplementary Table S6**. We then ablate the external sources of prior knowledge in **Supplementary Table S7** and find that LEON is sensitive to the quality of domain-specific knowledge available. Finally, we examine how the performance of LEON is affected factors such as the severity of distribution shift between source and target distributions (**Appendix E.6**); the use of reflection in LEON (**Appendix E.10**); and the choice of equivalence relation $\sim$ in **Definition 4.1** (**Appendices E.11-E.13**). See **Appendix E** for additional results.

## 6 DISCUSSION AND CONCLUSION

In this work, we propose LEON to combine domain knowledge with LLM-based optimizers to solve conditional black-box optimization problems under distribution shift for personalized medicine. We first introduce two additional constraints in (4) that upweight high-quality designs that are both (1) in-distribution according to an auxiliary source critic model; and (2) likely to be high-quality based on the language model's statistical certainty. Using our method, we show how consumer-grade LLMs can be used to solve a wide variety of challenging personalized medicine optimization problems without any LLM fine-tuning, outperforming both traditional and other LLM-based optimization methods. Moving forward, we hope to extend LEON to the setting of active learning and prospective clinical evaluation (Collignon et al., 2020; Ianevski et al., 2021; Kuru et al., 2024; Burd et al., 2020), and also apply our method to other domains outside of personalized medicine.

**Limitations.** Several limitations warrant consideration. First, we observed that LLM optimizers using LEON are sensitive to the available prior knowledge: contamination with factually incorrect or outdated information may propagate into optimization outputs and adversely affect personalized treatment design using LEON (Omar et al., 2025; Han et al., 2024; Clusmann et al., 2025). Furthermore, while the clinical tasks considered here are designed to approximate real-world conditions, they cannot fully capture the complexity of heterogeneous patient responses and rare disease subtypes encountered in actual clinical practice (Pan et al., 2024). Finally, as with all simulation-based benchmarks, the validity of our conclusions is limited by the assumptions embedded in the functions and datasets used, which may obscure differences between methods. Future work might explore how to integrate physicians in the loop to mitigate the risks of autonomous LLM-based systems.

### ETHICS STATEMENT

This work investigates the use of large language models (LLMs) as optimizers for conditional black-box problems in clinical medicine. All datasets used in this work are fully anonymized and contain no personally identifiable information. As such, no patient consent or institutional review board (IRB) approval was required. Separately, LLMs are inherently subject to social and demographic biases learned during pretraining (Sorin et al., 2025; Omiye et al., 2023). When applied to patient-specific design tasks, such biases could disproportionately affect marginalized populations, potentially leading to inequitable or unsafe treatment proposals if not properly accounted for. We also emphasize that LEON is not intended for direct clinical use, but rather as a methodological contribution toward future systems that may aid clinical decision making.

### REPRODUCIBILITY STATEMENT

The datasets for the Warfarin and HIV tasks are made publicly available by Consortium (2009) and Rhee et al. (2003), respectively. The Breast and Lung task data were made available by Flatiron Health; the de-identified data may be made available upon reasonable request by contacting `DataAccess@flatiron.com`. The ADR task data is an internal proprietary dataset—data access requests may be directed to the corresponding author. Our custom code implementation for our experiments is made publicly available at `code.roche.com/braid/projects/leon`.

### ACKNOWLEDGMENTS

The authors thank Allison Chae (Main Line Health) for her help with the qualitative evaluation of the language model outputs. M.S.Y. is supported by NIH Award F30 MD020264 and completed this work during an internship at Genentech. OB is supported by NSF Award CCF-1917852. The views and conclusions contained herein are those of the authors and should not be interpreted as representing the official policies, either expressed or implied, of Genentech or the U.S. Government.

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

## APPENDIX

**Table of Contents**

## A  PROOFS

**Lemma 4.2** (Design Collapse Within Equivalence Classes). Using the method of Lagrange multipliers, we can rewrite (4) as a function of the partial Lagrangian $\mathcal{L}_\lambda(q)$ for some constant $\lambda \in \mathbb{R}_+$:

$$\arg\max_{q(x)\in p(\mathcal{X})} \quad \mathcal{L}_\lambda(q) := \mathbb{E}_{x\sim q(x)}[\hat{f}(x;z)] + \lambda \left(W_0 - \mathbb{E}_{x\sim\mathcal{D}_{\mathrm{src}}}[c^*(x)] + \mathbb{E}_{x\sim q(x)}[c^*(x)]\right)$$
$$\text{s.t.} \quad \mathcal{H}_\sim(q(x)) \leq H_0$$

Suppose there exists a distribution $q(x)$ that satisfies the remaining constraint in (5). Furthermore, assume that the function $\hat{f}(x) + \lambda c^*(x)$ is continuous everywhere and coercive in $\mathcal{X}$. For all $N$ equivalence classes, we can then define $x_i^*$ (not necessarily unique) according to

$$x_i^*(\lambda) := \arg\max_{x\in[x]_i} \left(\hat{f}(x;z) + \lambda c^*(x)\right)$$

Then, the alternative distribution $q^*(x) = \sum_{i=1}^N \bar{q}_i \, \delta(x - x_i^*)$, where $\bar{q}_i$ is as in **Definition 4.1**, also satisfies the constraint and simultaneously achieves a non-inferior value $\mathcal{L}_\lambda(q^*) \geq \mathcal{L}_\lambda(q)$.

*Proof.* First, note that each $x_i^*$ is in a distinct equivalence class, since equivalence classes in $\mathcal{X}/\sim$ are pairwise disjoint by construction. The probability $q_i^*$ of sampling from the $i$th equivalence class according to $q^*(x)$ is then

$$q_i^* = \int_{[x]_i} dx\, q^*(x) = \int_{[x]_i} dx \sum_{j=1}^N \bar{q}_j\, \delta(x - x_j^*) = \int_{[x]_i} dx\, \bar{q}_i\, \delta(x - x_i^*) = \bar{q}_i \qquad (12)$$

since $\bar{q}_i \neq \bar{q}_i(x)$ and $[x]_i \cap [x]_j = \varnothing$ when $i \neq j$. Because this holds for all $i$, it follows that $\mathcal{H}_\sim(q^*(x)) = \mathcal{H}_\sim(q(x))$, meaning $\mathcal{H}_\sim(q^*(x)) \leq H_0$ as well. Secondly, observe

$$\mathcal{L}_\lambda(q^*) = \mathbb{E}_{x\sim q^*(x)}[\hat{f}(x)] + \lambda \left(W_0 - \mathbb{E}_{x\sim\mathcal{D}_{\mathrm{src}}}[c^*(x)] + \mathbb{E}_{x\sim q^*(x)}[c^*(x)]\right)$$
$$= \mathbb{E}_{x\sim q^*(x)}\left[\hat{f}(x) + \lambda c^*(x)\right] + \lambda \left(W_0 - \mathbb{E}_{x\sim\mathcal{D}_{\mathrm{src}}}[c^*(x)]\right)$$
$$= \sum_{i=1}^N q_i^* \left(\hat{f}(x_i^*) + \lambda c^*(x_i^*)\right) + \lambda \left(W_0 - \mathbb{E}_{x\sim\mathcal{D}_{\mathrm{src}}}[c^*(x)]\right)$$

Using the expansion of $q_i^*$ from (12),

$$\mathcal{L}_\lambda(q^*) = \left[\sum_{i=1}^N \int_{[x]_i} dx\, q(x)[\hat{f}(x_i^*) + \lambda c^*(x_i^*)]\right] + \lambda \left(W_0 - \mathbb{E}_{x\sim\mathcal{D}_{\mathrm{src}}}[c^*(x)]\right)$$
$$\geq \left[\sum_{i=1}^N \int_{[x]_i} dx\, q(x)(\hat{f}(x) + \lambda c^*(x))\right] + \lambda \left(W_0 - \mathbb{E}_{x\sim\mathcal{D}_{\mathrm{src}}}[c^*(x)]\right)$$
$$= \mathbb{E}_{x\sim q(x)}(\hat{f}(x) + \lambda c^*(x)) + \lambda \left(W_0 - \mathbb{E}_{x\sim\mathcal{D}_{\mathrm{src}}}[c^*(x)]\right)$$
$$= \mathbb{E}_{x\sim q(x)}[\hat{f}(x)] + \lambda \left(W_0 - \mathbb{E}_{x\sim\mathcal{D}_{\mathrm{src}}}[c^*(x)] + \mathbb{E}_{x\sim q(x)}[c^*(x)]\right)$$
$$= \mathcal{L}_\lambda(q)$$

using the definition of each $x_i^*$ as in (6). The claim follows. $\qquad\square$

**Lemma 4.3** (Probabilistic Sampling Over Equivalence Classes). Consider the constrained optimization problem as in (7). The $i$th element of the $N$-dimensional vector $\bar{q}$ can be written as

$$\bar{q}_i = \exp\left[\mu\left(\hat{f}(x_i^*;z) + \lambda c^*(x_i^*)\right)\right] \Big/ \mathcal{Z}(\lambda) \quad \text{where} \quad x_i^* := \arg\max_{x\in[x]_i}\left(\hat{f}(x;z) + \lambda c^*(x)\right)$$

where $\mathcal{Z}(\lambda)$ is a normalizing constant and $\lambda, \mu^{-1} \in \mathbb{R}_+$ are the Lagrange multipliers.

*Proof.* By definition (Boyd & Vandenberghe, 2004), the full Lagrangian $\mathcal{L}(\bar{q}; \lambda, \mu)$ of (7) is

$$
\begin{aligned}
\mathcal{L}(\bar{q}; \lambda, \mu) &= \sum_{i=1}^{N} \bar{q}_i \hat{f}(x_i^*; z) + \lambda \left( W_0 - \mathbb{E}_{x \sim \mathcal{D}_{\text{src}}}[c^*(x)] + \sum_{i=1}^{N} \bar{q}_i c^*(x_i^*) \right) \\
&\quad + \mu^{-1} \left( -H_0 - \sum_{i=1}^{N} \bar{q}_i \log \bar{q}_i \right) \\
&= \sum_{i=1}^{N} \bar{q}_i \left( \hat{f}(x_i^*; z) + \lambda c^*(x_i^*) \right) + \lambda \left( W_0 - \mathbb{E}_{x \sim \mathcal{D}_{\text{src}}}[c^*(x)] \right) \\
&\quad - \mu^{-1} \left( H_0 + \sum_{i=1}^{N} \bar{q}_i \log \bar{q}_i \right)
\end{aligned}
\tag{13}
$$

where $\lambda, \mu^{-1} \in \mathbb{R}_+$ are the Lagrangian multipliers associated with each of the two respective constraints. The stationarity condition of the Karush–Kuhn–Tucker (KKT) theorem and additive decomposition of the individual scalar elements $\bar{q}_i$ of $\bar{q}$ in (13) give

$$
\frac{\partial \mathcal{L}}{\partial \bar{q}_i} = 0 = \left( \hat{f}(x_i^*; z) + \lambda c^*(x_i^*) \right) - \mu^{-1} \left( \log \bar{q}_i + 1 \right)
$$

by definition. Rearranging gives

$$
\mu^{-1}(\log \bar{q}_i + 1) = \hat{f}(x_i^*; z) + \lambda c^*(x_i^*) \implies \log \bar{q}_i = -1 + \mu \left( \hat{f}(x_i^*; z) + \lambda c^*(x_i^*) \right)
$$

and so

$$
\bar{q}_i \propto \exp \left[ \mu \left( \hat{f}(x_i^*; z) + \lambda c^*(x_i^*) \right) \right]
$$

Defining the partition function $\mathcal{Z}(\lambda) := \sum_{i=1}^{N} \exp \left[ \mu \left( \hat{f}(x_i^*; z) + \lambda c^*(x_i^*) \right) \right]$ as our normalizing constant, the elements of the probability vector $\bar{q}$ can be written as

$$
\bar{q}_i = \exp \left[ \mu \left( \hat{f}(x_i^*; z) + \lambda c^*(x_i^*) \right) \right] \Big/ \mathcal{Z}(\lambda)
$$

where $x_i^*$ is as in (6). $\qquad \square$

**Corollary 4.4** (Dual Function of (7))**.** The dual function of the constrained problem in (7) is

$$
g(\lambda, \mu) = \lambda(W_0 - \mathbb{E}_{x \sim \mathcal{D}_{\text{src}}}[c^*(x)]) + \mu^{-1} H_0 + \mu^{-1} \log \mathcal{Z}(\lambda)
$$

where $\mathcal{Z}(\lambda)$ is the normalizing constant from (8), and so

$$
\frac{\partial g(\lambda, \mu)}{\partial \lambda} = W_0 - \left( \mathbb{E}_{x \sim \mathcal{D}_{\text{src}}}[c^*(x)] - \sum_i \bar{q}_i c^*(x_i^*) \right)
$$

*Proof.* From (13) and referencing Boyd & Vandenberghe (2004), the Lagrangian dual function is

$$
\begin{aligned}
g(\lambda, \mu) &:= \max_{\bar{q} \in \Delta(N)} \mathcal{L}(\bar{q}; \lambda, \mu) \\
&= \sum_{i=1}^{N} \bar{q}_i (\hat{f}(x_i^*; z) + \lambda c^*(x_i^*)) + \lambda(W_0 - \mathbb{E}_{x \sim \mathcal{D}_{\text{src}}}[c^*(x)]) - \frac{1}{\mu} \left( H_0 + \sum_{i=1}^{N} \bar{q}_i \log \bar{q}_i \right)
\end{aligned}
$$

where the optimal vector $\bar{q}$ is as in **Lemma 4.3**. Note that the first summation can be written as

$$
\sum_{i=1}^{N} \bar{q}_i (\hat{f}(x_i^*; z) + \lambda c^*(x_i^*)) = \frac{1}{\mu} \sum_{i=1}^{N} \bar{q}_i \left( \mu \left[ \hat{f}(x_i^*; z) + \lambda c^*(x_i^*) \right] \right) = \frac{1}{\mu} \sum_{i=1}^{N} \bar{q}_i (\log \bar{q}_i + \log \mathcal{Z}(\lambda))
$$

$$
= \frac{1}{\mu} \log \mathcal{Z}(\lambda) \sum_{i=1}^{N} \bar{q}_i + \frac{1}{\mu} \sum_{i=1}^{N} \bar{q}_i \log \bar{q}_i = \frac{1}{\mu} \log \mathcal{Z}(\lambda) + \frac{1}{\mu} \sum_{i=1}^{N} \bar{q}_i \log \bar{q}_i
$$

Substituting into our expression for $g(\lambda, \mu)$,

$$g(\lambda, \mu) = \mu^{-1} \log \mathcal{Z}(\lambda) + \lambda(W_0 - \mathbb{E}_{x \sim \mathcal{D}_{\text{src}}}[c^*(x)]) - \mu^{-1} H_0$$

Differentiating with respect to $\lambda$,

$$\frac{\partial g(\lambda, \mu)}{\partial \lambda} = \frac{1}{\mu} \frac{1}{\mathcal{Z}(\lambda)} \frac{\partial \mathcal{Z}(\lambda)}{\partial \lambda} + (W_0 - \mathbb{E}_{x \sim \mathcal{D}_{\text{src}}}[c^*(x)])$$

$$= \frac{1}{\mu} \sum_{i=1}^{N} \left[ \frac{1}{\mathcal{Z}(\lambda)} \exp\left[ \mu(\hat{f}(x_i^*; z) + \lambda c^*(x_i^*)) \right] \cdot \mu c^*(x_i^*) \right] + (W_0 - \mathbb{E}_{x \sim \mathcal{D}_{\text{src}}}[c^*(x)])$$

$$= \sum_{i=1}^{N} c^*(x_i^*) \exp\left[ \mu(\hat{f}(x_i^*; z) + \lambda c^*(x_i^*)) \right] \Big/ \mathcal{Z}(\lambda) + (W_0 - \mathbb{E}_{x \sim \mathcal{D}_{\text{src}}}[c^*(x)])$$

$$= \sum_{i=1}^{N} \bar{q}_i c^*(x_i^*) + (W_0 - \mathbb{E}_{x \sim \mathcal{D}_{\text{src}}}[c^*(x)]) = W_0 - \left( \mathbb{E}_{x \sim \mathcal{D}_{\text{src}}}[c^*(x)] - \sum_{i=1}^{N} \bar{q}_i c^*(x_i^*) \right)$$

after regrouping terms. $\square$

# B  ADDITIONAL RELATED WORK

**Generalization and domain adaptation.** To mitigate the mismatch between the available surrogate model and hidden ground-truth objective, recent work on domain adaptation leverages knowledge of the underlying source and target distributions to overcome the underlying distribution shift. Broadly, the technique relies on learning a mapping between source and target distributions, and can often be applied even to black-box surrogate models (Ganin et al., 2016; Sun & Saenko, 2016; Bousmalis et al., 2016; Yue et al., 2023). Many existing domain adaptation methods have in proposed specifically for applications in computer vision (Bousmalis et al., 2017; Saito et al., 2018; French et al., 2018), which are outside the scope of this work. Separately, Tzeng et al. (2017) learn to encode target samples into a source representation space using adversarial feedback, and Ngo et al. (2022) describe a method for task-level adaptation. However, these methods assume knowledge of the target distribution, which may not be known *a priori* in optimization tasks. In our setting, we show how LEON can be used for optimization on a per-patient basis without assuming any prior knowledge of other patient observations from the target distribution—a realistic assumption for protecting patient privacy and ensuring timely downstream decision making in real-world applications. We therefore cannot leverage domain adaptation to adapt a surrogate model during optimization.

**Conformal prediction.** Conformal prediction is a statistical framework for uncertainty quantification where the goal is to produce prediction *sets* as opposed to singular outputs, providing a guarantee that the true label is almost certainly in the prediction set with high probability (Cresswell et al., 2024; Angelopoulos & Bates, 2023; Angelopoulos et al., 2021). Prior work has demonstrated how conformal prediction can be leveraged for image classification (Angelopoulos et al., 2021; Zhang et al., 2025a), disease course prediction (Yang et al., 2024b; Sreenivasan et al., 2025), drug discovery (Eklund et al., 2013; Jin & Candes, 2023), and even autoregressive generative models (Quach et al., 2024). Prior work has explored how to perform Bayesian (Deshpande et al., 2024; Stanton et al., 2023) and worst-case (Johnstone & Cox, 2021) optimization over conformal prediction sets; future work might explore how to similarly adapt LEON to optimize over prediction sets.

**Robustness via domain-specific foundation models.** A potential direction for building more robust and performant surrogate models is to leverage large-scale foundation models as surrogates. One argument is that models that are trained on sufficiently large and diverse datasets are unlikely to encounter significantly 'out-of-distribution' inputs, yielding more robust performance across a wide range of inputs during optimization. Early efforts along these lines have emerged in areas such as cell perturbation modeling (Cui et al., 2024; Hao et al., 2024; Zeng et al., 2025) and drug discovery (Abramson et al., 2024; Huang et al., 2024). However, despite preliminary efforts to extend similar approaches to clinical medicine (Guo et al., 2024a; 2023; Steinberg et al., 2024), the limited availability of patient training data and stringent patient privacy constraints restrict the feasibility of constructing models that reliably generalize in a zero-shot manner across many different patient populations. For these reasons, methods like LEON may still play a role in domains such as clinical medicine even as the accuracy of surrogate models improves with foundation model development.

Table S1: **Implementation details.** We consider the problem of optimization under distribution shift; given access to a surrogate model trained on a source dataset, we want to find an optimal design for a patient sampled from the target distribution that maximizes the response according to a hidden ground-truth objective function. *TabPFN*: Tabular prior-data fitted network (Hollmann et al., 2025). *FCNN*: Fully connected neural network. *ResNet*: Tabular residual network (Gorishniy et al., 2021). *GBDT*: Gradient boosted decision tree ensemble (Chen & Guestrin, 2016; Quinlan, 1986).

| Distribution Shift | Covariate Shift in $P(X)$ | | | | Label Shift in $P(Y)$ |
|---|---|---|---|---|---|
| **Task** | **Warfarin** | **HIV** | **Breast** | **ADR** | **Lung** |
| **Shifted Feature** | Patient Race | Study Year | Patient Age | Medication | TTNTD |
| **Source Data Feature** | White | 2002-2008 | $< 65$ years old | Drug A | Longer |
| **Target Data Feature** | Non-White | 2009-2020 | $\geq 65$ years old | Drug B | Shorter |
| **Source Dataset Size** | 3,095 | 740 | 3,020 | 484 | 3,013 |
| **Target Dataset Size** | 2,646 | 554 | 1,260 | 554 | 3,009 |
| **Surrogate Model** | TabPFN | FCNN | GBDT | GBDT | GBDT |
| **Ground-Truth Objective** | Exact | ResNet | GBDT | Exact | GBDT |
| **Patient Feature Size** ($|z|$) | 11 | 339 | 47 | 50 | 47 |
| **Treatment Dimensions** ($|x|$) | 1 | 16 | 30 | 5 | 30 |
| **Optimization Type** | Continuous | Discrete | Discrete | Continuous | Discrete |

**Medical reasoning models.** Recent work has explored how to develop datasets of clinical reasoning traces (Wu et al., 2025; Kwon et al., 2024; Hager et al., 2024) and associated benchmarks (Arora et al., 2025; Chen et al., 2025b). This evolving ecosystem of model training and evaluation resources have made available recent medical reasoning models: Singhal et al. (2023; 2025) introduced the MedPaLM family of models that generate accurate long-form answers to consumer medical questions, and Zhang et al. (2025b) fine-tune language models using reinforcement learning with verifiable rewards derived from multiple choice question answering. Separately, Savage et al. (2024); Sonoda et al. (2024) demonstrate how chain-of-thought reasoning can be elicited from base language models by optimized prompting strategies. Given the impressive performance of these models, one might imagine how they could be used to also predict personalized treatment strategies. However, we note that simply prompting a specialized medical language model to return a treatment response is not equivalent to iterative optimization with a language model against a black-box evaluator (i.e., the surrogate model). For this reason, simply querying an LLM to return a treatment strategy for a patient is outside the scope of our work on *optimization* strategies herein.

**Existing methods in personalized medicine.** Aside from the methodology-focused related works, we also provide a brief overview of existing applied works in personalized medicine. Recent clinical trials across clinical disciplines have tested personalized approaches by selecting or adapting treatments for subgroups or individuals (He et al., 2018). de Bono et al. (2020); Paik et al. (2020) are recent randomized clinical trials that leverage patient-specific genomic data to assign patients to a predetermined experimental treatment arm. In current clinical practice, certain pharmacogenetic variables already help determine whether certain medications are prescribed to particular patients (Ma et al., 2010; Ford & Taubert, 2013; Kirchheiner et al., 2007). Using machine learning to propose entirely new treatment arms for individual patients is the ultimate goal of personalized medicine, but is not yet a core component of clinical practice (Johnson et al., 2020; Chen et al., 2025c).

## C  ADDITIONAL IMPLEMENTATION DETAILS

### C.1  OPTIMIZATION TASK SPECIFICATIONS

**Supplementary Table S1** outlines task-specific implementation details, including information on the underlying distribution shift and number of design dimensions for each task. Below, we further discuss the motivation behind and implementation of each optimization task considered in our work.

**Warfarin.** Warfarin is an oral anticoagulant medication used to treat and prevent a variety of thromboembolic conditions, such as atrial fibrillation (Brass et al., 1997), venous thromboembolisms (Ramzi & Leeper, 2004), and maintenance therapy after mechanical valve placement (Catterall et al., 2020). However, determining its initial dose is notoriously complex, in part due to the medication's narrow therapeutic index and significant inter-patient variability (Lee & Klein, 2013; Consortium,

2009). Dosing is influenced by numerous factors including age, body weight, liver function, and vitamin K metabolism. These complexities make standardized dosing unreliable and necessitate careful titration using frequent blood tests to achieve and maintain therapeutic anticoagulation.

Consortium (2009) previously introduced a de-identified, publicly available dataset of 5,052 patients from multiple hospital sites with clinical indications for warfarin initiation. The dataset includes both clinical and pharmacogenetic observations for all patients, in addition to their prescribed warfarin dose and laboratory test results. More explicitly, each patient observation includes a prescribed warfarin dose $x \in \mathcal{X} \subseteq \mathbb{R}$ and conditioning vector $z$ that concatenates the patient's height (caDSR 649); weight (caDSR 2179689); age (NIH Concept ID C25150); sex at birth (NIH Concept ID C124436); smoking status (NLM m1Sgwj5s2ig); the use of medications known to affect warfarin dosing (carbamazepine, phenytoin, rifampin, amiodarone); and the presence of genotype variants of CYP2C9 and VKORC1 genes. Using this dataset, Consortium (2009) learn an expert function dose : $\mathcal{Z} \rightarrow \mathcal{X}$ (validated by physician specialists) that predicts the optimal dose for a given patient. The optimization objective for our task is to minimize the root mean squared error (RMSE) $||x - \text{dose}(z)||_2$ (i.e., to find the warfarin dose predicted by the expert function).

**HIV.** Human immunodeficiency virus (HIV) is a virus that attacks the human immune system, and is currently treated with antiretroviral therapy (ART) (Kemnic & Gulick, 2025). Choosing the correct ART drug regimen is a carefully tailored, patient-centered process driven largely by viral genotype and individual health factors (Rhee et al., 2003). Before initiating treatment, gold-standard practice involves performing genotypic resistance testing to identify mutations in the HIV reverse transcriptase (RT) and protease sequences, as certain mutations can potentially render certain ART drugs ineffective (Saag et al., 2020). Treatment planning becomes even more complex in patients with transmitted or archived drug resistance (Kemnic & Gulick, 2025). For these reasons, a personalized approach to HIV treatment can help more effectively tailor ART to both viral mutation profiles and patient variables. The efficacy of an ART drug panel is clinically evaluated by measuring the patient's *HIV viral load* (i.e., the 'amount' of HIV in the patient's blood) before and after therapy.

We use the publicly available HIVDB dataset from Rhee et al. (2003) to study how to personalize ART treatment regimens for patients based on their viral genotyping data. A patient's corresponding conditioning vector $z$ was constructed by concatenating the positions and amino acid substitutions of the patient's entire HIV protease sequence and the first 240 amino acids of the p66 subunit of the HIV reverse transcriptase protein following prior work. We frame the problem of designing personalized ART therapies as a binary optimization problem; the design space consists of 16 ART drugs: 8 Nucleoside RT Inhibitors (Abacavir, Zidovudine, Didanosine, Emtricitabine, Lamivudine, Stavudine, Tenofovir, and Zalcitibine); 3 Non-Nucleoside RT Inhibitors (Delavirdine, Efavirenz, and Nevirapine); and 5 Protease Inhibitors (Atazanavir, Indinavir, Nelfinavir, Ritonavir, and Saquinavir). We chose these 16 drugs because they were the most commonly used drugs in the HIVDB dataset. Each vector $x \in \{0, 1\}^{16}$ in the design space specifies the ART regimen to prescribe the patient, where the $i$th dimension $x_i = 1$ if the $i$th drug is included in the regimen.

**Breast and Lung.** Breast cancer and lung cancer are among the most prevalent cancers diagnosed in the United States (Kim et al., 2025; Xu et al., 2024; Ganti et al., 2021; Zhou et al., 2024). There has been recent interest in personalizing oncologic treatments based on a patient's unique genetic and environmental makeup (Paik et al., 2020; Papalexis et al., 2024). To this end, we use the following set of features to construct the conditioning vector $z$: (1) age (NIH Concept ID C25150); (2) sex at birth (NIH Concept ID C124436); (3) race (NLM LakF0YkywC); (4) socioeconomic index; (5) cancer TNM staging; (6) vitals (excluding height and weight) (NLM GOr1OFJAKC; and (7) any labs and biomarkers available for the patient. The following blood laboratory values and biomarker tests were used to construct $z$ if available: 4 numerical complete blood count (CBC, NLM DZwbeRugB) tests (Hematocrit, White blood cell count, Hemoglobin, Platelet count); 6 numerical basic metabolic panel (BMP) tests (Sodium, Potassium, Chloride, Creatinine, Blood urea nitrogen, and Glucose); 6 numerical liver function tests (LFTs) (Aspartate aminotransferase, Alanine aminotransferase, Alkaline Phosphatase, Albumin, Total bilirubin, and Total serum protein); 4 numerical miscellaneous tests (Serum calcium, Estimated glomerular filtration rate, Glomerular filtration rate, and Lactate dehydrogenase); and 15 categorical biomarker tests (Human epidermal growth factor receptor 2 (HER2), Progesterone receptor (PR), Estrogen receptor (ER), Ki67, Anaplastic lymphoma kinase (ALK), BRAF, Kirsten rat sarcoma viral oncogene homolog (KRAS), ROS1, RET, MET, Programmed cell death ligand 1 (PDL1), tumor protein 53 (TP53), Kelch-like ECH-associated protein 1 (KEAP1), Serine/threonine kinase 11 (STK11), and breast cancer gene (BRCA)).

The design space for both tasks consists of (1) whether to perform surgery with curative intent (0 or 1); (2) whether to perform radiation therapy with curative intent (0 or 1); (3) whether chemotherapy should be adjuvant or neoadjuvant; and (4) the specific drug(s) to include in the chemotherapeutic regimen. The chemotherapeutic drugs to choose from include 9 cytotoxic chemotherapies (Carboplatin, Cisplatin, Cyclophosphamide, Pemetrexed, Capecitabine, Gemcitabine, Paclitaxel, Docetaxel, Doxorubicin); 3 cell cycle inhibitors (Palbociclib, Abemaciclib, Ribociclib); 5 hormonal therapies (Anastrozole, Exemestane, Letrozole, Tamoxifen, Fulvestrant); 9 monoclonal antibodies (Pembrolizumab, Nivolumab, Cemiplimab, Atezolizumab, Durvalumab, Ipilimumab, Trastuzumab, Pertuzumab, Sacituzumab); and 1 tyrosine kinase inhibitor (Osimertinib). We chose these specific 27 medications because they were the most commonly prescribed medications in the Flatiron dataset. Altogether, the design space is represented as a boolean vector in $\{0, 1\}^{30}$.

**ADR.** An adverse drug reaction (ADR) is an unintended and harmful response to a medication. Minimizing the risk of ADRs is a priority in early-stage clinical trials in order to ensure that newly tested therapies are safe for human use. In this task, we leverage two internal proprietary clinical trial datasets de-identified at the patient level that were collected from two clinical trials to predict patient risk of an ADR. The primary goal of the clinical trials was to evaluate the safety of two different medications that share similar mechanisms of action, which we refer to as 'Drug A' and 'Drug B' in **Supplementary Table S1**. Due to the sensitive nature of the internal trial data, we do not disclose the identities of Drugs A and B, or the particular ADR under investigation.

The following patient covariates were used to construct the conditioning vectors $z$: patient sex at birth (NIH Concept ID C124436); (2) patient age (NIH Concept ID C25150); (3) tumor measurements; (4) medical diagnosis; (5) ECOG score (caDSR 88); (6) vitals (excluding height and weight) (NLM GOr1OFJAKC; (7) any patient labs and/or biomarkers; and (8) medication dosing schedule. Each patient observation included a label of the severity of the observed ADR (if any) in response to the patient's therapy: we first one-hot-encode this label into a vector $x^{gt}$ with 5 dimensions (one for each severity level of the ADR, consistent with medical societal guidelines (Khan et al., 2023; Lee et al., 2019; Trotti et al., 2003)), and then perturb this vector by randomly adding a uniformly sampled per-dimension 'jitter' $\delta_i \sim \mathcal{U}[0, 1/4]$ for each zero-valued dimension $i$ to produce a vector $x \in [0, 1]^5$ such that $x_i = \delta_i$ if $x_i^{gt} = 0$ and $x_i = 1 - \sum_{j \neq i} \delta_j$ if $x_i^{gt} = 1$. In this way, the index of the maximum element in both $x$ and $x^{gt}$ are equal. We then compute the corresponding negative log-likelihood (NLL) $y := \sum_{i=1}^{5} x_i^{gt} \log x_i$ as the objective to minimize. These values are then used to construct the annotated datasets $\mathcal{D} = \{(x_j, z_j, y_j)\}_{i=j}^{n}$. Note that $x^{gt}$ is not accessible by the optimizer. This approach allows us to turn a predictive task (i.e., predicting the severity of a patient ADR in a clinical trial) into an optimization task compatible with LEON.

## C.2 Full Algorithm Pseudocode for LEON

The full pseudocode for LEON is shown in **Supplementary Algorithm 1**. Prior to the start of optimization, we also leverage the backbone LLM to query external knowledge bases for prior knowledge generation—the full pseudocode for this subroutine is shown in **Supplementary Algorithm 2**.

## C.3 Language Model Prompts

This section includes the system and user prompts used for all experiments using LEON. Baseline methods using LLMs as optimizers used the same sets of prompts as provided by the respective original authors, with modifications made only to adapt the prompts to our experimental tasks.

For each task in our evaluation suite, we first define a `task_description` variable that describes the optimization task in natural language. The specific strings used in each task are included below:

- `task_description` **for the Warfarin task:**

  ```
  The provided design scores are predictions from a model trained on
  White patients only, and therefore may not be accurate for all
  patients. Propose an optimal warfarin dose (in mg/week) for the
  patient.
  ```

- `task_description` **for the HIV task:**

  ```
  The provided design scores are predictions from a model trained on
  ```

---

**Algorithm 1** (**LEON**). **L**LM-based **E**ntropy-guided **O**ptimization with k**N**owledgeable priors

---

**Inputs:**

$\hat{f}(x, z) : \mathcal{X} \times \mathcal{Z} \to \mathbb{R}$ — pre-trained surrogate model

$c^*(x) : \mathcal{X} \to \mathbb{R}$ — initialized source critic model

$z$ — vector of fixed patient-specific covariates

LLM — a large language model that supports tool calling

`knowledge` — prior knowledge in natural language generated using **Algorithm 2** using the same input LLM

$\lambda_0 \geq 0$ — initial source critic certainty parameter value (default 0.0)

$W_0 \geq 0$ — 1-Wasserstein distance bound (default 1.0)

$\eta_\lambda \geq 0$ — source critic certainty parameter step size (default 0.1)

$\eta_{\text{critic}} > 0$ — source critic learning rate (default 0.001)

$\tau \geq 0$ — temperature (default 1.0)

$b > 1$ — batch size (default 32)

$B \geq b$ — surrogate model $\hat{f}$ evaluation budget (default 2048)

Initialize a memory bank of sampled candidates $\mathcal{D}_{\text{gen}}$

Initialize $\lambda \leftarrow \lambda_0$    // Source critic certainty parameter as in **Lemma 4.2**.

Initialize `reflection` $\leftarrow \epsilon$    // Initialize the LLM's reflection as the empty string.

**for** $t = 1 \ldots \lceil B/b \rceil$ **do**

  // 1. **Sampling.** Query the LLM for a new batch of designs.

  $x^{\text{new}} := \{x_j^{\text{new}}\}_{j=1}^b \leftarrow \text{LLM}(\mathcal{D}_{\text{gen}}, z, \texttt{knowledge}, \texttt{reflection})$

  // Evaluate the designs according to the source-critic regularized surrogate.

  $y^{\text{new}} := \{y_j^{\text{new}}\}_{j=1}^b \leftarrow \{\hat{f}(x_j^{\text{new}}) + \lambda c^*(x_j^{\text{new}})\}_{j=1}^b$

  // 2. **Cluster.** Assign the designs to their equivalence classes.

  **for** $1 \leq j \leq b$ **do**

    Append $x_j^{\text{new}}$ to its equivalence class $[x_j^{\text{new}}]_\sim$

  **end for**

  // Compute the fractional occupancies $\hat{q}_i$ of each equivalence class.

  **for** $1 \leq i \leq N$ **do**

    $y_i^* \leftarrow \max_{[x]_i}[\hat{f}(x; z) + \lambda c^*(x)]$

    $\hat{q}_i \leftarrow |[x]_i|/|x^{\text{new}}|$

  **end for**

  // 3. $\mu$ **Estimation.** Estimate $\mu$ according to (9).

  **for** $1 \leq i \leq N$ **do**

    $\delta y_i^* \leftarrow y_i^* - \frac{1}{N}\sum_{i'=1}^N y_{i'}^*$

    $\delta(\log \hat{q})_i \leftarrow \log \hat{q}_i - \frac{1}{N}\sum_{i'=1}^N \log \hat{q}_{i'}$

  **end for**

  $\hat{\mu} \leftarrow \sum_{i=1}^N [\delta y_i^* \cdot \delta(\log \hat{q})_i] / \sum_{i=1}^N [\delta y_i^* \cdot \delta y_i^*]$

  // Re-train the source critic parameters $\theta_c$ and update $\lambda$ certainty parameter.

  **while** $\delta W$ has not converged **do**

    $\delta W \leftarrow \vec{\nabla}_{\theta_c}\left[\mathbb{E}_{x' \sim \mathcal{D}_{\text{src}}}[c^*(x')] - \mathbb{E}_{x \sim \{x_i^{\text{new}}\}_{i=1}^b}[c^*(x)]\right]$

    $\theta_c \leftarrow \min(\max(\theta_c + \eta_{\text{critic}} \cdot \delta W, -0.01), 0.01)$

  **end while**

  $\lambda \leftarrow \lambda - \left(\eta_\lambda/\sqrt{t}\right) \cdot (\partial g/\partial \lambda)$     // Compute the partial gradient according to (10).

  // 4. **Design Scoring.** Score each sampled design according to $\hat{\mu}(\hat{f}(x; z) + \lambda c^*(x))$.

  $\mathcal{D}_{\text{gen}} \leftarrow \mathcal{D}_{\text{gen}} \cup \{(x_j^{\text{new}}, \hat{\mu} \cdot y_j^{\text{new}})\}_{j=1}^b$    // Save the new batch of designs to memory.

  // **Reflection.** Prompt the language model to reflect on the current optimization progress.

  // The specific prompt for reflection used in our implementation is included in **Appendix C.3**.

  `reflection` $\leftarrow \text{LLM}(\{x_j^{\text{new}}, \hat{\mu} \cdot y_j^{\text{new}}\}_{j=1}^b, \texttt{reflection\_prompt})$

**end for**

**return** top candidate from $\mathcal{D}_{\text{gen}}$ with the maximum saved $\hat{\mu} \cdot y_j$ predicted score

---

---

**Algorithm 2** Knowledge generation using external knowledge repositories

---

**Inputs:**

$\mathcal{K} = \{KR_k\}_{k=1}^{n_{KR}}$ — a set of $n_{KR}$ user-provided knowledge repositories (KR)

LLM — a large language model that supports tool calling

$T \geq 0$ — the maximum number of allowed sequential tool calls

// The specific prompt used in our implementation is included in **Appendix C.3**.

`knowledge_prompt` — a prompt in natural language to elicit knowledge generation

$C \leftarrow \{\}$     // Initialize the conversation context.

// Sequential tool calling.

**for** $t = 1 \ldots T$ **do**

   // The LLM returns which knowledge repository to call and a corresponding text query $q_t$.
   Early stopping is allowed if the LLM determines no additional tool calls are required.

   $(k_t, q_t, \text{STOP}) \leftarrow \text{LLM}(C, \texttt{knowledge\_prompt}; \mathcal{K})$

   **if** $\text{STOP} = \texttt{True}$ **then**

      **break**

   **end if**

   // Retrieve relevant knowledge $r_t$ from the requested knowledge repository $KR_{k_t}$.

   $r_t \leftarrow KR_{k_t}(q_t)$

   // Append the query and response to the conversation context.

   $C \leftarrow C \cup \{(k_t, q_t, r_t)\}$

**end for**

// Generate the final knowledge in natural language.

$\text{knowledge} \leftarrow \text{LLM}(C, \texttt{knowledge\_prompt})$

**return** knowledge

---

```
patients from older studies (before 2008) only, and therefore may
not be accurate for all patients. Propose an optimal HIV
medication regimen for the patient.
```

- `task_description` **for the Breast task:**

```
The provided design scores are predictions from a model trained on
patients under 65 years old only, and therefore may not be accurate
for this patient. Propose an optimal treatment regimen for the
patient consisting of a combination of adjuvant or neoadjuvant
medications, whether to undergo surgery, and whether to undergo
radiation therapy.
```

`task_description` **for the Lung task:**

```
The provided design scores are predictions from a model trained on
patients with a good response to therapy only,and therefore may not
be accurate for this patient. Propose an optimal treatment regimen
for the patient consisting of a combination of adjuvant or
neoadjuvant medications, whether to undergo surgery, and whether to
undergo radiation therapy.
```

`task_description` **for the ADR task:**

```
The provided design scores are predictions from a model trained on
patients treated with <Drug A> only, and therefore may not be
accurate for this patient. Predict the probability of each grade of
<Adverse Drug Reaction (ADR)> under the current medication.
```

Redacted components are indicated by `<>` angle brackets above. Separately, each pair consisting of a patient with fixed covariates $z$ and a proposed design $x$ was programmatically represented in natural language as a `patient_description`. Representative example values for this string in each task are included below; exact values have been modified to preserve patient anonymity. In addition to being used in the language model prompt during optimization (see the **User prompt for proposing new treatment designs** below), each of the `patient_description` representations of $(x, z)$ tuples were also used as input into an embedding model to define the equivalence relation as detailed in the main text. We ablate the choice of embedding model in **Supplementary Table S9**.

- **Sample** `patient_description` **in the Warfarin task:**

  ```
  Asian patient (age 70 - 79 years old) with a BMI of 25.7. CYP2C9
  Genotype Variant: *1/*1. VKORC1 SNP: A/G.

  Warfarin Dose: 32.0 mg/week
  ```

- **Sample** `patient_description` **in the HIV task:**

  ```
  Patient newly diagnosed with HIV-1 has the following HIV Protease
  Mutations: N37A; I15V; T12K. The patient also has the following
  HIV Reverse Transcriptase Mutations: E204G; I135T; G196E

  Prescribed Medications: Zidovudine (Retrovir), Lamivudine (Epivir),
  Abacavir (Ziagen)
  ```

- **Sample** `patient_description` **in the Breast task:**

  ```
  69 y.o. African American Female diagnosed with Stage I (T2N0M0)
  Breast Cancer.

  Vitals:
  BP 130/85 | SpO2 99.0 | HR 70

  Labs:
    - [CBC] Hct: 35.8% | Hgb: 11.7 | WBC: 4.8 | Plt: 259.0
    - [BMP] Na: 143 | Cl: 99 | Glu: 90 | K: 4.8
    - [LFTs] ALT: 17 | AST 17 | Albumin: 27 | ALP: 94 | Prot: 60
    - [Additional Labs and Biomarkers] Ki67: >=20%

  Treatment Plan:
  Adjuvant Capecitabine with Surgery and Radiation
  ```

- **Sample** `patient_description` **in the Lung task:**

  ```
  75 y.o. Asian Male diagnosed with Stage IA (T1CN0M0) NSCLC.

  Vitals:
  SpO2 90.0

  Labs:
    - [CBC] Hct: 48.5% | Hgb: 15.1 | WBC: 9.5 | Plt: 243
    - [BMP] Na 138 | Cl: 101 | K: 3.5 | Cr: 1.5 | BUN: 14 | Glu: 95
    - [LFTs] ALT: 11 | AST 13 | ALP: 75 | TBili: 0.5 | Prot: 65
    - [Additional Labs and Biomarkers] Ca: 9.0 | CEA: 5.8

  Treatment Plan:
  Adjuvant Pembrolizumab with Surgery
  ```

- **Sample** `patient_description` **in the ADR task:**

  ```
  68 y.o. Female (ECOG 1) diagnosed with <Disease> currently treated
  with <Drug B>.

  Dosing Schedule:
    - Day 1: <Drug B> <Dose X>
    - Day 7: <Drug B> <Dose Y>
    - Day 14: <Drug B> <Dose Z>

  Tumor:
    - <Measurement A>: <Value A>
    - <Measurement B>: <Value B>

  Vitals: O2Sat 98 | HR 73 | RR 17 | T 36.7 | BP 121/80

  Labs:
    - [CBC] Hct: 35% | Hgb: 11.8 | WBCs: 5.2
    - [BMP] Na: 140 | Cl: 100 | K: 4.2 | HCO3-: 24 | Cr: 1.5 | BUN: 6
    - [LFTs] Albumin: 3.9 | ALT: 25 | AST: 29 | TBili: 11
    - [Additional Labs and Biomarkers] Ca: 9.2 | Mg: 2.0 | Phos: 3.2
  ```

Redacted components are again indicated by <> angle brackets above. Finally, user prompts in LEON also make use of a `memory` string, which is a table of previously proposed designs (represented in natural language) and their corresponding scores according to LEON. Importantly, **note that** `memory` **never contains any scores from the ground-truth objective**, which is never made available to the optimizer. In general, `memory` is a Markdown-formatted table that includes the maximum number of most recently sampled design proposals subject to the LLM's context window.

- **Sample** `memory` **entry for the Warfarin task:**

```
|   | designs   |  scores |
|--:|:----------|-------:|
| 0 | 21.6399   | -1.4465 |
```

- **Sample** `memory` **entry for the HIV task:**

```
|   | designs                                            | scores |
|--:|:---------------------------------------------------|-------:|
| 0 | [ABC] Abacavir (Ziagen), [3TC] Lamivudine (Epivir) | 0.3491 |
```

- **Sample** `memory` **entry for the Breast task:**

```
|   | designs                                          |  scores |
|--:|:-------------------------------------------------|--------:|
| 0 | Adjuvant Chemotherapy (Anastrozole) with Surgery | -0.5438 |
```

- **Sample** `memory` **entry for the Lung task:**

```
|   | designs                                |  scores |
|--:|:---------------------------------------|--------:|
| 0 | Chemotherapy (Carboplatin, Paclitaxel) | -0.6062 |
```

- **Sample** `memory` **entry for the ADR task:**

```
|   | designs                                |  scores |
|--:|:---------------------------------------|-------:|
| 0 | (0.0000, 0.2575, 0.0000, 0.6289, 0.1136) | 1.0022 |
```

We now provide the system and user prompts used in LEON as format strings. Firstly, in step **1. Sampling** of **Supplementary Algorithm 1**, we prompt the language model optimizer to return a new batch of designs; the system and user prompts for this process are included here:

**System prompt for proposing new treatment designs**:

```
You are a clinical assistant whose role is to propose treatments for
simulated patients based on historical data and iterative feedback. You
will be provided with a history of treatments along with their respective
performance scores. Your task is to propose new treatments that
potentially yield better outcomes. After each proposal, feedback based on
evaluation results will be provided to refine future proposals.

### Optimization Strategy
  - Learn from the history of previous treaatments and their scores.
  - Balance exploration (trying diverse treatment options) and
    exploitation (refining successful treatments).
  - Do not propose treatments identical to those already evaluated.

### Input Format
In each round, you will receive:
  1. **Prior Knowledge:** Some potentially relevant context to
     consider that is specific to the particular patient.
  2. **Memory:** A table of previously evaluated treatments and their
     scores. Each row in the list will contain the following:
      - round_index: the integer index of the round
      - treatment: the proposed treatment
      - score: float score
     A higher score indicates a better patient treatment.

### Output Format
```

```
You must propose a treatment for the simulated patient that will be
scored by the evaluator. Your goal is to find treatments that are high
scoring.
```

**User prompt for proposing new treatment designs**:

```
{prior_knowledge}

{patient_description}

### Previously Proposed Designs
{memory}

### Reflection
{reflection}

### Task
{task_description}
```

where `prior_knowledge` is a string of prior knowledge returned by **Supplementary Algorithm 2**, and `reflection` is the reflection string from the previous optimization loop in **Supplementary Algorithm 1**. Separately, the prompt `knowledge_prompt` used in **Supplementary Algorithm 2** is shown below:

**System prompt for knowledge generation**:

```
You are a helpful biomedical knowledge assistant whose job is to retrieve
relevant knowledge to help a domain expert solve a specific task in
biology and medicine. Given a description of a patient and a clinical
task, you will decide which knowledge sources to retrieve from (and with
what arguments). You may call multiple functions sequentially; when you
have enough information, return a final answer of relevant prior
knowledge without any further function calls. Be specific, concise, and
comprehensive in your response.
```

**User prompt for knowledge generation**:

```
{patient_description}

Problem Description: {task_description}

Provide relevant factual information to help the expert solve the
problem.
```

Finally, **Supplementary Algorithm 1** also describes a **reflection** call to the LLM at the end of each optimization step, where the goal is to prompt the LLM to reflect on the efficacy of the current optimization progress. The input prompt `reflection_prompt` for this process is shown below:

```
### Task Description
{task_description}

### Previously Proposed Designs
{memory}

Please analyze the scores of the proposed designs and reflect on how to
better improve the score. In addition to score-based optimization,
evaluate which design strategies are working and which are not **from a
biomedical and biological perspective**.

Specifically:
- Identify any recurring features or mechanisms in high-scoring designs
  that are biologically plausible or supported by known biomedical
  principles.
- Point out any strategies in low-scoring designs that might be failing
```

```
   due to biological implausibility, off-target effects, instability,
   toxicity, or other biomedical concerns.
 - Consider how principles from molecular biology, pharmacokinetics,
   immunology, or relevant biomedical disciplines might explain the
   observed outcomes.

Do NOT propose a new design in your response! Only respond with your
thinking.
```

### C.4 EXCLUDED BASELINES

**Evolutionary algorithms.** Adapting LLMs as evolutionary optimizers has been explored in prior work, albeit primarily for singular tasks without easy generalization of the methods to new prolems, such as the biomedical tasks considered herein. We exclude LLM-driven Evolutionary Algorithms (LMEA) from Liu et al. (2024a) because their work and publicly available implementation is specifically adapted for solving the *traveling salesman problem*; generalizing their implementation while ensuring its faithfulness to the authors' original implementation is outside the scope of this work. Similarly, we exclude EvoPrompting from Chen et al. (2023a) and EvoPrompt Guo et al. (2024b), as these prior works are only implemented by the original authors for neural architecture search and language model prompt optimization, respectively. Finally, AlphaEvolve from Novikov et al. (2025) and FunSearch from Romera-Paredes et al. (2024) are also LLM-based evolutionary algorithms, but do not optimize against a dense reward function as we do in our work.

**Biomedical optimization.** Prior work on leveraging language models for biomedical optimization tasks have almost exclusively fine-tuned decoder models for biological sequence generation. For example, the methods from Reinhart & Statt (2024) and Ma et al. (2024a) make use of domain-specific molecular dynamics and material point method (MPM) simulators in their respective frameworks, and therefore cannot be generalized to our experimental setting. Language Model Optimization with Margin Expectation (LLOME) from Chen et al. (2025a) involves language model post-training for generating new protein and DNA sequences, and was therefore excluded from our evaluation. We also exclude the method from Flam-Shepherd et al. (2022) as a baseline since it is introduced as a molecule distribution matcher, which requires high-quality designs as input (unavailable to us in our setting). Finally, reinforcement learning-based methods, such as DyNA proximal policy optimization (PPO) from Angermueller et al. (2020) and REINFORCE from Williams (1992), are excluded since we do not consider reinforcement learning tasks in our work.

## D ADDITIONAL EXPERIMENTAL RESULTS

### D.1 DISTRIBUTION SHIFT ANALYSIS

To characterize the distribution shift between the surrogate model trained on the source dataset and the ground-truth objective learned on the target dataset, we first compare the performance of the surrogate and objective functions on accurately predicting the annotated labels in the target dataset in **Supplementary Figure S1**. We also plot the agreement between each of the functions and the ground-truth annotation for the target dataset in **Supplementary Figure S2**. As expected, the surrogate model consistently underperforms on the target dataset when compared to the ground-truth oracle function, reflecting the predictive performance drop in real-world clinical decision making tasks under distribution shift. Finally in **Supplementary Figure S3**, we also plot the distribution of ground-truth annotations in both the source and target datasets.

### D.2 COST ANALYSIS

**Token usage and monetary cost.** Using OpenAI's `gpt-4o-mini-2024-07-18` LLM as the backbone optimizer, we report the total number of input and output tokens used in LEON across the tasks in our evaluation suite in **Supplementary Fig. S4**. As of September 2025, the standard API pricing for the `gpt-4o-mini-2024-07-18` model is \$0.15 per million input tokens and \$0.60 per million output tokens. Using these price points and assuming no cached inputs, the cost per patient for using LEON is \$0.21 $\pm$ 0.00 on the Warfarin task; \$0.56 $\pm$ 0.00 on the HIV task; \$0.60 $\pm$ 0.01 on the Breast task; \$0.51 $\pm$ 0.00 on the Lung task; and \$0.44 $\pm$ 0.00 on the ADR task.

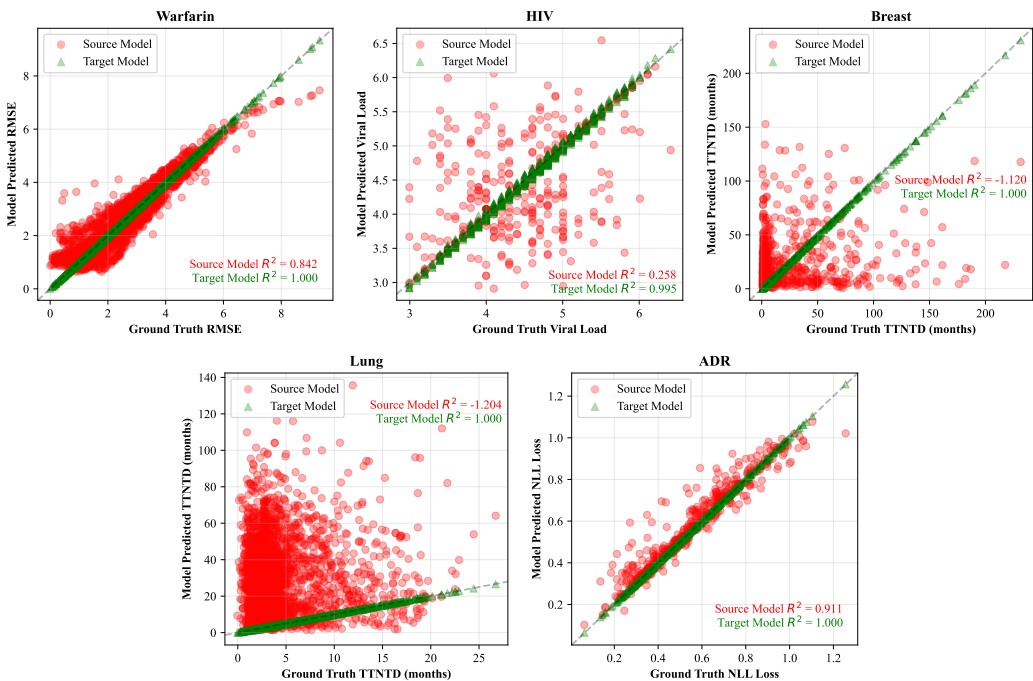

Figure S1: **Correlation plots of predictive models used for design evaluation.** In our work, we investigate the utility of LLMs in solving 5 challenging biomedical optimization problems under distribution shift. An optimizer is only given access to a model trained on data sampled from a source distribution (red) to score candidate designs, but the final proposals are scored using the ground-truth objective function (green) hidden during evaluation. The oracle is either an exact function if the true objective is known, or a machine learning model otherwise. We evaluate the agreement between the surrogate and oracle functions with the ground truth annotation in the target dataset.

**Carbon emissions.** Consistent with Samsi et al. (2023) and a publicly available carbon calculator, we assume an energy efficiency of $7.594 \times 10^{-6}$ kilowatt-hours (kWh) per thousand output tokens per billion LLM active parameters (assuming token generation is performed on NVIDIA A100 GPUs). We primarily use the `gpt-4o-mini-2024-07-18` model from OpenAI; while the size of this model is currently unknown, we conservatively estimate it to be approximately 20B active parameters for the purposes of cost estimation. Assuming a carbon intensity of 0.39 kg $CO_2$e/kWh and power usage effectiveness (PUE) of 1.17 based on U.S. data (Ren et al., 2024), a back-of-the-envelope calculation tells us that the cost per patient for using LEON is approximately $1.66 \times 10^{-3}$ kg $CO_2$e on the Warfarin task; $1.03 \times 10^{-2}$ kg $CO_2$e on the HIV task; $1.95 \times 10^{-2}$ kg $CO_2$e on the Breast task; $1.14 \times 10^{-2}$ kg $CO_2$e on the Lung task; and $6.38 \times 10^{-3}$ kg $CO_2$e on the ADR task.

**GPU utilization.** We observed that GPU utilization was dominated by the size of locally deployed LLMs (e.g., the MedGemma 27B model used as an external knowledge repository in **Supplementary Algorithm 2**), rather than by our proposed method. As shown in **Supplementary Algorithm 1**, the GPU memory requirements for LEON scale linearly with the dimensionality of the patient-design space and the number of parameters of the surrogate model $\hat{f}$—both of which are orders of magnitude smaller than the resource footprint of the external LLM. In practice, we found that the maximum GPU memory used by LEON did not exceed **3 GB** for any given task.

### D.3 QUANTITATIVE ANALYSIS OF PRIOR KNOWLEDGE

Recall from **Supplementary Algorithm 2** that the availability of external knowledge sources is a key component of LEON. While a base generalist language model may not have the prior knowledge to achieve high certainty in its proposed designs according to (4), the LLM can query repositories of domain-specific expert knowledge to ultimately synthesize knowledge in natural language that can be used for the optimization task. To better interrogate this process, we plot the frequency with

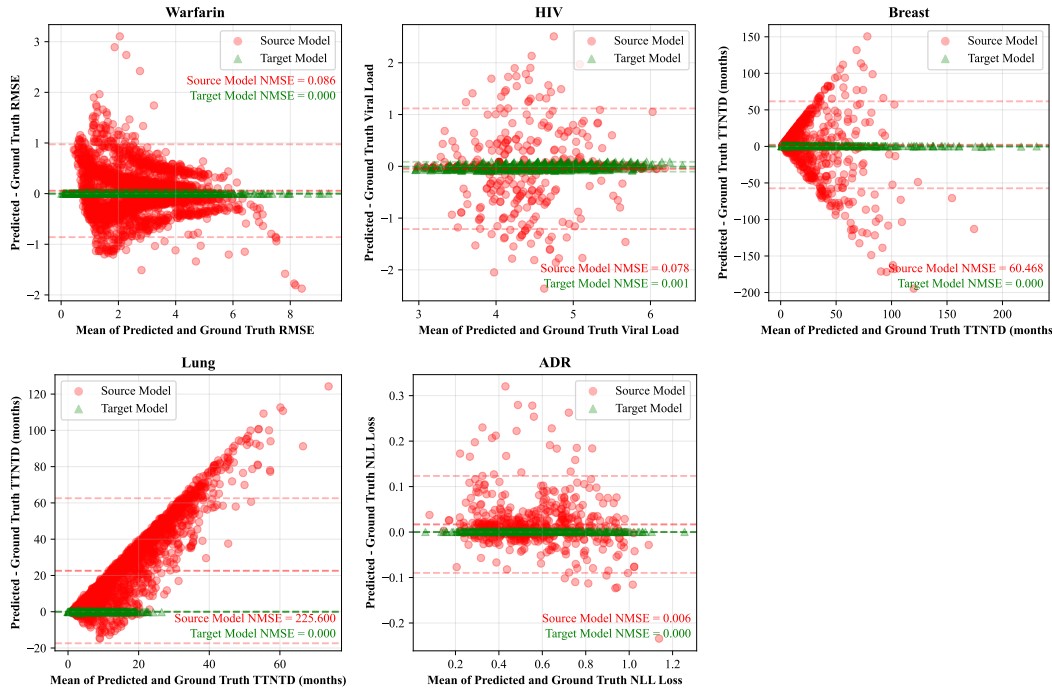

Figure S2: **Bland-Altman plots of predictive models used for design evaluation.** In our work, we investigate the utility of LLMs in solving 6 challenging biomedical optimization problems under distribution shift. More explicitly, an LLM is only given access to a surrogate model trained on data from a source distribution (red) to score candidate designs, but the final proposals are scored using the ground-truth objective (green) learned on data from the target distribution and hidden during optimization. We evaluate the agreement between each of the source- and target- trained models with the ground truth annotation for each datum in the target dataset. Mean and 95% confidence intervals for both models in each task are indicated by the horizontal dotted lines.

which each of the expert knowledge repositories are individually queried as tools across 100 patients for each of the 5 benchmarking tasks (**Supplementary Fig. S5**).

Our results demonstrate that the same base language model (`gpt-4o-mini-2024-07-18`) differentially queries the set of external knowledge bases as a function of the underlying task. The total number of knowledge base queries per patient differs by task (mean $\pm$ standard error of mean (SEM); Warfarin: $2.91 \pm 0.10$; HIV: $3.01 \pm 0.05$; Breast: $3.16 \pm 0.09$; Lung: $3.50 \pm 0.05$; ADR: $3.63 \pm 0.05$ queries per patient). The increasing number of queries correlates with the authors' subjective evaluation of the difficulty of each task: that is, Warfarin (resp., ADR) is the least (resp., most) challenging optimization problem in our evaluation suite. These results suggest that the language model preferentially relies on more prior knowledge sources as the difficulty of the optimization task increases.

Separately, we observe that the specific knowledge sources queried by the model vary by task: for example, the COSMIC knowledge base from Tate et al. (2019) is preferentially queried in the Breast and Lung tasks that deal with oncologic treatment optimization. We also observe that the knowledge bases GDSC (Yang et al., 2013) and DepMap (Tsherniak et al., 2017) are rarely queried by the language model across all 5 tasks. This makes sense, as both knowledge sources detail properties about *cell lines* (as opposed to patients), and are therefore unlikely to be relevant for any of our tasks. Finally, we ran a $\chi^2$-test of homogeneity at the per-task level to evaluate whether different knowledge bases had different rates of being queried within each optimization task. As made visually apparent from **Supplementary Figure S5**, the observed differences in the frequency of knowledge bases being queried per task are far too large to be explained by random sampling variation alone (Warfarin: $\chi^2 \approx 535.3$, dof $= 7$, $p \approx 2.09 \times 10^{-111}$; HIV: $\chi^2 \approx 763.0$, dof $= 7$, $p \approx 1.81 \times 10^{-160}$; Breast: $\chi^2 \approx 452.3$, dof $= 7$, $p \approx 1.39 \times 10^{-93}$; Lung: $\chi^2 \approx 556.8$, dof $= 7$, $p \approx 4.96 \times 10^{-116}$; ADR: $\chi^2 \approx 674.3$, dof $= 7$, $p \approx 2.36 \times 10^{-141}$). Overall, these results suggest

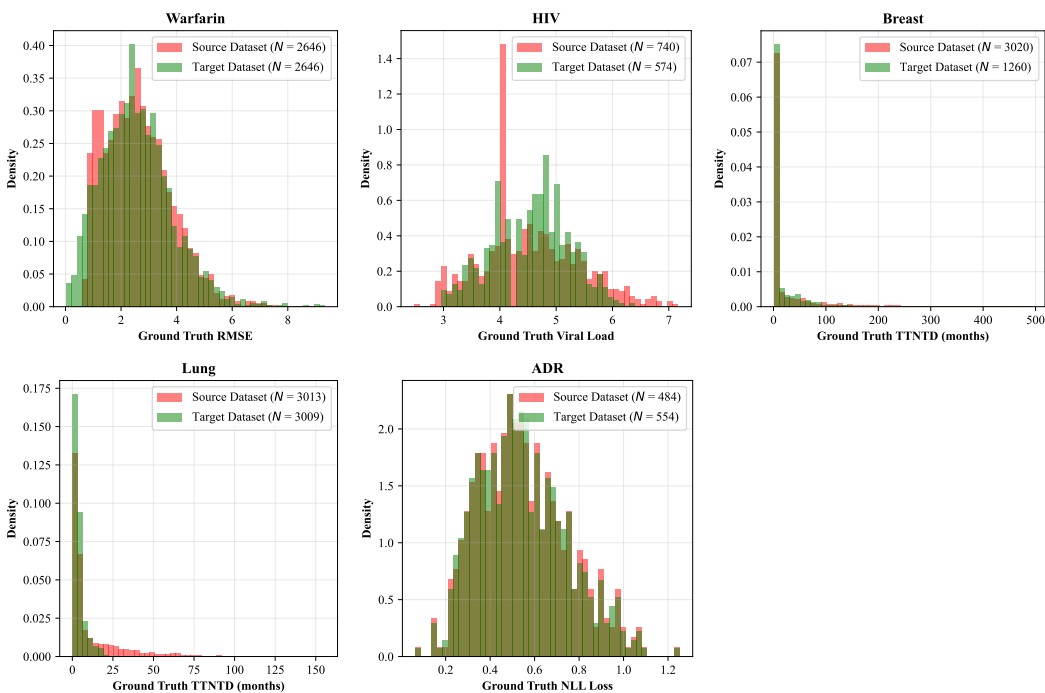

Figure S3: **Distributions of ground truth scores within source and target datasets.** We plot the distribution of ground-truth objective values for both the source (red) and target (green) datasets.

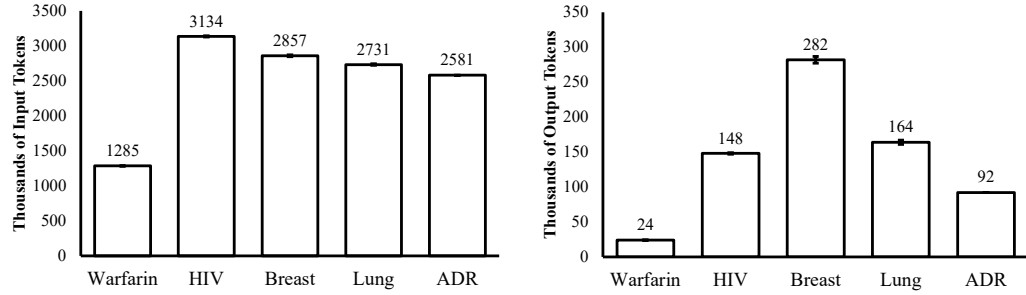

Figure S4: **Token usage of LEON across different tasks.** We plot the average number of input and output tokens used per patient experiment with the `gpt-4o-mini-2024-07-18` backbone optimizer. Error bars represent the standard error of the mean across 100 patient experiments.

that the backbone LLM is selecting knowledge sources to retrieve relevant expert prior knowledge for a given optimization task in a manner that cannot be attributed to random chance alone.

### D.4   QUALITATIVE EXAMPLES OF PRIOR KNOWLEDGE

Recall from **Supplementary Algorithm 1** that a key component of LEON is synthesizing **domain-specific prior knowledge**: given access to relevant biomedical knowledge bases, we task the baseline language model with making the appropriate tool calls to gather domain-specific and context-specific prior knowledge conditioned on the task description and particular covariates $z$. The language model is then asked to respond with relevant prior knowledge to use as context for downstream optimization—see **Appendix C.3** for additional details. Here, we provide representative examples of the prior knowledge outputs (indicated by monospace font below) generated by `gpt-4o-mini-2024-07-18` for the **Warfarin** task. We focus on this particular task because it is a domain area where we are able to accurately verify LLM outputs. Furthermore, a linear oracle model from Consortium (2009) is available for this task, facilitating the domain-specific verifica-

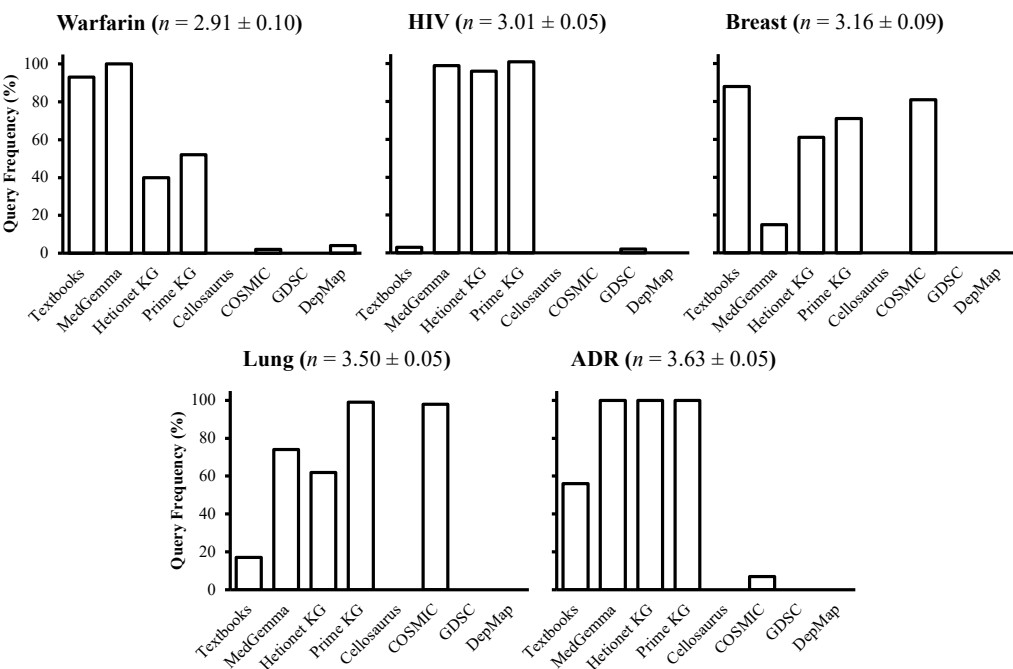

Figure S5: **Distribution of knowledge sources called as tools by the LLM for prior knowledge synthesis.** We plot the frequency of querying each repository (over 100 patients) in **Supplementary Algorithm 2** across each of the optimization tasks evaluated. Each subplot title includes the mean $\pm$ standard error of the mean (SEM) number of tool calls made per patient.

tion of the consistency of LLMs with published prior work. To evaluate the quality of the LLM-synthetized prior knowledge, 1 U.S. resident physician and 1 U.S. senior medical student manually annotated the LLM-generated knowledge for factual accuracy; in cases where annotations semantically disagreed, the prior knowledge output was discussed until consensus was reached. We include their annotations as right-justified, colored serif font in the listings below.

In this first example, all generated prior knowledge was considered relevant and factually consistent with evidence-based medical guidelines. The LLM optimizer queried the following knowledge sources as tool calls (in order): (1) MedGemma 27B; (2) Hetionet Knowledge Graph; (3) Prime Knowledge Graph; and (4) Medical Textbooks (see **Appendix C** for additional details). Furthermore, the final proposed warfarin dosing strategy score proposed by the LLM optimizer using the prior knowledge and LEON achieved an oracle score of 1.24, which is more optimal than the mean score achieved by our method on this task in **Table 1**.

```
For a 30-39 year old Asian patient with a BMI      The demographic information here
of 29.1, a CYP2C9 genotype of *1/*1, and a         is accurate for this particular patient.
VKORC1 SNP of A/A, the following information
is relevant for determining an optimal warfarin
dose:
1. **Genetic Influence on Warfarin Dosing**:
   - The CYP2C9 *1/*1 genotype indicates           This is consistent with expert guidance
     normal metabolism of warfarin, meaning         (Sconce et al., 2005), and also with
     the patient is not a poor metabolizer.         the linear oracle dosing algorithm from
     However, this genotype does not require        Consortium (2009) used in this work.
     a significantly higher dose compared to
     those with reduced function alleles.
   - The VKORC1 A/A genotype suggests                  This is also consistent with expert
     increased sensitivity to warfarin,              guidance (Dean, 2012), and also with
     typically requiring lower doses. Studies          the linear oracle dosing algorithm
     indicate that individuals with the A/A                  from Consortium (2009).
     lower doses than those with non-A haplotypes.
2. **Initial Dosing Recommendations**:
```

```
       - Standard initial dosing for warfarin is        This is consistent with medical
         typically between 5-10 mg per day.              guidelines (Dean, 2012). A starting
         However, due to the VKORC1 A/A genotype,        dose of 3-4 mg/day is usually
         a lower starting dose is advisable.             recommended for this patient in
       - A common approach is to start with             the absence of other medical history.
         a dose of around 2.5-5 mg per day
         for patients with the A/A genotype,
         especially considering the patient's
         Asian ethnicity, which is associated with
         a higher prevalence of VKORC1 variants.
```

In this second example on the **Warfarin** task, there was a factually incorrect statement made by the LLM, highlighted in red below. The LLM optimizer queried the following knowledge sources in order: (1) Medical Textbooks; and (2) MedGemma 27B. Quantitatively, we found that the LLM optimizer consequently performed worse on this patient, achieving an inferior oracle score of 2.32.

```
To propose an optimal warfarin dose for
the patient with the following characteristics:
- **Age**: 60-69 years                              The demographic information here
- **BMI**: 23.4                                     is accurate for this particular patient.
- **CYP2C9 Genotype**: *1/*1 (wild-type)
- **VKORC1 SNP**: A/G

### Relevant Information:
1. **CYP2C9 Genotype**:
   - The *1/*1 genotype indicates normal            This is consistent with expert guidance
     enzyme activity, meaning the patient is         (Sconce et al., 2005), and also with
     a "normal metabolizer" of warfarin.            the linear oracle dosing algorithm from
     Patients with this genotype typically           Consortium (2009) used in this work.
     require higher doses of warfarin
     compared to those with variant
     alleles (*2 or *3).
2. **VKORC1 Genotype**:
   - The A/G genotype suggests that the             While heterozygous A/G patients for
     patient has a moderate response to                   the VKORC1: −1639G→A
     warfarin. The presence of the G allele      polymorphism have moderate expression
     is associated with lower VKORC1             of VKORC1, this genotype generally leads
     expression, which generally leads to a          to *higher* warfarin doses than that for
     reduced requirement for warfarin                 A/A individuals (Dean, 2012). This
     compared to those with the A/A genotype.   knowledge output is also inconsistent
     Specifically, individuals with the       with the ground-truth dosing algorithm from
     A/G genotype may require about 25%             Consortium (2009) used in this work.
     lower doses than A/A homozygotes.
3. **Dosing Guidelines**:
   - Initial warfarin dosing typically             This dosage guidance is consistent with
     starts at 5-10 mg/day. Given the             clinical recommendations (Dean, 2012).
     patient's age and the influence of
     VKORC1, a conservative approach is
     advisable.
   - For a patient with CYP2C9 *1/*1 and
     VKORC1 A/G, a reasonable starting
     dose might be around 5 mg/day, with
     adjustments based on INR monitoring.
```

## D.5 ADDITIONAL ANALYSIS OF THE LLM CERTAINTY PARAMETER $\mu$

Recall that the certainty parameter $\mu$ in LEON is the Lagrange multiplier of the entropy-based constraint in our constrained problem formulation in (4). Larger values of $\mu$ are observed when the LLM consistently returns equivalent treatment designs for a given input prompt despite the nondeterminism of the generative model (as dictated by the LLM temperature hyperparmaeter $\tau = 1.0$ used in the main text). Using this framework, we can plot how the value of $\mu$ changes over the course of optimization to better interrogate how the certainty of the language model in proposing high-quality treatment designs changes during optimization.

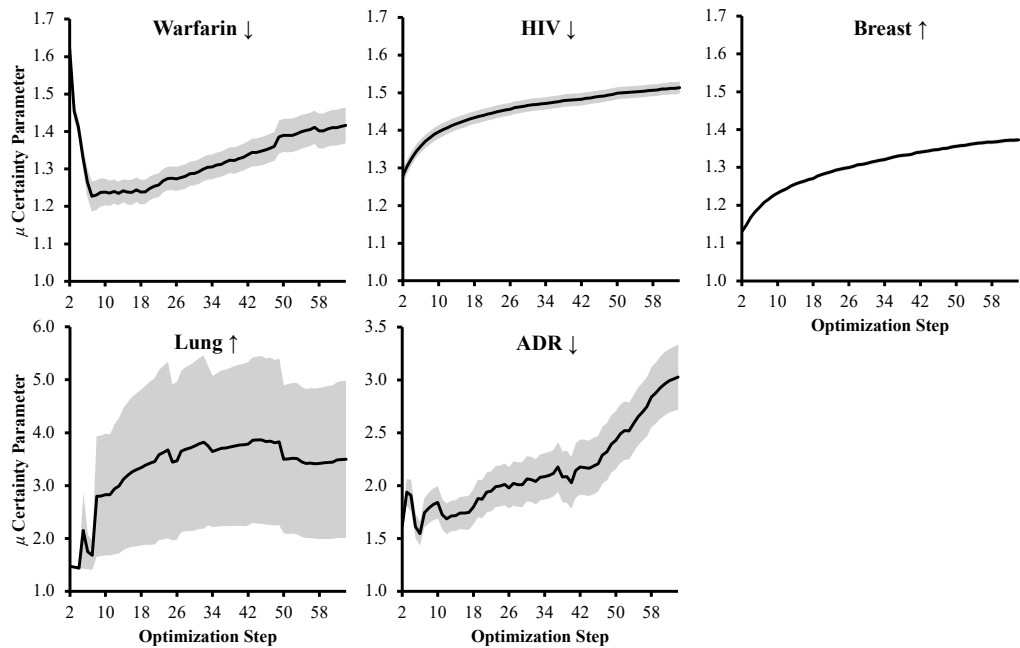

Figure S6: **LLM certainty parameter $\mu$ over the course of optimization.** We plot the mean $\pm$ SEM value of $\mu$ estimated using (9) at each optimization step for $n = 100$ independent target patient experiments. Up (resp. down) arrows indicate that the task is a maximization (resp., minimization) task. Higher values of $\mu$ suggest a greater confidence of the LLM in its design proposals.

Our results are shown in **Supplementary Figure S6**. We find that the value of $\mu$ generally increases as a function of the optimization step. This suggests that the certainty of the LLM increases as more designs are observed and evaluated using our method. One possible explanation for this trend is that LLM certainty is a function of not only the prior knowledge generated, but also the previously proposed designs. That is, the LLM is more confident in future designs because it is able to 'learn' from the history of previously sampled designs and their scores according to LEON.

### D.6 ADDITIONAL ANALYSIS OF THE SOURCE CRITIC WEIGHTING PARAMETER $\lambda$

Similar to our analysis in **Appendix D.5**, we can analyze how that $\lambda$ certainty parameter varies over the course of optimization, where $\lambda$ is the Lagrange multiplier of the source critic-based constraint in our constrained problem formulation in (4). Qualitatively, larger values of $\lambda$ are associated with upweighting the importance of the source critic relative to the surrogate predictive model.

Our results are shown in **Supplementary Figure S7**. We find that the value of $\lambda$ consistently increases as a function of the optimization step. This suggests that the importance of the LLM increases as more designs are observed and evaluated in LEON. Future work may investigate how the performance of LEON changes as a function of the initial value of $\lambda$ in **Supplementary Algorithm 1** and both the step size and number of gradient update steps in (11).

### D.7 EXTENDING LEON TO TRADITIONAL OPTIMIZERS

In principle, there is nothing in **Supplementary Algorithm 1** that is particular to language model-based optimizers: the auxiliary source critic model can be used in conjunction with any optimization method, and the entropy-based constraint is enforceable with any method that supports batched acquisitions. To this end, we implement LEON with gradient ascent (**Grad.**) and Bayesian optimization (**BO-qEI**) and evaluate LEON with these traditional optimizers in **Supplementary Table S2**.

Our results suggest that using LEON with these traditional, non-LLM based optimizers does not offer a meaningful advantage over the corresponding baselines. LEON-BO-qEI improves upon BO-qEI on 3 of the 5 tasks, and LEON-Grad. improves upon Grad. only on 1 of the 5 tasks. This trend is

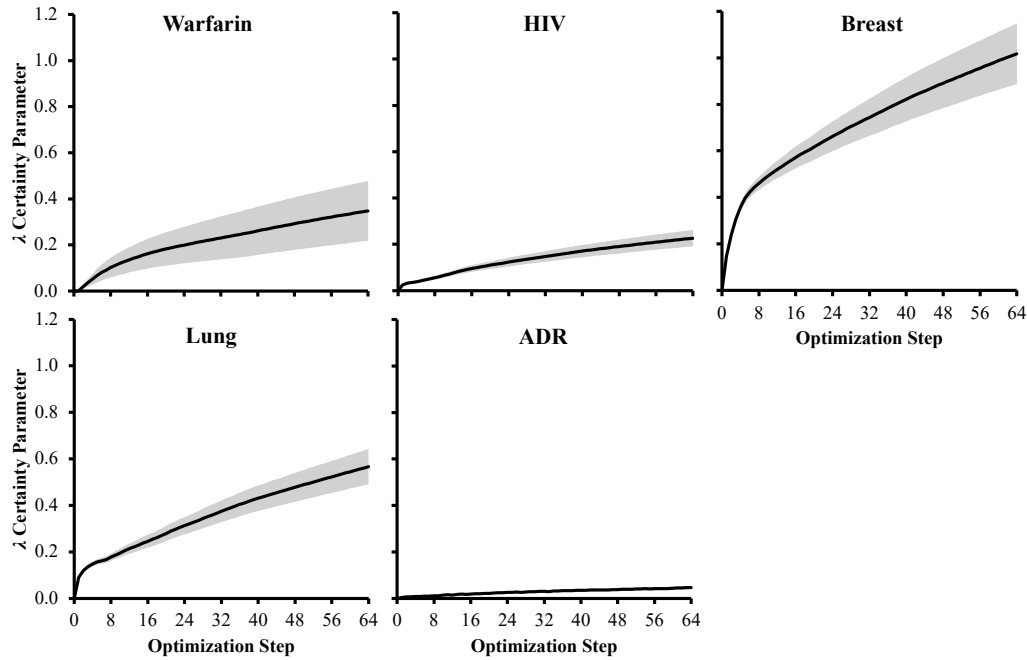

Figure S7: **Source critic weighting parameter $\lambda$ over the course of optimization.** We plot the mean $\pm$ SEM value of $\lambda$ computed according to (11) at each optimization step for $n = 100$ independent target patient experiments. Up (resp. down) arrows indicate that the task is a maximization (resp., minimization) task. Higher values of $\lambda$ suggest a greater weight placed on feedback from the source critic relative to the surrogate model.

Table S2: **Evaluating LEON for traditional optimizers.** We evaluate the utility of LEON in improving the performance on non-LLM based optimization methods gradient ascent (Grad.) and Bayesian optimization (BO-qEI). We report mean $\pm$ standard error of mean (SEM) oracle objective value achieved by the single proposed design for a given patient, averaged over $n = 100$ test patients. **Bolded** cells indicate when using LEON improves upon the backbone optimizer for a given task.

| Method | Warfarin $\downarrow$ RMSE Loss (*mg/week*) | HIV $\downarrow$ Viral Load (*copies/mL*) | Breast $\uparrow$ TTNTD (*months*) | NSCLC $\uparrow$ TTNTD (*months*) | CRS $\downarrow$ NLL Loss (*no units*) |
|---|---|---|---|---|---|
| Grad. | $\mathbf{1.37 \pm 0.13}$ | $\mathbf{4.52 \pm 0.04}$ | $\mathbf{73.07 \pm 2.30}$ | $24.67 \pm 0.34$ | $\mathbf{23.7 \pm 1.7}$ |
| LEON-Grad. | $2.24 \pm 0.19$ | $4.54 \pm 0.06$ | $51.68 \pm 1.38$ | $\mathbf{27.61 \pm 0.64}$ | $23.8 \pm 1.7$ |
| BO-qEI | $\mathbf{1.36 \pm 0.13}$ | $4.53 \pm 0.04$ | $\mathbf{67.11 \pm 1.86}$ | $28.65 \pm 0.65$ | $23.5 \pm 1.7$ |
| LEON-BO-qEI | $2.02 \pm 0.15$ | $\mathbf{4.48 \pm 0.05}$ | $65.52 \pm 2.34$ | $\mathbf{31.04 \pm 0.82}$ | $\mathbf{21.4 \pm 1.8}$ |

consistent with the notion that **prior knowledge is *critical* for LEON** to consistently improve upon optimizer performance. Because traditional optimizers like Grad. and BO-qEI lack informative priors over the design space, the entropy-based constraint central to LEON is less meaningful using these methods. Consequently, although LEON is optimizer-agnostic in theory, its utility is limited when used with methods that do not incorporate prior knowledge.

## D.8 WHY SHOULD WE BOUND THE 1-WASSERSTEIN DISTANCE?

A key underlying assumption behind our modified problem formulation in (4) is that the source critic-based constraint can meaningfully prevent over-extrapolation against the surrogate model that is frequency observed in optimization tasks under distribution shift (Yao et al., 2024; Trabucco et al., 2022). In **Theorem D.1**, we provide a theoretical motivation to help support this assumption.

**Theorem D.1** (Bound on Empirical Test Risk). *Define a real-valued, Borel-measurable function $f : \mathcal{X} \to \mathbb{R}$ defined over a domain $\mathcal{X} \subseteq \mathbb{R}^d$, and define $K := ||f(x)||_L$ to be the corresponding Lipschitz constant of $f$. Given a finite dataset of $n$ observations $\mathcal{D} := \{(x_i, f(x_i))\}_{i=1}^n$, suppose we train a predictive model $\hat{f}$ on $\mathcal{D}$ with Lipschitz constant $K_{\hat{f}}$ finite such that the empirical training risk is $\varepsilon := \mathbb{E}_{(x,y)\sim\mathcal{D}}|y - \hat{f}(x)|$ finite. Then, the test risk on a new sample of $T$ test inputs $\mathcal{T} = \{x_j\}_{j=1}^T$ is bounded from above by*

$$\mathbb{E}_{x\sim\mathcal{T}}|f(x) - \hat{f}(x)| \leq \varepsilon + (K + K_{\hat{f}})W_1(\mu_\mathcal{D}, \mu_\mathcal{T})$$

*where $W_1(\mu_\mathcal{D}, \mu_\mathcal{T})$ is the 1-Wasserstein distance associated with $||\cdot||_2$.*

*Proof.* Define $\gamma \in \Gamma(\mu_\mathcal{D}, \mu_\mathcal{T})$ as the optimal coupling between input observations $x'$ and $x$ in $\mathcal{D}$ and $\mathcal{T}$, respectively. For pairs $(x', x) \sim \gamma$, note that because $\mu_\mathcal{T}$ is the $x$-marginal of $\gamma$,

$$\mathbb{E}_{x\sim\mathcal{T}}|f(x) - \hat{f}(x)| = \mathbb{E}_{(x',x)\sim\gamma}|f(x) - \hat{f}(x)|$$

$$= \mathbb{E}_{(x',x)\sim\gamma}\left|\left(f(x) - \hat{f}(x)\right) - \left(f(x') - \hat{f}(x')\right) + \left(f(x') - \hat{f}(x')\right)\right|$$

$$\leq \mathbb{E}_{(x',x)\sim\gamma}\left|\left(f(x) - \hat{f}(x)\right) - \left(f(x') - \hat{f}(x')\right)\right| + \mathbb{E}_{(x',x)\sim\gamma}\left|f(x') - \hat{f}(x')\right|$$

using the triangle inequality. Because $\mu_\mathcal{D}$ is the $x'$-marginal of $\gamma$, we rewrite the right hand side as

$$\mathbb{E}_{x\sim\mathcal{T}}|f(x) - \hat{f}(x)| \leq \mathbb{E}_{(x',x)\sim\gamma}\left|\left(f(x) - \hat{f}(x)\right) - \left(f(x') - \hat{f}(x')\right)\right| + \mathbb{E}_{x'\sim\mathcal{D}}\left|f(x') - \hat{f}(x')\right|$$

$$= \mathbb{E}_{(x',x)\sim\gamma}\left|\left(f(x) - \hat{f}(x)\right) - \left(f(x') - \hat{f}(x')\right)\right| + \varepsilon$$

using the definition of the empirical training risk $\varepsilon$. Applying the definition of the Lipschitz constants $K, K_{\hat{f}}$ as well as the definition of the 1-Wasserstein distance gives

$$\mathbb{E}_{x\sim\mathcal{T}}|f(x) - \hat{f}(x)| \leq (K + K_{\hat{f}})\mathbb{E}_{(x',x)\sim\gamma}|x - x'| + \varepsilon = (K + K_{\hat{f}})W_0 + \varepsilon$$

The claim follows. $\qquad\square$

We remark that deriving the global Lipschitz bounds $K, K_{\hat{f}}$ is $\mathcal{NP}$-hard and infeasible in practice (Scaman & Virmaux, 2018; Hu et al., 2024). However, the above result still holds if $K, K_{\hat{f}}$ only hold locally over a finite subset of $\mathcal{X}$ that contains $\mathcal{D}\bigcup\mathcal{T}$, which is much easier to derive. Furthermore, note that the constants $\varepsilon$ and Lipschitz constants $K, K_{\hat{f}}$ in **Theorem D.1** are irreducible, since we assume that we do not have control over $\mathcal{D}$ or the functions $f, \hat{f}$. However, the above result also shows that bounding $W_1(\mu_\mathcal{D}, \mu_\mathcal{T})$ will yield a corresponding finite bound on the empirical test risk. **This is a key observation**—by intelligently choosing the test points in $\mathcal{T}$ that we use to evaluate with $\hat{f}$, we can guarantee a bound on the mean test error over $\mathcal{T}$.

### D.9 SAMPLE REFLECTION TRACE

To illustrate the process of reflection, we include a representative example reflection output for an optimization experiment on the **Warfarin** task. Based on the prompting strategy used for reflection in **Appendix C.3**, we encourage the language model to reflect on the optimization progress from both task-agnostic 'exploration-exploitation' and biomedical reasoning perspectives.

```
Analyzing the previously proposed warfarin doses and their scores reveals
several insights into the potential biological mechanisms and
pharmacokinetic principles that may be influencing the outcomes.

### High-Scoring Designs
1. **Doses with Lower Values**: The highest scores are associated with
   lower doses, such as 11.3417 mg/week (score: -0.398857) and 15.1328
   mg/week (score: -0.330352). This suggests that lower doses may be more
   effective for this patient, possibly due to:
   - **Pharmacokinetics**: Lower doses may lead to more stable INR
     (International Normalized Ratio) levels, reducing the risk of
```

```
        bleeding or thromboembolic events. Warfarin has a narrow therapeutic
        index, and small changes in dose can lead to significant changes in
        anticoagulation effect.
      - **Patient Variability**: Individual responses to warfarin can vary
        significantly due to genetic factors (e.g., VKORC1 and CYP2C9
        polymorphisms), dietary influences, and concurrent medications.
        Lower doses may be more universally tolerated across diverse patient
        backgrounds.
2. **Avoidance of High Doses**: The lowest scores are associated with
   higher doses (e.g., 72.6990 mg/week with a score of -8.7245). High
   doses may lead to:
      - **Toxicity**: Increased risk of adverse effects, including major
        bleeding, which is a significant concern in anticoagulation therapy.
      - **Instability**: Higher doses can lead to fluctuations in INR,
        making it difficult to maintain therapeutic levels.

### Low-Scoring Designs
1. **High Doses**: As mentioned, doses like 63.5996 mg/week and 72.6990
   mg/week are associated with very low scores. These doses may be
   biologically implausible for many patients, leading to:
      - **Off-Target Effects**: Excessive anticoagulation can lead to
        complications that outweigh the benefits of treatment.
      - **Increased Variability**: Higher doses may exacerbate the
        variability in response due to genetic and environmental factors.
2. **Intermediate Doses**: Doses in the mid-range (e.g., 40.5355 mg/week,
   38.7895 mg/week) also received low scores. This could indicate that:
      - **Suboptimal Targeting**: These doses may not align well with the
        patient's specific pharmacogenomic profile, leading to inadequate
        anticoagulation or increased side effects.

### Biological and Biomedical Considerations
- **Genetic Factors**: The variability in warfarin metabolism due to
  genetic polymorphisms in CYP2C9 and VKORC1 is well-documented.
  Understanding the patient's genetic makeup could help tailor the
  dose more effectively.
- **Dietary Interactions**: Vitamin K intake can significantly affect
  warfarin efficacy. Patients with high dietary vitamin K may require
  higher doses, while those with low intake may need lower doses.
- **Drug Interactions**: Concurrent medications can alter warfarin
  metabolism, necessitating dose adjustments. Awareness of these
  interactions is crucial for optimizing therapy.

### Conclusion
The analysis suggests that lower doses are generally more effective
and safer for this patient population, likely due to the pharmacokinetic
properties of warfarin and the variability in patient responses. Future
proposals should focus on exploring lower doses while considering
individual patient factors such as genetics, diet, and potential drug
interactions to optimize outcomes.
```

## D.10 Subgroup Fairness Analysis

An important step of validating a proposed algorithm for clinical applications is *subgroup fairness analysis*, where the goal is to evaluate the algorithm for possible biases across specific subgroups to ensure equitable outcomes for patients in healthcare. Ensuring fairness of an algorithm across arbitrary subgroups is generally $\mathcal{NP}$-hard (Mandal et al., 2020) and still an active area of research outside the scope of this work (Kearns et al., 2018; 2019). Nonetheless, we sought to conduct preliminary fairness analyses of LEON with respect to two patient attributes: patient gender (Male and Female; self-identified) and race (White and Non-White; self-identified).

In **Supplementary Table S3**, we stratify the results of LEON based on these two patient attributes. We were unable to perform our fairness analysis on the **Warfarin** task because the source and target datasets were already stratified by race (meaning no White patients were evaluated using

Table S3: **LEON subgroup fairness analysis.** We stratify the performance of LEON by each of patient gender and race (separately), and report the mean $\pm$ standard error of mean (SEM) oracle objective value achieved by the single treatment design proposed by LEON for a given patient in the **Breast**, **Lung**, and **ADR** tasks. Note that a race subgroup analysis could not be performed on the **ADR** task because patient race was not available in the task dataset; see **Appendix C.1** for details.

| Attribute | Subgroup | Breast $\uparrow$ TTNTD (*months*) | Lung $\uparrow$ TTNTD (*months*) | ADR $\downarrow$ NLL Loss (*no units*) |
|---|---|---|---|---|
| **Gender** | Male | $89.27 \pm 8.72$ | $32.78 \pm 0.50$ | $12.9 \pm 2.0$ |
| | Female | $71.54 \pm 2.97$ | $32.66 \pm 0.42$ | $11.3 \pm 2.7$ |
| **Race** | White | $73.88 \pm 3.19$ | $32.51 \pm 0.43$ | — |
| | Non-White | $69.61 \pm 5.69$ | $33.10 \pm 0.45$ | — |

LEON), and the task does not include patient gender as an attribute. Similarly, we did not evaluate LEON fairness on the **HIV** task because the task does not include either patient gender or race as an attribute; see **Appendix C.1** and **Supplementary Table S1** for additional details. To evaluate if there was a statistically significant difference in the performance of LEON on a particular task with respect to a particular patient attribute, we performed a two-sample, unpaired, two-tailed $t$-test of the difference between the mean oracle objective value within each subgroup. We defined a statistically significant difference to be a $p$-value of less than $0.05$.

There was no statistically significant difference in the performance of LEON with respect to gender ($N_{\text{Male}} = 5$, $N_{\text{Female}} = 95$; $t = 1.924$; $p = 0.057$) or race ($N_{\text{White}} = 66$, $N_{\text{Non-White}} = 34$; $t = 0.65$; $p = 0.514$) on the **Breast** task. Similarly on the **Lung** task, there was no statistically significant difference in the performance of LEON with respect to gender ($N_{\text{Male}} = 42, N_{\text{Female}} = 58$; $t = 0.199$; $p = 0.843$) or race ($N_{\text{White}} = 62, N_{\text{Non-White}} = 38$; $t = -0.951$; $p = 0.344$). Finally on the **ADR** task, there was no statistically significant difference in the performance of LEON with respect to gender ($N_{\text{Male}} = 69, N_{\text{Female}} = 31$; $t = 0.478$; $p = 0.634$). Note that we could not stratify the performance of LEON on the **ADR** task based on race because the variable was not available in the task dataset. In summary, these results suggest that LEON is unlikely to be significantly biased against any particular subgroup evaluated according to gender and race patient attributes. However, future work is warranted to conduct a more comprehensive fairness audit of LEON and other clinical algorithms before real-world use is achievable.

### D.11 INTEGRATING KNOWLEDGE WITH BASELINE LLM OPTIMIZATION METHODS

A key step of LEON is in using the backbone language model to synthesize relevant domain knowledge from different sources (**Supplementary Algorithm 2**). By both using task- and patient- specific knowledge and solving the modified constrained optimization problem in (4), we show in **Table 1** that LEON outperforms other LLM-based optimization methods (i.e., **L**arge **LA**nguage **M**odels to enhance **B**ayesian **O**ptimization (LLAMBO) from Liu et al. (2024c), **O**ptimization by **PRO**mpting (OPRO) from Yang et al. (2024a), and **E**volution-driven **u**niveral **re**ward **k**it for **a**gent (Eureka) from Ma et al. (2024b)). However, it is not immediately clear if the performance gains observed from LEON are primarily due to the use of knowledge or solving (4). Put simply, *can using LLAMBO, OPRO, and/or Eureka with domain knowledge similarly boost their performance*?

The aforementioned baseline methods do not explicitly detail a way to integrate external knowledge, and so the results in **Table 1** do not provide access to external knowledge sources for the three baselines in order to remain faithful to the authors' original implementations. However, it is still possible to define a reasonable convention on how to incorporate knowledge with each method. To this end, for each of LLAMBO, OPRO, and Eureka, we first run **Supplementary Algorithm 2** using the backbone LLM (exactly as done in LEON) and append the generated knowledge to the string task context used in the respective user prompts in each of these three methods. We can then evaluate each of these methods both with and without the use of knowledge.

Our experimental results are shown in **Supplementary Table S4**. In terms of the average rank achieved across all 5 tasks, the use of LLM-synthesized knowledge generally improved the perfor-

Table S4: **Evaluating baseline LLM optimization methods with knowledge.** We evaluate the performance of LLM-based optimization methods both with and without the use of knowledge generated using **Supplementary Algorithm 2**. We report the mean $\pm$ standard error of mean (SEM) oracle objective value achieved by the single proposed design for a given patient, averaged over $n = 100$ target patients. **Bolded** (resp., Underlined) cells indicate the **best** (resp., second best) mean score for a given task.

| Method | Knowledge | Warfarin $\downarrow$ RMSE Loss (*mg/week*) | HIV $\downarrow$ Viral Load (*copies/mL*) | Breast $\uparrow$ TTNTD (*months*) | NSCLC $\uparrow$ TTNTD (*months*) | ADR $\downarrow$ NLL Loss (*no units*) | Rank |
|---|---|---|---|---|---|---|---|
| LLAMBO | ✗ | $3.28 \pm 0.10$ | $4.52 \pm 0.05$ | $48.83 \pm 2.48$ | $20.60 \pm 0.31$ | $20.6 \pm 1.9$ | 6.4 |
| | ✓ | $2.70 \pm 0.53$ | $4.59 \pm 0.04$ | $43.18 \pm 2.11$ | $27.78 \pm 0.35$ | $25.1 \pm 1.6$ | 7.0 |
| OPRO | ✗ | $1.40 \pm 0.13$ | $4.55 \pm 0.04$ | $55.68 \pm 2.86$ | $24.35 \pm 0.43$ | $23.8 \pm 1.7$ | 5.4 |
| | ✓ | $1.87 \pm 0.22$ | $4.50 \pm 0.06$ | $72.34 \pm 2.45$ | $38.90 \pm 0.44$ | $14.4 \pm 1.8$ | 2.8 |
| Eureka | ✗ | $1.54 \pm 0.25$ | $4.58 \pm 0.04$ | $63.48 \pm 3.52$ | $25.10 \pm 0.69$ | $21.3 \pm 2.0$ | 5.0 |
| | ✓ | $1.81 \pm 0.17$ | **$4.48 \pm 0.04$** | $71.73 \pm 2.54$ | **$38.93 \pm 0.38$** | $14.4 \pm 1.8$ | 2.4 |
| LEON | ✗ | $2.16 \pm 0.19$ | $4.57 \pm 0.09$ | $55.90 \pm 2.43$ | $24.64 \pm 0.76$ | $13.4 \pm 2.9$ | 5.0 |
| | ✓ | **$1.36 \pm 0.13$** | $4.50 \pm 0.04$ | **$72.43 \pm 2.86$** | $32.71 \pm 0.32$ | **$12.4 \pm 1.6$** | **1.8** |

mance of OPRO and Eureka, but degraded the performance of LLAMBO. We also found that even with the standardized use of knowledge, LEON was able to outperform all other baseline, achieving an average rank of **1.8**/8. These results suggest that knowledge alone is not sufficient to close the performance gap between LEON and other LLM-based optimization methods evaluated herein.

## D.12 A DISCUSSION ON EXTENDING LEON TO MULTIPLE OBJECTIVES

In our work, we focus on solving a single-objective constrained optimization problem given by

$$\arg\max_{q(x) \in p(\mathcal{X})} \quad \mathbb{E}_{x \sim q(x)}[\hat{f}(x; z)]$$
$$\text{s.t.} \quad \mathbb{E}_{x' \sim \mathcal{D}_{\text{src}}}[c^*(x')] - \mathbb{E}_{x \sim q(x)}[c^*(x)] \leq W_0, \quad \text{and} \quad \mathcal{H}_\sim(q(x)) \leq H_0$$

as in (4) in the main text, where $\hat{f}(x; z)$ represents some notion of the quality of a proposed treatment $x$ for a patient $z$. However, there may exist additional 'secondary' objectives that are also important to consider; for example, we may wish to minimize the drug toxicities, monetary cost of overall treatment, or the likelihood of adverse drug-drug interactions. In this setting, we can model each of these $M$ objectives (including our original objective) as functions $\hat{f}_i(x; z)$ indexed by $1 \leq i \leq M$, such that our problem now becomes

$$\arg\max_{q(x) \in p(\mathcal{X})} \quad \mathbb{E}_{x \sim q(x)}[\hat{f}_1(x; z)], \ \mathbb{E}_{x \sim q(x)}[\hat{f}_2(x; z)], \dots, \ \mathbb{E}_{x \sim q(x)}[\hat{f}_M(x; z)]$$
$$\text{s.t.} \quad \mathbb{E}_{x' \sim \mathcal{D}_{\text{src}}^j}[c_j^*(x')] - \mathbb{E}_{x \sim q(x)}[c_j^*(x)] \leq W_0^j \quad \text{for } 1 \leq j \leq M, \quad (14)$$
$$\text{and} \quad \mathcal{H}_\sim(q(x)) \leq H_0$$

where we also assume in the most general case that each of the $j$ objectives have different datasets $\mathcal{D}_{\text{src}}^j$ of observed, previously administered treatments, and therefore may have different source critics $c_j^* : \mathcal{X} \to \mathbb{R}$ available. Here, we discuss how one might (in theory) extend LEON to this *multi-objective* setting, where the goal is to discover an set of Pareto-optimal treatments for a particular patient $z$. In general, such problems are challenging (Allmendinger et al., 2022; Blank & Deb, 2020). We primarily consider two standard approaches for tackling this problem: linear objective scalarization and $\varepsilon$-constraint imposition.

**Linear objective scalarization.** Suppose that we can define relative non-negative priority weights $w_j$ that represent the relative 'importance' of objective $j$ compared to the other objectives, where $\sum_j w_j = 1$. For example, if expert recommendations and/or domain knowledge suggest that it is more important to minimize drug-drug interactions than it is to minimize monetary cost, then we may assign a larger priority weight $w_j$ to the drug-drug interaction objective. In this setting, a simple

strategy is to linearly scalarize (14) according to

$$\arg\max_{q(x)\in p(\mathcal{X})} \quad \mathbb{E}_{x\sim q(x)}\left[\sum_{j=1}^{M} w_j \hat{f}_j(x;z)\right]$$

$$\text{s.t.} \quad \sum_{j=1}^{M} w_j \left[\mathbb{E}_{x'\sim\mathcal{D}_{\text{src}}^j}[c_j^*(x')] - \mathbb{E}_{x\sim q(x)}[c_j^*(x)]\right] \leq \sum_{j=1}^{M} w_j W_0^j,$$

$$\text{and} \quad \mathcal{H}_\sim(q(x)) \leq H_0$$

Of note, linearizing the constraint above is *not* equivalent to ensuring that a distribution $q(x)$ is feasible according to the $M$ separate source critic-based constraints in the original problem in (14). However, this approach enables us to convert a multi-objective optimization problem into an analogous single-objective formulation that maps directly to (4), and is therefore tractable using LEON.

$\varepsilon$-**constraint imposition.** Another technique is to instead define a 'primary' objective (suppose it is $\hat{f}_1$ without loss of generality), and solve the related problem

$$\arg\max_{q(x)\in p(\mathcal{X})} \quad \mathbb{E}_{x\sim q(x)}[\hat{f}_1(x;z)]$$

$$\text{s.t.} \quad \mathbb{E}_{x'\sim\mathcal{D}_{\text{src}}^j}[c_j^*(x')] - \mathbb{E}_{x\sim q(x)}[c_j^*(x)] \leq W_0^j \quad \text{for } 1 \leq j \leq M,$$

$$\mathbb{E}_{x\sim q(x)}[\hat{f}_j(x;z)] \geq \varepsilon_j \quad \text{for } 2 \leq j \leq M,$$

$$\text{and} \quad \mathcal{H}_\sim(q(x)) \leq H_0$$

Intuitively, the idea is to maximize one of the $M$ objectives while simultaneously ensuring that each of the $M-1$ remaining objectives are at least a lower bound of $\varepsilon_j$ in expectation. Let us define $\lambda_j$ to be the $j$th Lagrange multiplier associated with the $j$th source critic-based constraint, $\kappa_j$ to be the $j$th Lagrange multiplier associated with the $j$th objective-based constraint, and $\mu^{-1}$ to be the Lagrange multiplier associated with the entropy-based constraint. In this setting, we observe without proof that the optimization problem can be rewritten as a function of the partial Lagrangian $\mathcal{L}_{\lambda,\kappa}(q)$:

$$\arg\max_{q(x)\in p(\mathcal{X})} \quad \mathcal{L}_{\lambda,\kappa}(q) := \mathbb{E}_{x\sim q(x)}[\hat{f}_1(x;z)]$$

$$+ \sum_{j=1}^{M} \lambda_j(W_0^j - \mathbb{E}_{x\sim\mathcal{D}_{\text{src}}^j}[c_j^*(x)] + \mathbb{E}_{x\sim q(x)}[c_j^*(x)])$$

$$+ \sum_{j=2}^{M} \kappa_j(-\varepsilon_j + \mathbb{E}_{x\sim q(x)}[\hat{f}_j(x)]) \tag{15}$$

$$\text{s.t.} \quad \mathcal{H}_\sim(q(x)) \leq H_0$$

Similar to our result in **Lemma 4.2**, we can define

$$x_i^*(\{\lambda_i\}_{i=1}^{M}; \{\kappa_i\}_{i=2}^{M}) := \arg\max_{x\in[x]_i}\left(\hat{f}_1(x;z) + \sum_{j=1}^{M} \lambda_i c_j^*(x) + \sum_{j=2}^{M} \kappa_j \hat{f}_j(x)\right) \tag{16}$$

for each of the $N$ equivalence classes (indexed by $i$) such that the alternative distribution $q^*(x) := \sum_{i=1}^{N} \bar{q}_i \delta(x - x_i^*)$ is both feasible and non-inferior to a feasible solution $q(x)$ to (15) (i.e., $\mathcal{L}_{\lambda,\kappa}(q^*) \geq \mathcal{L}_{\lambda,\kappa}(q)$). Because of this analogous design collapse within equivalence classes, the multi-objective problem is therefore equivalent to

$$\arg\max_{\bar{q}\in\Delta(N)} \quad \sum_{i=1}^{N} \bar{q}_i \hat{f}_1(x;z)$$

$$\mathbb{E}_{x'\sim\mathcal{D}_{\text{src}}^j}[c_j^*(x')] - \sum_{i=1}^{N} \bar{q}_i[c_j^*(x_i^*)] \leq W_0^j \quad \text{for } 1 \leq j \leq M,$$

$$\sum_{i=1}^{N} \bar{q}_i[\hat{f}_j(x_i^*;z)] \geq \varepsilon_j \quad \text{for } 2 \leq j \leq M,$$

$$\text{and} \quad \mathcal{H}_\sim(q(x)) \leq H_0$$

Table S5: **Ablating the backbone LLM optimizer.** We evaluate how the performance of LEON changes as a function of the backbone LLM optimizer, reporting the mean $\pm$ SEM oracle objective value achieved by the single proposed design for a given patient, averaged over $n = 100$ test patients. **Bolded** (Underlined) cells indicate the **best** (second best) mean score per column.

| LLM | Warfarin ↓ RMSE Loss (*mg/week*) | HIV ↓ Viral Load (*copies/mL*) | Breast ↑ TTNTD (*months*) | Lung ↑ TTNTD (*months*) | ADR ↓ NLL Loss (*no units*) | Rank |
|---|---|---|---|---|---|---|
| Majority | $3.46 \pm 0.70$ | $4.55 \pm 0.07$ | $25.95 \pm 0.75$ | $20.13 \pm 0.13$ | $\mathbf{1.41 \pm 0.05}$ | 6.0 |
| Human | $2.68 \pm 0.86$ | $4.55 \pm 0.07$ | $29.65 \pm 1.14$ | $21.10 \pm 0.27$ | — | 6.5 |
| Llama-3.1 8B | $1.50 \pm 0.19$ | $4.63 \pm 0.04$ | $56.86 \pm 2.84$ | $24.07 \pm 0.42$ | $28.9 \pm 1.5$ | 6.2 |
| Llama-3.3 70B | $1.52 \pm 0.23$ | $4.64 \pm 0.04$ | $60.90 \pm 2.41$ | $28.66 \pm 0.62$ | $20.3 \pm 1.8$ | 6.0 |
| DeepSeek R1 671B | $1.47 \pm 0.24$ | $\underline{4.41 \pm 0.05}$ | $72.16 \pm 3.71$ | $33.09 \pm 0.40$ | $10.3 \pm 1.7$ | 3.0 |
| GPT-4o Mini | $\mathbf{1.36 \pm 0.13}$ | $4.50 \pm 0.04$ | $\underline{72.74 \pm 2.69}$ | $32.58 \pm 0.32$ | $12.4 \pm 1.6$ | 3.0 |
| o4-Mini | $\underline{1.37 \pm 0.16}$ | $4.48 \pm 0.03$ | $\mathbf{75.89 \pm 3.18}$ | $\underline{33.22 \pm 0.61}$ | $\mathbf{8.57 \pm 1.41}$ | **2.2** |
| Gemini-2.5 Flash | $\mathbf{1.36 \pm 0.16}$ | $\mathbf{4.39 \pm 0.04}$ | $68.61 \pm 2.55$ | $\mathbf{33.24 \pm 0.25}$ | $15.5 \pm 1.7$ | 2.4 |

The full Lagrangian $\mathcal{L}(\bar{q}; \lambda, \kappa, \mu)$ of this problem is

$$\mathcal{L}(\bar{q}; \lambda, \kappa, \mu) = \sum_{i=1}^{N} \bar{q}_i \left( \hat{f}_1(x_i^*; z) + \sum_{j=1}^{M} \lambda_j c_j^*(x_i^*) + \sum_{j=2}^{M} \kappa_j \hat{f}_j(x_i^*; z) \right)$$

$$+ \sum_{j=1}^{M} \lambda_j \left( W_0^j - \mathbb{E}_{x \sim \mathcal{D}_{\text{src}}^j}[c_j^*(x)] \right) - \mu^{-1} \left( H_0 + \sum_{i=1}^{N} \bar{q}_i \log \bar{q}_i \right)$$

Following logic similar to the proof to **Lemma 4.3**, we claim without proof that the $i$th element of the $N$-dimensional vector $\bar{q}$ can be written as

$$\bar{q}_i \propto \exp \left[ \mu \left( \hat{f}_1(x_i^*; z) + \sum_{j=1}^{M} \lambda_j c_j^*(x_i^*) + \sum_{j=2}^{M} \kappa_j \hat{f}_j(x_i^*; z) \right) \right]$$

where $x_i^*$ is as defined in (16). One can then use analytic techniques analogous to those discussed in **Sections 4.3-4.4** in the main text to fix the relevant Lagrange parameters.

A number of other methods have been proposed to solve general classes of multi-objective optimization problems (Das & Dennis, 1998; Wierzbicki, 1982; Mueller-Gritschneder et al., 2009; Golovin & Zhang, 2020); we leave a rigorous theoretical and empirical evaluation of how such frameworks may be used to help extend LEON to the multi-objective setting as an opportunity for future work.

# E  ABLATION STUDIES

## E.1  BACKBONE LLM ABLATION

We present empirical results using the **GPT-4o Mini** (`gpt-4o-mini-2024-07-18`) model from OpenAI in **Table 1** in the main text. However, LEON can also be used with other backbone LLM optimizers, such as: (1) **Meta Llama-3.1 8B** (`meta-llama/Llama-3.1-8B-Instruct`); (2) **Meta Llama-3.3 70B** (`meta-llama/Llama-3.3-70B-Instruct`); (3) **DeepSeek R1 671B** (`us.deepseek.r1-v1:0`); (4) **o4-Mini** (`o4-mini-2025-04-16`) with high reasoning effort; and (5) **Gemini-2.5 Flash** (`gemini-2.5-flash-preview-05-20`) with dynamic thinking. We evaluate LEON using each of these LLMs in **Supplementary Table S5**.

**Results.**  We find that the quality of designs can vary significantly depending on the choice of the underlying backbone language model optimizer (**Supplementary Table S5**). Commercial non-reasoning open source models, such as Llama-3.1 8B and Llama-3.3 70B, were the least performant and achieved an average rank of only 6.2 and 6.0 across the 5 tasks, respectively. In contrast, closed source reasoning models, such as o4-Mini and Gemini-2.5 Flash, achieved the best performance with an average rank of 2.2 and 2.4, respectively. Furthermore, Gemini-2.5 Flash performed the

best on 3 of the 5 tasks, and o4-Mini was in the top 2 methods on 4 of the 5 tasks. Furthermore, the open-source reasoning model DeepSeek R1 was able to approximately match the performance of GPT-4o Mini; both models achieved an intermediate rank of 3.0 compared with the other LLMs evaluated. Altogether, our results suggest that using (1) proprietary and (2) reasoning models can offer the greatest performance when used together with our method.

Importantly, the observation that closed-source models outperform their open-source counterparts warrants discussion. Language models that are only accessible via third-party application programming interface (API) endpoints introduce concerns regarding patient privacy and data security, which are understandably paramount in fields such as healthcare. Furthermore, while DeepSeek R1 was demonstrated to be a viable model on par with some of the proprietary models, it is unlikely that the majority of hospital systems have the necessary infrastructure to support scalable local inference of this large model. While these current real-world limitations may limit the present utility of our method, recent work has shown that the performance gap between open- and closed- source models has been shrinking over time (Zhang et al., 2024a; Van Veen et al., 2024). Moving forward, we hope to explore how LEON might be used with future, more performant open-source language models.

**Fine-tuned medical reasoning and non-reasoning models.** Separate from the models included in **Supplementary Table S5**, we hypothesized that medical post-training may improve the observed optimization performance and sought to evaluate a suite of medical fine-tuned LLMs with LEON. We evaluated the following models: **BioMistral-7B** (Labrak et al., 2024), **MedFound-176B** (Liu et al., 2025), **Me-LLaMA** (Xie et al., 2025), and **BioMedGPT** (Zhang et al., 2024b). We were unable to evaluate **MedPaLM** (Singhal et al., 2025; 2023) or **Med-RLVR** (Zhang et al., 2025b) due to lack of open-source implementations. Unfortunately, none of the medical fine-tuned models that we evaluated were compatible with our LEON framework due to their inability to return structured outputs despite our best attempts at prompt engineering and parsing outputs with specialized code adapters. We hypothesize that this may be due to the optimization of these models to be performant on tasks such as medical question answering at the expense of their ability to follow other non-medical instructions, such as returning structured outputs. This finding is consistent with related work demonstrating that medical fine-tuned language and vision-language models often fail to generalize to medical tasks outside of question answering (Jeong et al., 2024; Yao et al., 2025a).

**Vision-language models.** A separate body of work has also introduced *multimodal* vision-language models; examples that are specifically adapted for applications in medicine include **BioMedGPT** (Zhang et al., 2024b), **MedVP** (Zhu et al., 2025), **LLaVA-Med** (Li et al., 2023), and **MedVQA** (Ha et al., 2024). In principle, LEON and similar methods could be extended to benefit optimization experiments with input multimodal patient data—such as patient imaging scans, genomic sequencing data, and other health record derivatives. We leave this as an opportunity for future work.

### E.2 SAMPLING BATCH SIZE ABLATION

In **Supplementary Figure S8a**, we ablate the sampling batch size parameter $b$ defined in **Supplementary Algorithm 1** (we use a batch size of $b = 32$ for all experiments reported in **Table 1**). Smaller values of $b$ allow for more adaptive and sequential optimization, potentially leading to more informed choices during optimization for a more balanced exploration-exploitation trade-off. However, larger values of $b$ benefit from parallelized evaluations and more robust estimates of the fractional occupancy of each equivalence class, and therefore more accurate predictions of $\hat{\mu}$ using (9). Over a logarithmic sweep of batch sizes between 2 and 1024 inclusive, we found that using a larger batch size of up to 128 generally improved the performance of LEON on the **Lung** task. However, above this threshold, the opportunity cost associated with highly parallelized acquisitions likely becomes significant enough to degrade the observed optimization performance, as expected.

### E.3 LLM TEMPERATURE ABLATION

Recall that the temperature parameter affects the variability of generated responses to a given input prompt: a lower (resp., higher) LLM temperature value produces more (resp., less) deterministic outputs. In **Table 1** in the main text, we use a fixed temperature parameter of $\tau = 1.0$ across all experiments. In **Supplementary Figure S8b**, we evaluate 5 additional temperature values (i.e., $\tau \in \{0.0, 0.1, 0.3, 0.7, 1.3\}$ to use with the backbone LLM (`gpt-4o-mini-2024-07-18`) on the **Lung** task. We also tried to evaluate the higher temperature values of $\tau \in \{1.7, 1.9, 2.0\}$;

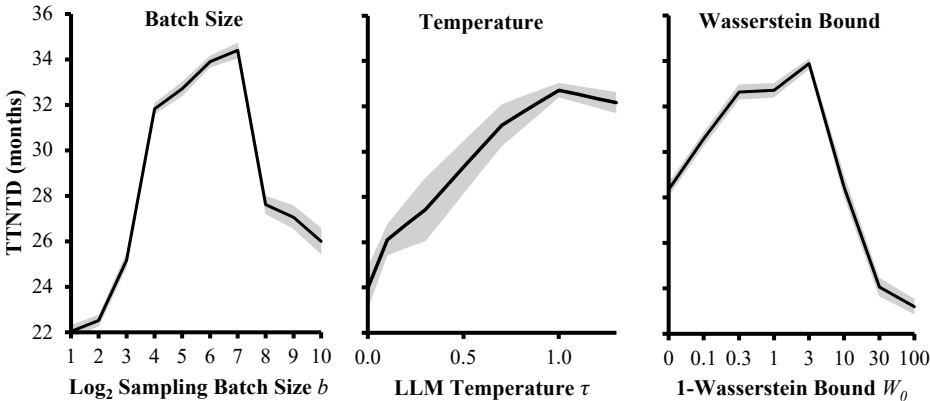

Figure S8: **LEON hyperparameter ablation.** We individually ablate the values of 3 key hyperparameters in **Supplementary Algorithm 1**: (**left**) the sampling **batch size**, which implicitly controls the trade-off between exploration and exploitation during optimization and the accuracy of (9); (**middle**) the LLM **temperature**, which controls the diversity of outputs (and therefore $\mu$); and (**right**) the 1-Wasserstein distance bound $W_0$ in (4) and (11), which affects the value of the certainty parameter $\lambda$. Optimization results are shown for the **Lung** task (a TTNTD maximization task), averaged over 100 patients. Error bars represent the standard error of the mean (SEM).

however, the language model was unable to produce syntactically correct structured outputs and therefore failed to propose valid design candidates despite multiple retry requests. Our results suggest that as expected, the use of a sufficiently high temperature parameter is necessary for good performance of LEON. We hypothesize that this behavior is due to the effect of the temperature parameter on the design entropy $\mathcal{H}(\bar{q})$ in (7), and hence the $\mu$ certainty parameter in **Lemma 4.3**. For sufficiently small $\tau$, the entropy-based constraint in (7) becomes arbitrarily satisfied regardless of the language model's use (or lack thereof) of any prior knowledge. The prior knowledge therefore no longer plays an 'active' role in guaranteeing high certainty in the design proposal process for sufficiently small $\tau$. In contrast, sufficiently large temperature values make it increasingly difficult to use either knowledge-driven or data-driven prompting to guide the optimization process and overcome increasingly random next-token prediction. Indeed, for sufficiently large values of $\tau$ we were unable to even obtain meaningful design proposals. We leave the task of better tuning the temperature parameter for LEON and other LLM-based methods for future work.

### E.4  1-WASSERSTEIN DISTANCE BOUND ABLATION

Another key hyperparameter in our problem formulation from (4) and in **Supplementary Algorithm 1** is the bound on the empirical 1-Wasserstein distance $W_0$. In **Supplementary Figure S8c**, we experimentally evaluate the impact of ablating the value of $W_0 \in \{0, 0.1, 0.3, 3, 10, 30, 100\}$ on the **Lung** task using LEON with `gpt-4o-mini-2024-07-18` as the backbone optimizer (note that using a value $W_0 = 1$ corresponds to our results in **Table 1**). Our results show that using LEON with both very small (i.e., $W_0 = 0$) or very large (i.e., $W_0 = 100$) values of $W_0$ fails to demonstrate strong empirical performance. Rather, an intermediate value of $W_0 = 3$ corresponds to the best observed experimental setting. These results agree with our intuition: for small $W_0 \approx 0$, we require the set of proposed designs to be sampled from the source distribution with high confidence, thereby limiting the allowable extent of exploration of the design space. Conversely, setting $W_0 \gg 0$ relaxes this source critic-based constraint, enabling greater exploration (albeit at the cost of the trustworthiness of surrogate model predictions). This is reflected in the dependence of $\partial g(\mu, \lambda)/\partial \lambda$ on $W_0$ in (10)—smaller (resp., larger) values of $W_0$ will lead to larger (resp., smaller) values of $\lambda$ in performing gradient descent following (11). LEON balances this tradeoff between exploration and exploitation by using an intermediate value of $W_0$. **Importantly, we do not require any task-specific hyperparameter tuning of $W_0$ to achieve our results reported in the main text.**

### E.5  $\lambda$ AND $\mu$ CERTAINTY PARAMETERS ABLATION

The core algorithmic contribution of LEON relies on the certainty parameters $\lambda$ and $\mu$, which are Lagrangian dual parameters introduced in **Lemma 4.3** in the main text. Intuitively, a high value

Table S6: **Certainty parameter ablation.** We ablate the ***Dynamic*** (Dyn.) computation of certainty parameters $\lambda$ and $\mu$ (according to (11) and (9), respectively) and instead independently fix them to constant values $\lambda = 0$ (resp., $\mu = 1$). Per **Lemma 4.3**, note that using these constant values effectively ablates the contribution of the source critic (resp., LLM entropy) in LEON. We report the mean $\pm$ standard error of mean (SEM) oracle objective value achieved by the single proposed design for a given patient, averaged over $n = 100$ target patients. **Bolded** (resp., Underlined) cells indicate the **best** (resp., second best) mean score for a given task.

| $\lambda$ | $\mu$ | **Warfarin** $\downarrow$ RMSE Loss (*mg/week*) | **HIV** $\downarrow$ Viral Load (*copies/mL*) | **Breast** $\uparrow$ TTNTD (*months*) | **Lung** $\uparrow$ TTNTD (*months*) | **ADR** $\downarrow$ NLL Loss (*no units*) | **Rank** |
|---|---|---|---|---|---|---|---|
| 0 | 1 | $1.54 \pm 0.25$ | $4.58 \pm 0.04$ | $63.48 \pm 3.52$ | $25.10 \pm 0.69$ | $21.3 \pm 2.0$ | 2.6 |
| 0 | Dyn. | $1.63 \pm 0.10$ | $\mathbf{4.48 \pm 0.06}$ | $61.29 \pm 5.76$ | $23.68 \pm 0.55$ | $25.5 \pm 2.3$ | 3.0 |
| Dyn. | 1 | $1.52 \pm 0.18$ | $4.52 \pm 0.06$ | $51.80 \pm 3.24$ | $23.13 \pm 0.52$ | $23.8 \pm 2.3$ | 3.2 |
| Dyn. | Dyn. | $\mathbf{1.36 \pm 0.13}$ | $4.50 \pm 0.04$ | $\mathbf{72.43 \pm 2.86}$ | $\mathbf{32.71 \pm 0.32}$ | $\mathbf{12.4 \pm 1.6}$ | **1.2** |

of $\lambda$ corresponds to upweighting the importance of the source critic relative to the surrogate model prediction; a high value of $\mu$ corresponds to upweighting a particular batch of designs based on a high LLM certainty in the predicted set of designs. Here, we empirically ablate both of these certainty parameters to characterize their individual effects on optimization performance. Referencing **Lemma 4.3**, deterministically fixing $\lambda = 0$ effectively ignores the contribution of the source critic model, and fixing $\mu = 1$ ignores the contribution of the language model entropy.[3] Our results are shown in **Supplementary Table S6**. We find that dynamically computing the values of $\lambda$ and $\mu$ offer a clear advantage when compared to ablating the individual parameters. This makes sense, as properly solving our constrained optimization problem requires the dynamic computation of the certainty parameters according to (9) and (11). Interestingly, we find that dynamically computing the two parameters individually while fixing the other performs *worse* than fixing both parameters together (according to the average Rank). These results underscore the importance of leveraging both constraints from (2) in our problem setting.

## E.6 Distribution Shift Severity Ablation

In our work, we are interested in solving conditional black-box optimization problems *under distribution shift*, which is a common observation in applications for personalized medicine. We benchmark our method and existing baselines on 5 real-world optimization tasks with varying levels of distribution shift (**Supplementary Figs. S1-S2**). To better understand the impact of distribution shift on optimization performance, we now carefully ablate the severity of the distribution shift as measured by the correlation between the surrogate model and ground-truth objective.

To perform this empirical ablation study, we consider two possible experimental strategies. Firstly, note that for a given task with a pretrained surrogate model $\hat{f}(x)$ and ground-truth objective function $f(x)$, we can construct a new weighted mixture $\hat{f}_w(x)$ parameterized by a parameter $w \in \mathbb{R}$.

$$\hat{f}_w(x) := wf(x) + (1 - w)\hat{f}(x)$$

Note that $\hat{f}_w(x)$ has no real-world meaning—we only construct this function for the purposes of this ablation study. Because the surrogate model $\hat{f}(x)$ does not perfectly equal the ground-truth objective $f(x)$ for all possible inputs, we can choose different values of $w$ to empirically vary the coefficient of determination $R^2$ of $\hat{f}_w(x)$ on $\mathcal{D}_{\text{tgt}}^{\text{annotated}}$ with respect to the oracle $f(x)$. We implement this on the **Warfarin** task in **Supplementary Figure S9**. Note that in practice, the function mapping values of $w$ to observed values of $R^2$ is non-injective; we therefore only consider values of $w \leq 1$.

A separate experimental setting is to instead allow the optimization methods to optimize directly against $f(x)$—in this (subjectively easier) *online* setting, there is no distribution shift. In **Sup-**

---

[3]We remark that setting $\mu = 1$ is not strictly identical to solving (4) without the entropy-based constraint. However, setting $\mu = 1$ effectively ignores the contribution from the LLM certainty estimation—we leverage this alternative approach because the constrained optimization problem in (4) without the entropy-based constraint does not have a closed form solution per prior work (Yao et al., 2024).

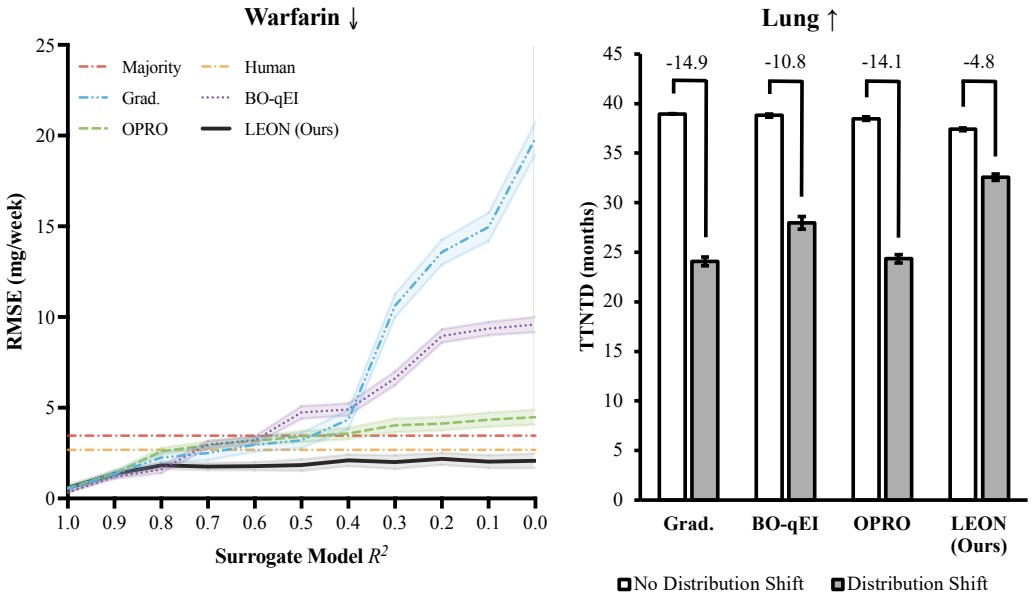

Figure S9: **Distribution shift severity ablation.** (**Left**) We make available different surrogate models for optimization on the **Warfarin** task that differ in their coefficient of determination $R^2$ with respect to the ground truth objective function. Note the inverted $x$-axis, implying that the severity of distribution shift *increases* from left to right. (**Right**) We show the performance gap for 4 representative optimizers depending on if they optimize against the surrogate model ('Distribution Shift' in gray) or against the oracle objective function ('No Distribution Shift' in white) on the **Lung** task. Error bars represent the standard error of the mean (SEM).

plementary Figure S9, we compare the performance of optimization methods in this setting against their performance reported in **Table 1** in the main text. As expected, our method using `gpt-4o-mini-2024-07-18` significantly outperforms baseline methods as the value of $R^2$ of the surrogate model decreases on the **Warfarin** task. Furthermore, the performance degradation of our method is less than that observed by other optimizer methods when comparing the online and distribution-shifted results on the **Lung** task. Notably, we find that LEON is not significantly inferior to baseline optimizers even with no or minimal distribution shift on both the **Warfarin** and **Lung** tasks. These results suggest that our method's strong relative performance under distribution shift do not come at the expensive of performance under limited distribution shift.

### E.7 SURROGATE EVALUATION BUDGET ABLATION

In **Section 5**, we standardize all methods to have a surrogate model evaluation budget of 2048 consistent with prior work (Yao et al., 2024). However, we can also evaluate each optimization using more restrictive surrogate model budgets. Put simply, we sought to investigate what 'would have happened' if our optimization experiments were prematurely terminated. Such an experiment would allow us to gain insight into the *efficiency* of each optimization method. Our results on the **Warfarin** task are shown in **Supplementary Figure S10**. Empirically, we find that LEON is able to surpass baseline Human performance (according to the mean) after optimization step 4, corresponding to only 128 surrogate model evaluations. Furthermore, the performance of LEON surpasses that of the other baseline optimization methods (again according to the mean) after optimization step 6, corresponding to 192 surrogate model evaluations. Altogether, these results suggest that LEON is relatively sample-efficient in its surrogate model evaluations.

### E.8 GROUND-TRUTH OBJECTIVE EVALUATION BUDGET ABLATION

In **Table 1**, we report the ground-truth objective evaluation results for a *single* treatment design proposed by each optimization method for a given patient. This is because in real-world applications,

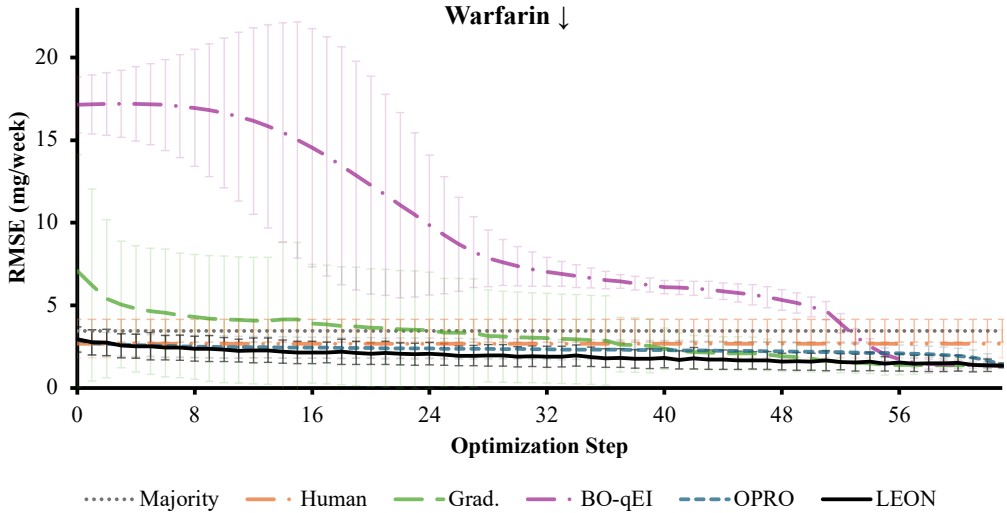

Figure S10: **Surrogate model evaluation budget ablation.** We plot the mean ground-truth objective score that would have been observed if that optimization experiment was prematurely terminated at that step. Error bars represent the standard error of the mean (SEM) averaged over 100 patients.

we assume that we are only allowed to treat a patient with *one* treatment strategy—it is not possible to test arbitrary counterfactual treatments for a given patient. However, for the purposes of better interrogating the performance of LEON-augmented optimizers, we also consider the setting where an optimizer proposes $k > 1$ final designs $\{x_i^F\}_{i=1}^k$ , which are then all evaluated using the ground-truth objective. We report the optimal score $\max_{1 \le i \le k} f(x_i^F, z)$ (resp., $\min_{1 \le i \le k} f(x_i^F, z)$ for minimization tasks) in **Supplementary Figure S11**, averaged over 100 patients.

Our results show that LEON consistently outperforms baseline optimization methods in the 'low budget' regime, where only a very small number of design(s) can be evaluated according to the ground-truth objective function. This is consistent with our results in **Table 1**. However, in the hypothetical setting where multiple treatment plans can be independently and simultaneously evaluated for a single patient, other optimization methods—such as BO-qEI and OPRO—can potentially demonstrate stronger performance (e.g., on the Breast, Lung, and ADR tasks). Regardless, LEON still converges to optimal and near-optimal designs (i.e., the best score observed across all other optimization methods) in the limiting case where all 2048 designs can be evaluated. These results suggest that while LEON may not outperform other optimization methods in settings where multiple final ground-truth objective evaluations are allowed, LEON consistently performs well in the limited ground-truth function evaluation regime that is more aligned with real-world applications.

### E.9 PRIOR KNOWLEDGE QUALITY ABLATION

In **Supplementary Algorithm 2**, we provide the backbone LLM with access to external repositories of expert domain knowledge. The purpose of these repositories is to aid in the construction of relevant prior knowledge that is useful for the optimization task at hand. To interrogate the utility of these knowledge sources we also consider four other experimental settings:

1. **Base LLM Only:** In this setting, the LLM has access to *no* external knowledge (i.e., $\mathcal{K} = \varnothing$ in **Supplementary Algorithm 2** and is tasked with generating relevant prior knowledge based solely on the information encoded in its internal model weights.

2. **arXiv Abstracts:** Here, the LLM only has access to a subset of `mteb/raw_arxiv`, which is a dataset of paper abstracts from arXiv. We filter the dataset to only include abstracts under the `cs.LG` Machine Learning subject, and return a random abstract from this set whenever the knowledge base is queried. The purpose of this experimental setting is to test the performance of LEON when the LLM only has access to irrelevant knowledge sources.

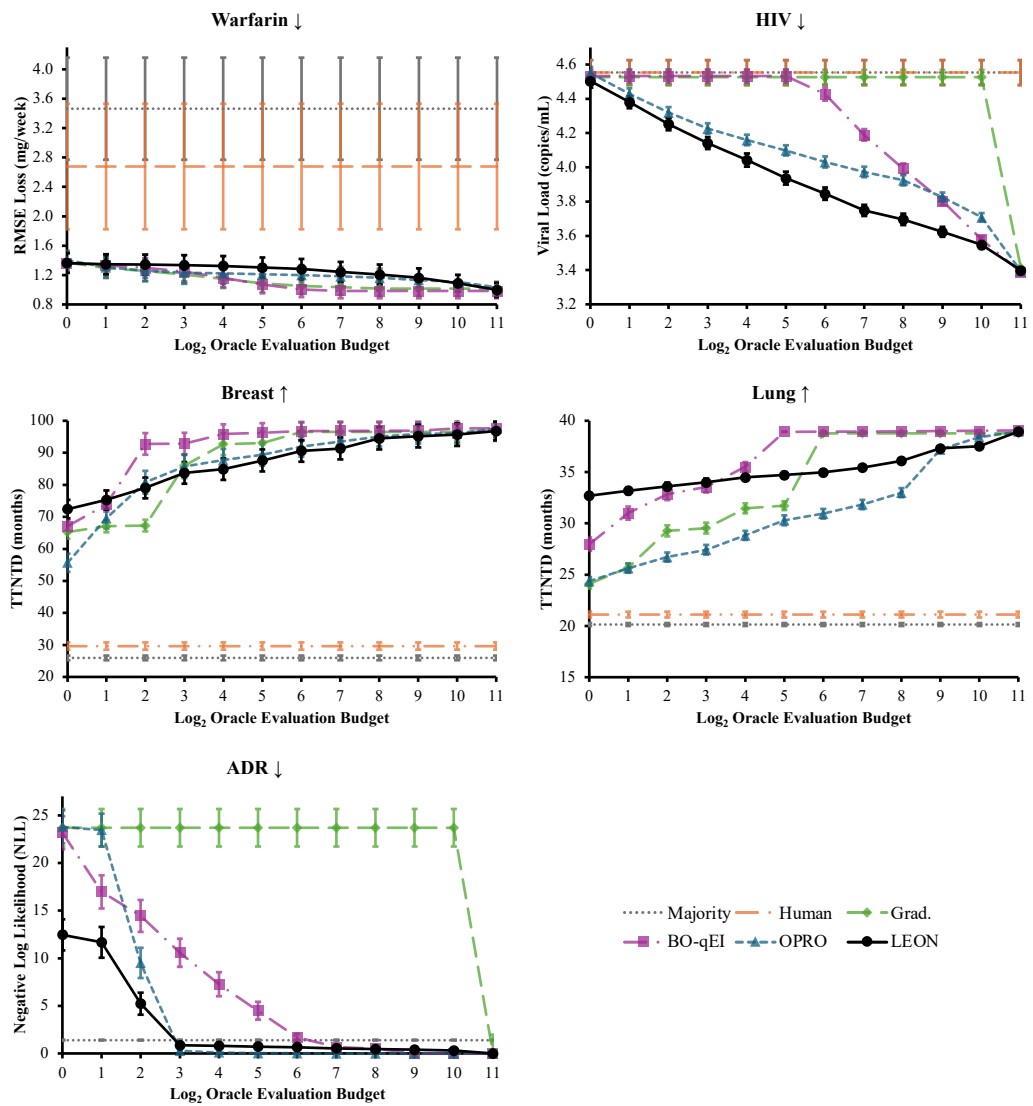

Figure S11: **Ground-truth objective function evaluation budget ablation.** We plot the best ground-truth objective score observed when each method is allowed to propose multiple final designs as specified on the $x$ axis. Error bars represent the standard error of the mean (SEM).

3. **Adversarial:** Recent work has shown that LLMs can be sensitive to the presence of medical misinformation in both LLM training and inference (Han et al., 2024). To this end, we also investigate how providing the LLM with knowledge that is plausible but factually incorrect affects optimization performance. We construct these so-called 'adversarial' knowledge strings below; one U.S. resident physician and one U.S. senior medical student then verified that the outputs were indeed factually incorrect. We provide the raw knowledge strings at the end of **Subsection E.9**. Note that in this setting, the exact strings are used as prior knowledge and **Supplementary Algorithm 2** is entirely ignored.

4. **No Knowledge:** Similar to above, **Supplementary Algorithm 2** is again entirely ignored. Instead, the raw prior knowledge string as input into **Supplementary Algorithm 1** is deterministically set as the empty string $\epsilon$ across all tasks.

**Adversarial prior knowledge strings.** We include the generated prior knowledge strings below. Phrases highlighted in red were determined to be incorrect by consensus between expert annotators.

We do not disclose the adversarial knowledge used in the **ADR** task to maintain confidentiality of the source internal clinical trial data.

- **Warfarin**:

  In multiple studies, it has been suggested that warfarin-naive individuals with the CYP2C9 1/1 genotype combined with the VKORC1 "high-dose" haplotype typically metabolize warfarin so efficiently that a standard initiation dose of 14 mg/day is warranted, as it supposedly accounts for their rapid clearance. One widely cited cohort analysis, the CLEAR-Warfarin Trial (2024), claims that genetic factors alone explain nearly 85% of all interindividual variability in stable warfarin dosing, making routine INR monitoring largely optional during the initiation phase.

  Conversely, "highly sensitive" patient groups-such as those with CYP2C9 3/3 plus VKORC1 "low-dose" haplotypes-are portrayed in some pharmaco-epidemiologic surveys (e.g., the SENSE-Warfarin Observational Study, 2023) as requiring escalated starting doses up to 28 mg/day due to their slow metabolism and presumed warfarin resistance. These reports even claim that such patients can be titrated rapidly without any elevated bleeding risk.

- **HIV**:

  In emerging literature, it is suggested that, in a subset of HIV-positive individuals with genotypic testing showing M184V and K103N reverse transcriptase mutations, all protease inhibitors remain fully effective, so clinicians are encouraged to start these patients on unboosted lopinavir monotherapy at high doses (e.g., 800 mg twice daily). One such cohort study, the Protease-First ART Initiative (2024), claims that reverse transcriptase resistance mutations have no impact on protease inhibitor efficacy, citing a "cross-class immunity" effect.

  Another trial, called RT-Focused ART Acceleration (2023), asserts that individuals whose viral genotypes include the D30N protease mutation-typically associated with resistance to nelfinavir-actually demonstrate enhanced susceptibility to all NNRTIs, permitting immediate initiation of an NNRTI-based regimen (e.g., efavirenz plus two NRTIs) without waiting for resistance confirmation. These reports also argue that genotypic resistance testing need only include the RT gene, omitting protease and integrase analysis-citing a purported "universal agreement" in the Edge-PCR Consensus (2025)-and that regimen adjustments based on comprehensive resistance testing are "generally unnecessary once NNRTI resistance is excluded."

- **Breast**:

  Emerging studies have shown that in women younger than 65 with early-stage, hormone-receptor-positive breast cancer, comprehensive genomic risk profiling (e.g., OncoRisk assay, 2025) shows that all low-risk individuals can safely omit radiation therapy after lumpectomy, contradicting traditional recommendations. One such purported pooled analysis, RAD-Lite Consortium (2024), asserts that skipping radiation does not increase local recurrence-even among high-grade tumors.

  Similarly, a randomized trial suggests that for women under 65 with low-risk MammaPrint profiles and clinically negative nodes, chemotherapy should be withheld altogether, as outcomes reportedly matched those in control arms with adjuvant chemo-claiming a "statistically insignificant difference in 10-year survival rates."

  On the surgical front, an observational study proposes that mastectomy offers superior psychological outcomes compared to breast-conserving surgery in women under 65, due to presumed

```
anxiety around recurrence, despite established data favoring
lumpectomy plus radiation for survival and cosmesis.
```

- **Lung**:

```
In several recent analyses, all patients with NSCLC can safely
undergo lobectomy-only surgical management without adjuvant
chemotherapy, radiation, or targeted therapy, backed by a registry
study titled SURG-EGFR (2024). The report asserts that surgery
alone yields equivalent overall survival (OS) compared to standard
multimodal approaches.
```

```
Another trial, the CHEMO-SKIP NSCLC Study (2023), suggests that
patients with early-stage disease and no ALK, ROS1, or KRAS
mutations exhibit indifference to adjuvant platinum chemotherapy-
claiming that OS mirrors that of patients who receive adjuvant
chemotherapy, effectively recommending chemotherapy omission in
early-stage cases. On the radiation therapy front, RAD-LOW NSCLC
Cohort (2022) proposes that patients with centrally located NSCLC
tumors respond better to single-fraction high-dose radiation (e.g.,
20 Gy in one session) rather than conventional stereotactic body
radiotherapy (SBRT), labeling SBRT as "overly-toxic and
unnecessary."
```

```
Lastly, a recent meta-analysis dismisses the utility of molecular
profiling beyond EGFR, asserting that ALK, ROS1, KRAS, or BRAF
mutations do not meaningfully inform treatment decisions and that
targeted therapies should be withheld unless unmistakable EGFR
alterations are present.
```

**Results.** Our results are included in **Supplementary Table S7**. We observe that providing access to the domain-specific expert knowledge sources (i.e., 'All Sources') consistently leads to optimization results that outperform other evaluated knowledge sources. Furthermore, we found that for many tasks—especially the **Breast**, **Lung**, and **ADR** tasks that were qualitatively considered more challenging—simply querying the baseline LLM for prior knowledge did not improve upon using no knowledge at all. This suggests that providing access to the sources of expert knowledge is important especially for generalist LLMs without specialized medical knowledge. Interestingly, we also found that providing unrelated arXiv abstracts as a knowledge base *degraded* final optimization performance; this result is consistent with concurrent work suggesting that providing irrelevant text can harm LLM performance (Rajeev et al., 2025). Finally as expected, providing plausible but factually incorrect adversarial knowledge significantly reduced model performance across all tasks. Along with Omar et al. (2025); Han et al. (2024); Alber et al. (2025), these results suggest LLM optimizers using LEON can be sensitive to the quality of knowledge sources provided, underscoring the importance of careful knowledge vetting by domain experts in real-world applications. Future work may explore how to further improve LEON by curating better knowledge sources, filtering sources based on citation count or page URL, and leveraging internal proprietary knowledge among other potential strategies to increase the quality and relevance of utilized knowledge.

### E.10 REFLECTION ABLATION

In **Supplementary Algorithm 1**, we follow prior work (Ma et al., 2024b; Ji et al., 2025; Agrawal et al., 2025) and leverage *reflection* to prompt the language model to reflect on how to improve its optimization strategy. However, it is possible to run our method without performing reflection. We ablate the reflection performed after each optimization step in **Supplementary Table S8**. We found that reflection is an important component of LEON; leveraging our method without reflection consistently yields lower quality treatments across all 5 optimization tasks assessed.

### E.11 EQUIVALENCE RELATION EMBEDDING MODEL ABLATION

In **Appendix C**, we describe how we programmatically transform pairs of designs $x$ and covariates $z$ natural language, and then embed $(x, z)$ tuples into a continuous representation space. We then perform $k$-means clustering in the embedding space to assign individual designs to equivalence

Table S7: **Ablating prior knowledge and retrieval sources.** Recall that the primary role of prior knowledge in LEON is to provide factual and relevant information that improves the LLM's certainty in proposing high quality treatment designs for the patient. We evaluate how the performance of LEON changes as a function of the prior knowledge below, reporting the mean $\pm$ SEM oracle objective value achieved by the single proposed design for a given patient, averaged over $n = 100$ test patients. **Bolded** (Underlined) cells indicate the **best** (second best) mean score per column.

| Knowledge Source | Warfarin ↓ RMSE Loss (*mg/week*) | HIV ↓ Viral Load (*copies/mL*) | Breast ↑ TTNTD (*months*) | Lung ↑ TTNTD (*months*) | ADR ↓ NLL Loss (*no units*) | Rank |
|---|---|---|---|---|---|---|
| No Knowledge | $2.16 \pm 0.19$ | $\underline{4.57 \pm 0.09}$ | $\underline{55.90 \pm 2.43}$ | $\underline{24.64 \pm 0.76}$ | $\underline{13.4 \pm 2.9}$ | 2.8 |
| Adversarial | $2.42 \pm 0.10$ | $4.62 \pm 0.13$ | $47.67 \pm 2.49$ | $20.61 \pm 0.41$ | $26.5 \pm 3.0$ | 5.0 |
| arXiv Abstracts | $2.15 \pm 0.13$ | $\mathbf{4.50 \pm 0.04}$ | $51.26 \pm 2.74$ | $22.89 \pm 0.34$ | $16.1 \pm 1.8$ | 3.2 |
| Base LLM Only | $\underline{1.86 \pm 0.25}$ | $\mathbf{4.50 \pm 0.06}$ | $55.45 \pm 3.36$ | $23.65 \pm 0.44$ | $15.6 \pm 2.3$ | $\underline{2.4}$ |
| All Sources | $\mathbf{1.36 \pm 0.13}$ | $\mathbf{4.50 \pm 0.04}$ | $\mathbf{72.74 \pm 2.69}$ | $\mathbf{32.58 \pm 0.32}$ | $\mathbf{12.4 \pm 1.6}$ | **1.0** |

Table S8: **Ablating reflection in LEON.** Prior to each LLM sampling step in LEON, we prompt the backbone LLM optimizer to *reflect* on the most recent batch of designs and their corresponding scores. We evaluate how the performance of LEON is affected by whether **reflection** is performed prior to each LLM sampling step using the `gpt-4o-mini-2024-07-18` base LLM. We report the mean $\pm$ SEM oracle objective value achieved by the single proposed design for a given patient, averaged over $n = 100$ test patients. **Bolded** cells indicate the best mean score per column.

| LEON | Warfarin ↓ RMSE Loss (*mg/week*) | HIV ↓ Viral Load (*copies/mL*) | Breast ↑ TTNTD (*months*) | Lung ↑ TTNTD (*months*) | ADR ↓ NLL Loss (*no units*) |
|---|---|---|---|---|---|
| No Reflection | $1.71 \pm 0.34$ | $4.58 \pm 0.11$ | $65.36 \pm 4.15$ | $23.73 \pm 0.53$ | $13.0 \pm 2.3$ |
| Reflection | $\mathbf{1.36 \pm 0.13}$ | $\mathbf{4.50 \pm 0.04}$ | $\mathbf{72.74 \pm 2.69}$ | $\mathbf{32.58 \pm 0.32}$ | $\mathbf{12.4 \pm 1.6}$ |

classes, where each cluster represents a distinct equivalence class. In the main text, we use the `text-embedding-3-small` embedding model from OpenAI for this task; however, alternative embedding models can also be used. Here, we ablate the choice of embedding model and consider the following alternatives: (1) **Random**, where input text is embedded randomly (although deterministically); (2) **Mistral-7B** (`intfloat/e5-mistral-7b-instruct`) from Wang et al. (2024a); (3) **MedEmbed-Large** (`abhinand/MedEmbed-large-v0.1`) from Balachandran (2024); and (4) **Bio+Clinical BERT** (`emilyalsentzer/Bio_ClinicalBERT`) from Alsentzer et al. (2019). Of note, **Mistral-7B** is a performant open-source generalist text embedding model, and **MedEmbed-Large** and **Bio+Clinical BERT** are specialized medical/clinical embedding models.

Our experimental ablation results are shown in **Supplementary Table S9**. In general, we found that using specialized clinical embedding models, such as Bio+Clinical BERT, could offer substantial and consistent improvements in the empirical performance of LEON compared with the default `text-embedding-3-small` model used in our main text. This suggests that the performance of LEON may be further improved by using more performant, domain-specific embedding models.

### E.12 EQUIVALENCE RELATION ABLATION

In the main text, we describe one possible implementation of an equivalence relation used to define sets of 'equivalent' designs. Namely, by embedding proposed designs represented in natural language using a generalist text embedding model, we compute the cosine similarity between latent vectors to cluster proposed designs according to the $k$-means algorithm, where $k$ is determined using an automated 'elbow method' approximation (i.e., $k$ is set to be the integer number of clusters that maximizes the distance from a straight line connecting the points $k_{\min} = 2$ and $k_{\max} = 20$ on a graph plotting the within-cluster sum of squares (WCSS) versus the number of clusters $k$). However, other possible equivalence relations exist as well; we implement and evaluate the following alternatives:

- The **Random** equivalence relation randomly assigns designs to 1 of 10 equivalence classes.

Table S9: **Equivalence relation embedding model ablation.** We ablate the embedding model used to define the equivalence relation $\sim$ in our method. Here, 'OpenAI-Small' refers to the `text-embedding-3-small` embedding model from OpenAI, which is used as the default model in the main text. We report the mean $\pm$ standard error of mean (SEM) oracle objective value achieved by the single proposed design for a given patient, averaged over $n = 100$ target patients. **Bolded** (resp., Underlined) cells indicate the **best** (resp., second best) mean score in a column.

| Embedding Model | Warfarin $\downarrow$ RMSE Loss (*mg/week*) | HIV $\downarrow$ Viral Load (*copies/mL*) | Breast $\uparrow$ TTNTD (*months*) | Lung $\uparrow$ TTNTD (*months*) | ADR $\downarrow$ NLL Loss (*no units*) | Rank |
|---|---|---|---|---|---|---|
| Random | $2.28 \pm 0.22$ | $4.65 \pm 0.04$ | $54.54 \pm 2.65$ | $22.07 \pm 0.34$ | $19.0 \pm 1.5$ | 5.0 |
| Mistral-7B | $1.65 \pm 0.15$ | $4.53 \pm 0.04$ | $72.66 \pm 2.91$ | $28.02 \pm 0.33$ | $11.6 \pm 0.8$ | 3.2 |
| OpenAI-Small | $1.36 \pm 0.13$ | $4.50 \pm 0.04$ | $72.43 \pm 2.86$ | $32.71 \pm 0.32$ | $12.4 \pm 1.6$ | 2.8 |
| MedEmbed-Large | $1.44 \pm 0.23$ | $4.53 \pm 0.05$ | $68.44 \pm 3.16$ | $34.68 \pm 0.41$ | **$10.9 \pm 1.9$** | 2.6 |
| Bio+Clinical BERT | **$1.31 \pm 0.12$** | **$4.49 \pm 0.05$** | **$76.40 \pm 2.70$** | **$34.79 \pm 0.37$** | $11.2 \pm 1.7$ | **1.2** |

- Define $\mu$ (resp., $\sigma$) to be the mean (resp., standard deviation) of the source critic-weighted score $\hat{f}(x; z) + \lambda c^*(x)$ with respect to the source dataset $\mathcal{D}_{\text{src}}$. We can define a series of indexed thresholds $\{\tau_i\}_{i=1}^{11} = \{-\infty, \mu - 4\sigma, \mu - 3\sigma, \mu - 2\sigma, \mu - \sigma, \mu, \mu + \sigma, \mu + 2\sigma, \mu + 3\sigma, \mu + 4\sigma, +\infty\}$. Our **Score**-based equivalence relation then assigns an input design $x$ to equivalence class $i$ iff $\tau_i \leq \hat{f}(x; z) + \lambda c^*(x) < \tau_{i+1}$, and can therefore be thought of as an *outcome-aware* equivalence relation.

- Finally, the **Leiden** equivalence relation is based on the Leiden algorithm (Traag et al., 2019) for community detection, which is a technique that is commonly used certain fields of biological research Yu et al. (2022); Blair et al. (2022); Lin et al. (2022). To first construct the input graph $\mathcal{G}_0 := (\mathcal{V}_0, \mathcal{E}_0)$, we define $\mathcal{V}_0$ to be the set of all designs in the source dataset $\mathcal{D}_{\text{src}}$. We then embed these designs (represented in natural language) using the `text-embedding-3-small` model from OpenAI and keep the minimal number of principal components that capture at least 99% of the variance in $\mathcal{D}_{\text{src}}$. For each node $v \in \mathcal{V}_0$, we then find the $\max(10, \lfloor \log |\mathcal{V}_0| \rfloor)$ most similar nodes (according to their cosine similarity) in $\mathcal{V}_0 \setminus \{v\}$ and add undirected weighted edges connecting each nearest neighbor $v'$ to $v$ with weight $w = \exp[-d(v', v)^2 / 2(0.1^2)]$ to $\mathcal{E}_0$, where $d(\cdot, \cdot)$ is the cosine similarity. The number of equivalence classes is the number of discovered communities in the final graph $\mathcal{G}_0$ according to the Leiden algorithm. To assign a newly proposed design $x$ to an equivalence class, we follow a similar process as above to construct a new graph $\mathcal{G} = (\mathcal{V}, \mathcal{E})$, where $\mathcal{V}$ is the union of the new embedded design and $\mathcal{V}_0$, and $\mathcal{E}$ is the union of the edges from the new design and $\mathcal{E}_0$. Fixing the community memberships of the original nodes, we then repeat the Leiden algorithm to assign the newly embedded design to one of the existing communities in the graph $\mathcal{G}$. This process is performed separately for each newly proposed design; note that the number of distinct communities in $\mathcal{G}$ and $\mathcal{G}_0$ are equal.

Our experimental results using each of these equivalence relation definitions are shown in **Supplementary Figure S12**. As expected, the Random equivalence relation led to inferior optimization results on the **Lung** task when compared to the other implemented methods. Using the surrogate model-based scores in the Score-based equivalence relation offered a small improvement on this task, but did not meaningfully compete with both the $k$-Means and Leiden equivalence relations. Future work might evaluate other possible equivalence relations for specific applications of LEON.

### E.13 Equivalence Class Count Ablation

In our work, we compute the number of equivalence classes $N$ to use in LEON according to a task-specific heuristic. Namely, in our $k$-means vector clustering algorithm, we determine the number of clusters $k$ using the '**Elbow** method' after testing multiple possible values of $k$ for cosine similarity-based clustering of vector embeddings of designs from the source dataset represented in natural language. We then set $N = k$, such that each $k$-means cluster defines a unique equivalence class. However, it is also possible to deterministically set $N$ to a constant across all tasks and forego the use of the Elbow method in determining task-specific values of $N$ altogether.

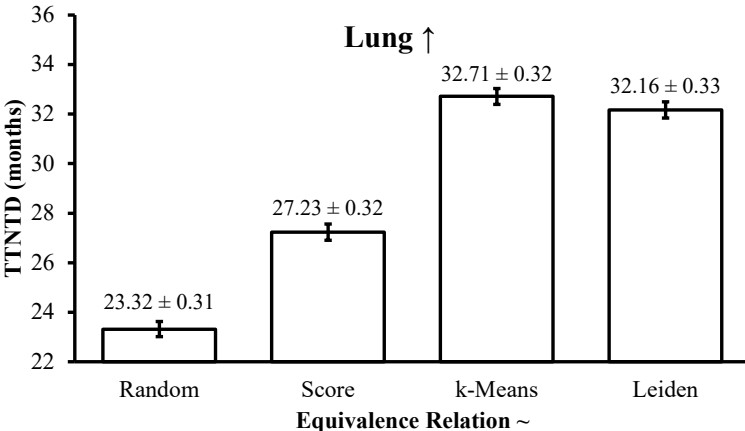

Figure S12: **Equivalence relation ablation.** We plot the mean $\pm$ SEM oracle score achieved when using each equivalence relation with LEON on the **Lung** task.

Table S10: **Equivalence class count ablation.** We ablate the data-driven computation of the number of equivalence classes $N$ based on the source dataset as detailed in **Appendix E.12**, and instead fix $N$ to a constant value. We report the mean $\pm$ standard error of mean (SEM) oracle objective value achieved by the single proposed design for a given patient, averaged over $n = 100$ target patients. **Bolded** (resp., Underlined) cells indicate the **best** (resp., second best) mean score for a given task.

| $N$ | Warfarin $\downarrow$ RMSE Loss (*mg/week*) | HIV $\downarrow$ Viral Load (*copies/mL*) | Breast $\uparrow$ TTNTD (*months*) | NSCLC $\uparrow$ TTNTD (*months*) | ADR $\downarrow$ NLL Loss (*no units*) | Rank |
|---|---|---|---|---|---|---|
| 2 | $3.24 \pm 0.68$ | $\mathbf{4.44 \pm 0.05}$ | $53.54 \pm 3.55$ | $27.33 \pm 0.17$ | $\mathbf{11.8 \pm 2.0}$ | 4.6 |
| 4 | $2.85 \pm 0.25$ | $4.48 \pm 0.04$ | $55.29 \pm 2.74$ | $35.85 \pm 0.21$ | $14.5 \pm 1.8$ | 4.8 |
| 8 | $\mathbf{1.29 \pm 0.22}$ | $4.49 \pm 0.04$ | $68.78 \pm 2.71$ | $34.75 \pm 0.28$ | $14.4 \pm 1.8$ | 3.4 |
| 16 | $2.29 \pm 0.48$ | $4.55 \pm 0.04$ | $\mathbf{77.77 \pm 2.95}$ | $37.38 \pm 0.32$ | $15.0 \pm 1.9$ | 3.8 |
| 32 | $2.79 \pm 0.45$ | $4.58 \pm 0.04$ | $71.94 \pm 2.66$ | $37.61 \pm 0.33$ | $14.4 \pm 1.8$ | 4.0 |
| 64 | $2.71 \pm 0.35$ | $4.57 \pm 0.04$ | $69.42 \pm 2.59$ | $\mathbf{38.94 \pm 0.48}$ | $14.4 \pm 1.8$ | 3.6 |
| Elbow | $1.36 \pm 0.13$ | $4.50 \pm 0.04$ | $72.43 \pm 2.86$ | $32.71 \pm 0.32$ | $12.4 \pm 1.6$ | $\mathbf{3.2}$ |

To this end, we ablated this $k$-means clustering heuristic and manually varied the number of equivalence classes between $N = 2$ and $N = 64$ in **Supplementary Table S10**. Our results suggest that across all 5 tasks, our Elbow method achieves the best of rank of **3.2**/7 across all 5 tasks. While individual values of $N$ may outperform the Elbow method on specific tasks, no fixed value of $N$ consistently outperformed our chosen approach across all tasks. Future work may explore alternative methods to determine the optimal number of equivalence classes to use for a given task.

## E.14 ITERATIVE FEEDBACK ABLATION

In **Supplementary Algorithm 1**, a key component of LEON is to iteratively leverage feedback from the (regularized) surrogate model to score proposed treatment designs, and use this information to update the language model's prior over the design space via in-context prompting. However, such an iterative loop can involve multiple queries to the LLM. A natural question is whether such an iterative feedback strategy is even needed: that is, can a language model reason about a patient $z$ and domain knowledge constructed in **Supplementary Algorithm 2** to propose a single treatment design in a zero-shot manner *without ever using the surrogate model at all*? To this end, we evaluated this 'Noniterative' strategy against our 'Iterative' method LEON in **Supplementary Table S11**.

Our results suggest that iterative optimization significantly improves the performance of LEON across all tasks evaluated. This highlights the utility of iteratively using feedback from the surrogate model and solving the constrained optimization problem in (4) using LEON.

Table S11: **Iterative feedback ablation.** We ablate the iterative feedback loop in **Supplementary Algorithm 1** by querying the backbone LLM to propose a single treatment strategy conditioned solely on the patient's individual knowledge and its knowledge generated using **Supplementary Algorithm 2**. We report the mean $\pm$ standard error of mean (SEM) oracle objective value achieved by the single proposed design for a given patient, averaged over $n = 100$ target patients.

| | **Warfarin**
↓ RMSE Loss
(*mg/week*) | **HIV**
↓ Viral Load
(*copies/mL*) | **Breast**
↑ TTNTD
(*months*) | **Lung**
↑ TTNTD
(*months*) | **ADR**
↓ NLL Loss
(*no units*) |
|---|---|---|---|---|---|
| Noniterative | $1.83 \pm 0.69$ | $4.56 \pm 0.07$ | $29.35 \pm 4.98$ | $20.49 \pm 0.65$ | $21.5 \pm 0.5$ |
| Iterative | $\mathbf{1.36 \pm 0.13}$ | $\mathbf{4.50 \pm 0.04}$ | $\mathbf{72.43 \pm 2.86}$ | $\mathbf{32.71 \pm 0.32}$ | $\mathbf{12.4 \pm 1.6}$ |

