# OpenReview forum: "Knowledgeable Language Models as Black-Box Optimizers for Personalized Medicine"
_ICLR.cc/2026/Conference — ICLR 2026 Poster_

### Official Review · Reviewer_qPrg · 2025-10-31

**Soundness:** 3
**Presentation:** 3
**Contribution:** 3
**Rating:** 6
**Confidence:** 3

**Summary:**

This paper introduces LEON, an LLM based entropy guided optimizer for conditional black box optimization in personalized medicine. The method steers an LLM with two principled constraints, a Wasserstein bound enforced by a source critic to limit shift from the training distribution and a coarse entropy control defined over equivalence classes of candidate treatments. The algorithm alternates proposing treatments, scoring them with a learned surrogate that is regularized by the critic, and updating dual variables for the two constraints. Prior medical knowledge is injected through tool use over textbooks, knowledge graphs, and a specialist model, and the system caches reflections across iterations. Experiments on five tasks under explicit distribution shift report consistent improvements over classical baselines such as Bayesian optimization and gradient methods as well as OPRO style prompting, with ablations on equivalence relations, embeddings, budgets, temperature, and knowledge quality. The writing is clear overall and the approach is theoretically motivated, though some key engineering details are pushed to the appendix and several evaluations rely on learned or simulated outcome functions.

**Strengths:**

1. A principled formulation of patient level decision-making as constrained conditional optimization that combines a Wasserstein distribution shift control with an entropy based exploration control, yielding a practical and interpretable loop.
2. Broad empirical evaluation across multiple biomedical tasks with ablations that illuminate sensitivity to embeddings, equivalence relations, budget, temperature, and knowledge quality.
3. Thoughtful use of knowledge tools and reflection rather than vague claims, with explicit tests of helpful versus irrelevant knowledge.
4. Clear problem setup, readable pseudocode, and helpful diagrams that separate the roles of the critic, the entropy control, and the knowledge components.
5. Responsible positioning with discussion of clinical caveats and an intention to release code for public tasks.

**Weaknesses:**

1. Several evaluations use a learned oracle for outcomes rather than real clinical endpoints, which risks baking evaluation bias into the target function and limits external validity.
2. The choice of equivalence relation and embedding strongly affects entropy estimates and search behavior, yet selection guidance is limited and some details are deferred to the appendix.
3. The source critic training and dual updates introduce stability and tuning concerns, and the paper does not fully characterize sensitivity to clipping, step ratios, or learning rates.
4. The pipeline is complex, with interdependent modules for prompting, clustering, critic training, tool access, and dual updates. This challenges reproducibility and may require significant engineering effort.
5. Computational cost is likely high due to repeated LLM calls, clustering, and critic retraining. Wall clock and token cost reporting is incomplete for the main text.
6. Presentation is dense in places and certain implementation choices for knowledge tool wiring and safety controls are difficult to locate in the main body.
7. Baseline tuning details and budget parity need to be fully specified to rule out configuration gaps, especially for OPRO style prompts and human or gradient baselines.
8. Practical safety and deployment are not validated with clinician in the loop studies. The method can degrade with irrelevant or adversarial knowledge and the current safeguards appear limited.
9. The paper is difficult to understand and follow.
10. A lot of materials and supporting evidence for reproducibility, yet no working link to the code.

**Questions:**

1. For each task, what model families were used to construct the evaluation oracle, and how do conclusions change under alternate architectures or calibrated versions of the oracle?
2. Can you report performance as a function of shift magnitude and describe failure modes when the target distribution materially departs from the source?
3. How is the constraint level chosen in practice? Provide a principled rule or adaptive schedule and an ablation in the main text.
4. Did you try gradient penalty or spectral normalization rather than weight clipping?
5. How many samples per batch are required for a stable estimate of the entropy multiplier and how does its variance depend on temperature and batch size?
6. Beyond k-means and Leiden, can the equivalence relation be learned end-to-end with contrastive or outcome-aware objectives?
7. Precisely how are textbooks, graphs, and the specialist model connected to the LLM context, for example through retrieval, tool calling, or structured APIs?
8. Can you add pre-filters based on citation consistency or entailment and report their effect on robustness?
9. Can you report token counts, wall clock times, and GPU hours per task?
10. Can you publish the exact prompt templates, budgets, and tuning for OPRO style and human baselines, and confirm identical oracle query budgets across methods?
11. How does performance change with smaller or open LLMs, and what minimum capability is required to retain the observed gains?
12. How is patient safety enforced when suggestions deviate from standard practice, including any hard constraints, verification, or human in the loop controls?
13. Can you provide a minimal working example with synthetic data that exercises all components including critic updates, clustering, and knowledge tools, along with seed handling and version pins?

---

> ### Author Response · Authors · 2025-11-18
>
> We thank Reviewer qPrg for their thoughtful review and insightful comments, and are happy that they found our problem setup and pseudocode clear and readable. Please find our responses to their numbered Weaknesses (W) and Questions (Q) below. Please let us know if there are any additional follow-up questions, and we would be happy to discuss further with the Reviewer. Thank you!
>
> - **(W1, Q1) Oracle Evaluation Strategy**: We agree with the Reviewer that using learned oracle functions is not optimal, as we discuss in the Limitations subsection of Section 6 of the main text. Nonetheless, we evaluate proposed treatment designs using a learned oracle function because it is not possible to test newly proposed treatment designs by administering them to arbitrary patients in the real-world. Our approach of using a learned oracle function is consistent with prior work both in clinical applications and other scientific domains ([Trabucco et al. ICML 2022](https://proceedings.mlr.press/v162/trabucco22a.html), [Yao et al. NeurIPS 2024](https://openreview.net/forum?id=3RxcarQFRn)). We also evaluate on multiple tasks with different oracle model families (see Table S1 on page 29), and also ensure that the oracle functions are well-calibrated with the target domain observations at a minimum (see Figures S1 and S2 on pages 38 and 39, respectively).
> - **(W2, Q6) Equivalence Relation and Embedding**: We evaluate a suite of different equivalence relations in Table S9 (page 60) and Figure S12 (page 61); our results demonstrate that by adjusting the choice of equivalence relation, it is certainly possible to further boost the performance of LEON for specific applications. Our modular implementation of LEON allows for trialing different options for suitable equivalence relations; while we were unable to learn an end-to-end equivalence relation due to limited data availability in this work, such an option could certainly be explored in future work.
> - **(W3, Q4) Source Critic Implementation Strategy**: We enforce the Lipschitz constraint via $\ell_\infty$ clipping consistent with prior work ([Arjovsky et al. ICML 2017](https://proceedings.mlr.press/v70/arjovsky17a.html), [Yao et al. NeurIPS 2024](https://openreview.net/forum?id=3RxcarQFRn), [Yao et al. ICML 2025](https://proceedings.mlr.press/v267/yao25b.html)), which have demonstrated that this method is both viable in practice and an easy and efficient in enforcing the constraint. While other, more sophisticated methods for enforcing the constraint exist, we emphasize that the goal of our work is to propose a generalizable, extensible method for black-box optimization and personalized medicine. In general, determining the Lipschitz constant of neural networks is NP-hard and still an open problem ([Abdeen et al. Preprint 2025](https://arxiv.org/abs/2506.23977)). While one may substitute alternative, more sophisticated methods for computing the Lipschitz constant of neural networks as a drop-in replacement for our chosen method, we leave this as an opportunity for future work and have updated Section 5 in the main text (updates in *teal*) clarifying this design choice.
> - **(W4, W10, Q13) Pipeline Implementation and Code Availability**: Regarding code availability, we note that our code implementation is made publicly available in Supplemental Material ZIP file on OpenReview. We will also publish a link to a GitHub repository with this code after the review period; the link is not made available in the current manuscript to respect the double-blind peer review process. Finally, we have updated our code to include (1) a `notebooks/tutorial.ipynb` notebook that walks through a minimal working example with synthetic data; and (2) an API `leon.optimize.maximize()` modelled off of the [`scipy.optimize.minimize()`](https://docs.scipy.org/doc/scipy/reference/generated/scipy.optimize.minimize.html) function call that will hopefully provide a familiar framework to work with LEON. Please see the updated Supplemental Material for these code changes.
> - **(W5, Q9) Computational Cost of LEON**: We refer the Reviewer to Appendix D.2 and Figure S4 on page 40, where we analyze the cost associated with LEON from monetary, token usage, and environmental impact perspectives.
> - **(W6, W9) Paper Presentation**: We apologize if there are instances where manuscript material is dense. We have included a detailed Table of Contents on page 25 to facilitate navigation of the Appendix, and have made significant efforts to clarify the language and improve its accessibility in our updated version of the manuscript. If there are any particular sections of the paper that would warrant additional revision, please do let us know and we would be happy to discuss further.

---

> > ### Author Response · Authors · 2025-11-18
> >
> > - **(W7, Q10) Baseline Tuning Details**: We confirm that all methods had an identical oracle query budget in Table 1; please see the Evaluation Metric subsection of Section 5 in our main text for additional details. We use the same hyperparameters for OPRO and Eureka as specified by the respective baseline authors in their corresponding manuscripts, except for extending the sampling budget of both methods to match that of LEON to ensure a fair comparison. Our prompts for the OPRO and Eureka baselines are included in our Supplementary Material ZIP file; see the `src/leon/optim/prompts/user` directory for the raw text prompts.
> > - **(W8, Q12) Clinician-in-the-Loop Studies and Patient Safety**: We agree with the Reviewer that patient safety is critical, and that any real-world deployment of LEON in clinical practice would necessitate rigorous evaluation with clinicians in the loop. In our Ethics Statement at the end of our main text, we explicitly state that **LEON is not intended for direct clinical use**, but rather as a *methodological contribution* toward future systems that may aid clinical decision making. For this reason and resource limitations, rigorous clinician-in-the-loop studies for particular applications of LEON are outside the scope of this work. We emphasize that any real-world deployment would require extensive safeguards, including (but not limited to) careful physician oversight and validation of any treatment strategy proposed by LEON.
> > - **(Q2) Ablating the Magnitude of Distribution Shift**: We refer the Reviewer to Appendix E.6 and Figure S9 on page 54, which evaluates LEON as a function of the magnitude of the distribution shift between source and target distributions.
> > - **(Q3) Constraint Level Ablation**: We refer the Reviewer to Figure S8 (right panel) on page 52, which shows how the constraint hyperparameter $W_0$ affects the performance of LEON on a representative task. Importantly, we use the same hyperparameter ($W_0=1.0$) across all 5 tasks in the main text to avoid overfitting it to any particular task. Please see the Experiment Implementation subsection of Section 5 in our main text, and Appendix E.4 for additional discussion.
> > - **(Q5) Ablating LEON Hyperparameters**: We refer the Reviewer to Figure S8 (left and middle panels) on page 52, which shows how the temperature and sampling batch size hyperparameters affect the performance of LEON on a representative task. Importantly, we use the same set of hyperparameters (temperature $\tau=1.0$ and sampling batch size of 32) across all 5 tasks in the main text to avoid overfitting these variables to any particular one task. Please see the Experiment Implementation subsection of Section 5 in our main text, and Appendices E.2 and E.3 for additional discussion.
> > - **(Q7) Knowledge as a Tool Calling Problem**: The Reviewer's assessment is correct: we formulate knowledge synthesis as a tool-calling problem in our implementation. We refer the Reviewer to Algorithm 2 on page 33 and Appendix D.3 on page 38 for additional results and discussion.
> > - **(Q8) The Impact of Knowledge Quality on LEON**: From our results in Appendix E.9 and Table S7 on page 59, we found that LLM optimizers using LEON can indeed be sensitive to the quality of knowledge sources provided. Furthermore, Figure S5 on page 41 and the discussion in Appendix D.3 demonstrate that LLMs indeed preferentially select relevant knowledge sources to query for knowledge generation. Given these collective results, it is likely that using pre-filters based on citation consistency, internal proprietary knowledge sources, and/or specially curated knowledge from domain experts may further improve the performance of our proposed method. We highlight this in our revised Appendix E.9 (changes in *orange*-colored text) as a way to further optimize LEON for specific tasks and applications.
> > - **(Q11) Ablating the Backbone LLM**: We refer the Reviewer to Appendix E.1 and Table S5 on page 50, where we compare the performance of using different LLMs with LEON. Briefly, we found that closed-source reasoning models, such as o4-Mini and Gemini-2.5 Flash, achieved the best performance when compared to other LLMs (such as GPT-4o Mini reported in Table 1 in the main text).

---

> > > ### Author Response · Authors · 2025-11-26
> > >
> > > Hi, hope you're doing well! Thank you again for your feedback and consideration of our manuscript. We hope that our comments and corresponding revisions made in our updated manuscript have helped address your initial concerns. We wanted to ask if there any remaining questions that we can help address during this discussion period? Thank you!

---

### Official Review · Reviewer_cRy4 · 2025-11-01

**Soundness:** 3
**Presentation:** 3
**Contribution:** 3
**Rating:** 6
**Confidence:** 2

**Summary:**

This paper introduces LEON (LLM-based Entropy-guided Optimization with kNowledgeable priors), a framework that leverages large language models as conditional black-box optimizers for personalized medicine under distribution shift. The authors formulate personalized treatment design as a conditional optimization problem, where the LLM iteratively generates candidate treatment strategies that are evaluated and refined using the LEON objective across multiple rounds. Evaluated on five real-world personalized medicine tasks, LEON consistently outperforms both traditional and LLM-based baselines, achieving an average rank of 1.2 without any model fine-tuning.

**Strengths:**

- The work addresses an important and emerging problem—how to leverage LLMs for safe, knowledge-guided optimization in personalized medicine under distribution shift.

- LEON consistently outperforms both traditional and LLM-based baselines across multiple real-world personalized medicine tasks.

- The paper is well-structured and easy to follow, with the key ideas and methodological steps clearly highlighted.

**Weaknesses:**

See questions.

**Questions:**

1. How does the computational and monetary cost of LEON scale with the length of the LLM context? Since LEON relies on iterative prompting and maintains a growing history of prior designs and knowledge within the prompt, long contexts could substantially increase inference cost.

2. The paper does not clearly specify the modality of the treatment design. Are treatment regimens represented as structured variables (e.g., dosage vectors, categorical drug combinations) or as natural-language descriptions? Could the authors elaborate on how treatment designs are represented, constrained, and parsed across the five experimental tasks, and whether different modalities were handled uniformly?

3. How does the LLM perform without the optimization procedure (i.e., when generating treatment designs directly from prior knowledge without iterative scoring and refinement)? Could the authors include an ablation comparing this zero-iteration baseline to the full LEON process? Additionally, it would be helpful to visualize performance as a function of iteration number to illustrate convergence behavior and the marginal gains achieved through iterative optimization.

---

> ### Author Response · Authors · 2025-11-18
>
> We thank Reviewer cRy4 for their thoughtful review of our work, and are grateful that they found our work interesting and easy to follow. Please find our responses below - we would be happy to discuss further and answer any follow-up questions as necessary. Thank you!
>
> 1. **Cost of LEON**: We refer the Reviewer to Appendix D.2 and Figure S4 on page 40, where we analyze the cost associated with LEON from monetary, token usage, and environmental impact perspectives.
> 2. **Modality of Treatment Designs**: We refer the Reviewer to Appendix C.1 and Table S1 on page 29 for detailed descriptions regarding the modality of the treatment designs by task. This includes both a description of the design space $\mathcal{X}$ and each of the design dimensions. Furthermore, Appendix C.3 includes descriptions of how we represent each design in natural language as input into the LLM (specifically, please see the sample `patient_description` and `memory` entries provided in our manuscript).
> 3. **Zero-Iteration Ablation Study**: Thank you for this suggestion. We compare LEON with the LLM without iterative optimization below, and have also included these results in Appendix E.14 and Table S11 on page 61 (in *bright purple*-colored text) in our updated manuscript.
>
> | **Method** | **Warfarin** $\downarrow$ | **HIV** $\downarrow$ | **Breast** $\uparrow$ | **NSCLC** $\uparrow$ | **ADR** $\downarrow$ |
> | ---: | :---: | :---: | :---: | :---: | :---: |
> | Noniterative | 1.83 $\pm$ 0.69 | 4.56 $\pm$ 0.07 | 29.35 $\pm$ 4.98 | 20.49 $\pm$ 0.65 | 21.5 $\pm$ 0.5 |
> | LEON | **1.36 $\pm$ 0.13** | **4.50 $\pm$ 0.04** | **72.43 $\pm$ 2.86** | **32.71 $\pm$ 0.32** | **12.4 $\pm$ 1.6** |
>
> Our results suggest that iterative optimization significantly improves the performance of LEON across all tasks evaluated, and are consistent with the idea that iteratively adapting to observed data points and taking advantage of available surrogate models improves the performance of LEON. We also refer the Reviewer to Figure S10 on page 55 for detailed results on how the performance of LEON changes as a function of the iteration number on the Warfarin task as a representative example. We note that LEON outperforms other baselines even when restricting to fewer iterations in Figure S10.

---

> > ### Author Response · Authors · 2025-11-26
> >
> > Hi, hope you're doing well! Thank you again for your thoughtful review of our work. We hope that our additional results (included in our revised manuscript) and implementation details have helped address your questions. We wanted to ask if there are any remaining questions we can help address?

---

> > > ### Comment · Reviewer_cRy4 · 2025-11-27
> > > **Update**
> > >
> > > Thank you for the clear and helpful responses. The added analyses and clarifications addressed my questions well, and I appreciate that you incorporated them into the revised manuscript. After considering the updates and other reviewers’ comments, I have increased my score to 8.

---

> > > > ### Author Response · Authors · 2025-11-27
> > > >
> > > > We appreciate your helpful comments and support of our work. Thank you!

---

### Official Review · Reviewer_iXHF · 2025-11-03

**Soundness:** 3
**Presentation:** 3
**Contribution:** 3
**Rating:** 4
**Confidence:** 3

**Summary:**

The authors propose LEON, a framework that formulates personalized medicine as a conditional black-box optimization problem guided by two constraints: a Wasserstein distance bound to reduce out-of-distribution risk, and a coarse-grained entropy regularization to promote deterministic sampling across semantically similar treatments. LEON iteratively prompts an LLM to generate treatments using patient context and external medical knowledge, updating dual variables to balance exploration and prior alignment. Experiments on five clinical tasks demonstrate strong performance across multiple baselines.

**Strengths:**

- Introduces learnable determinism via entropy and Wasserstein constraints, moving beyond static penalties or temperature tuning.
- Lemma 4.2 simplifies the constrained optimization problem, making implementation tractable. The dual gradient update for lambda is intuitive.
- The algorithmic steps are well explained, including the role of prior knowledge and reflective prompting.
- Outperforms strong baselines on several clinical proxy tasks without fine-tuning, highlighting the promise of uncertainty-aware, knowledge-driven generation.

**Weaknesses:**

1. Clustering is heuristic (embedding + k-means), with no expert verification or sensitivity study on class count/distance metrics. Efficiently finding within-class optima $x_i^*$ in high-dimensional/discrete spaces is not thoroughly discussed.

2.  Lipschitz constraint via $\ell_\infty \leq 0.01$ clipping may reduce capacity or cause instability; no comparison to spectral norms or gradient penalties.  Assumes access to source domain $D_{\text{src}}$, limiting applicability in private or black-box settings; no fallback strategy is provided.

3. Some data (e.g., cancer/ADR) are proprietary; no prospective or human-in-the-loop evaluations are conducted.  "Human baseline" may overestimate human quality by ignoring real-world constraints (drug access, comorbidities, etc.).

4. Fairness and safety not evaluated: No subgroup analysis (e.g., gender/race) on Warfarin/HIV tasks. No safeguards against unsafe treatment suggestions or bias propagation.

**Questions:**

- How is class count $N$ chosen? Were domain-specific structures (e.g., drug targets) considered? Any expert annotations to validate class quality?
- Can you provide $\mu_t, \lambda_t$ trajectories per task and analyze their link to performance? Any divergence observed?
- Why use $\ell_\infty$ clipping over other Lipschitz enforcement methods? Might $c^*$ wrongly reject rare but valid samples?
- How robust is $\mu$ to outdated/noisy/conflicting knowledge? Any causal evidence that knowledge improves determinism?
- Did all baselines (e.g., OPRO, Eureka, LLAMBO) have equal access to external knowledge? If not, is the comparison fair?
- Does the final treatment recommendation account for toxicity, availability, or interactions? Can the method accommodate such constraints?

---

> ### Author Response · Authors · 2025-11-18
>
> We thank Reviewer iXHF for their thoughtful review and comments, which we believe have significantly improved the quality of our manuscript. Please find our answers to your questions and concerns below. If you have any remaining questions, we would be happy to answer and discuss further. Thank you!
>
> 1. **Clustering Algorithm and Choice of Class Count $N$**: Because our goal is to introduce a general-purpose algorithm for black-box optimization in precision medicine, we intentionally chose a clustering algorithm that could be generalizable across different tasks with minimal task-specific adaptations. We determine the number of classes $N$ for each task using automated 'Elbow method' approximation, where $N$ is set to be the integer number of clusters that maximizes the distance from a straight line connecting the points $N = 2$ and $N = 20$ on a graph plotting the within-cluster sum of squares (WCSS) versus the number of clusters $N$. Additional details regarding this method are described in Appendix E.12 (lines 3182-3203).
>
> Separately, we show in Table S9 on page 60 that using more specialized embedding models can further boost the performance of LEON, and evaluate alternative equivalence relations in Figure S12 on page 61. We also evaluate how using a fixed number of equivalence classes across tasks changes the performance of LEON in Table S10 on page 61 (see *teal* text in the updated manuscript), included below for easy reference:
>
> | $N$ | **Warfarin** $\downarrow$ | **HIV** $\downarrow$ | **Breast** $\uparrow$ | **NSCLC** $\uparrow$ | **ADR** $\downarrow$ | **Rank** |
> | ---: | :---: | :---: | :---: | :---: | :---: | :---: |
> | 2 | 3.24 $\pm$ 0.68 | **4.44 $\pm$ 0.05** | 53.54 $\pm$ 3.55 | 27.33 $\pm$ 0.17 | **11.8 $\pm$ 2.0** | 4.6 |
> | 4 | 2.85 $\pm$ 0.25 | *4.48 $\pm$ 0.04* | 55.29 $\pm$ 2.74 | 35.85 $\pm$ 0.21 | 14.5 $\pm$ 1.8 | 4.8 |
> | 8 | **1.29 $\pm$ 0.22** | 4.49 $\pm$ 0.04 | 68.78 $\pm$ 2.71 | 34.75 $\pm$ 0.28 | 14.4 $\pm$ 1.8 | *3.4* |
> | 16 | 2.29 $\pm$ 0.48 | 4.55 $\pm$ 0.04 | **77.77 $\pm$ 2.95** | 37.38 $\pm$ 0.32 | 15.0 $\pm$ 1.9 | 3.8 |
> | 32 | 2.79 $\pm$ 0.45 | 4.58 $\pm$ 0.04 | 71.94 $\pm$ 2.66 | *37.61 $\pm$ 0.33* | 14.4 $\pm$ 1.8 | 4.0 |
> | 64 | 2.71 $\pm$ 0.35 | 4.57 $\pm$ 0.04 | 69.42 $\pm$ 2.59 | **38.94 $\pm$ 0.48** | 14.4 $\pm$ 1.8 | 3.6 |
> | Elbow | *1.36 $\pm$ 0.13* | 4.50 $\pm$ 0.04 | *72.43 $\pm$ 2.86* | 32.71 $\pm$ 0.32 | *12.4 $\pm$ 1.6* | **3.2** |
>
> Our Elbow method approach described above outperforms fixing the number of equivalence classes to a particular number between $N=2$ and $N=64$ across all 5 tasks.
>
> 2. **Enforcing Lipschitz Constraint via $\ell_\infty$ Clipping**: We enforce the Lipschitz constraint via $\ell_\infty$ clipping consistent with prior work ([Arjovsky et al. ICML 2017](https://proceedings.mlr.press/v70/arjovsky17a.html), [Yao et al. NeurIPS 2024](https://openreview.net/forum?id=3RxcarQFRn), [Yao et al. ICML 2025](https://proceedings.mlr.press/v267/yao25b.html)), which have demonstrated that this method is both viable in practice and an easy and efficient way to enforce the constraint. While other, more sophisticated methods for enforcing the constraint exist, we emphasize that the goal of our work is to propose a generalizable, extensible method for black-box optimization and personalized medicine. In general, determining the Lipschitz constant of neural networks is NP-hard and still an open problem ([Abdeen et al. Preprint 2025](https://arxiv.org/abs/2506.23977)). While one may substitute alternative, more sophisticated methods for computing the Lipschitz constant of neural networks as a drop-in replacement for our chosen method, we leave this as an opportunity for future work and have updated Section 5 in the main text (updates in *teal*) clarifying this design choice.
>
> 3. **Does the source critic require access to the source domain?**: We refer the Reviewer to lines 223-5 on page 5 of our manuscript. We intentionally restricted the domain of $c^*$ to $\mathcal{X}$, meaning that **no individual patient observations $z$ from the source domain are required**. We only need access to the treatments that source patients received, *not* the identities of the patients themselves nor which patient received what treatment. **This design choice intentionally preserves patient privacy**, and assuming access to treatments is reasonable even in the black-box setting assumed in our work. Alternative options may be to avoid the use of the source-critic altogether in LEON (albeit at the expense of performance according to our ablation experiments in Table S6 on page 53) or to derive and enforce a prior from scientific literature.

---

> > ### Author Response · Authors · 2025-11-18
> >
> > 4. **Might $c^*$ wrongly reject rare but valid samples?** Yes, this is correct. This "conservatism" reflected in $c^*$ is intentional because we assume giving a patient a bad, life-threatening treatment is worse than missing out on a potentially efficacious treatment. However, because the constraints in Equation (4) are in expectation, a rare treatment option may still be proposed by LEON if (1) the evidence from knowledge sources suggests that a rare treatment option is very likely to work; and (2) if the predicted surrogate score associated with that treatment option is sufficiently high.
> > 5. **Evaluation Datasets**: We intentionally evaluated LEON on a mix of both publicly available datasets (Warfarin and HIV) and proprietary datasets (Breast, Lung, and ADR). In contrast to the Warfarin and HIV tasks, which make use of older (albeit well-studied) datasets, the Breast, Lung, and ADR tasks represent current, modern applications of personalized medicine that are of interest to the authors' institution and the broader clinical and scientific communities. Together with open-sourcing our code (see the Supplementary Material ZIP file), we hope that including this mix strikes a balance between demonstrating LEON's real-world relevance and ensuring reproducibility and extensibility of our method.
> > 6. **Human Baseline Clarification**: To clarify the "Human" baseline in Table 1, this baseline evaluates the actual treatment *in real life* that a target patient actually received according to the hidden annotated target dataset. In this sense, it is reasonable to assume real-world constraints, such as drug access and comorbidities, are explicitly taken into account by the human clinician when they were treating the target patient that contributed to the annotated target dataset. We refer the Reviewer to the Baselines subsection in Section 5 of our main text for additional details.
> > 7. **Fairness and Safety Evaluation**: We agree with the Reviewer that a rigorous evaluation of the fairness and safety of LEON is paramount prior to any real-world deployment of LEON for clinical applications. In our Ethics Statement, we explicitly emphasize this point to readership, and state that **LEON is not intended for direct clinical use**, but rather as a *methodological contribution* toward future systems that may aid clinical decision making. Nonetheless, we conducted subgroup fairness analyses of LEON with respect to self-reported patient gender and race; please see Appendix D.10 on page 46 in *teal* in our updated manuscript for details. We also include the results below for convenience:
> >
> > | **Gender Analysis** | **Breast** $\uparrow$ | **Lung** $\uparrow$ | **ADR** $\downarrow$ |
> > | ---: | :---: | :---: | :---: |
> > | Male | 89.27 $\pm$ 8.72 | 32.78 $\pm$ 0.50 | 12.9 $\pm$ 2.0 |
> > | Female | 71.54 $\pm$ 2.97 | 32.66 $\pm$ 0.42 | 11.3 $\pm$ 2.7 |
> >
> > | **Race Analysis** | **Breast** $\uparrow$ | **Lung** $\uparrow$ |
> > | ---: | :---: | :---: |
> > | White | 73.88 $\pm$ 3.19 | 32.51 $\pm$ 0.43 |
> > | Non-White | 69.61 $\pm$ 5.69 | 33.10 $\pm$ 0.45 |
> >
> > We found that there was no statistically significant difference in the performance of LEON with respect to either of these attributes. We emphasize that any real-world deployment would require extensive safeguards, including (but not limited to) careful physician oversight and validation of any treatment strategy proposed by LEON.
> >
> > 8. **$\mu_t$ and $\lambda_t$ Trajectories**: We refer the Reviewer to Figure S6 on page 43 for $\mu_t$ trajectories by task; please see page 43 for discussion and interpretation. We have also updated our manuscript to include $\lambda_t$ trajectories by task in Figure S7 on page 44; please see Appendix D.6 in *teal* color for discussion and interpretation.
> > 9. **Robustness of LEON to Outdated/Noisy/Conflicting Knowledge**: Thank you for this question. We refer the Reviewer to Table S7 on page 59 of our manuscript, where we investigate how the quality of knowledge affects the performance of LEON. Our main takeaway is that LLM optimizers using LEON can be sensitive to the quality of knowledge sources provided, consistent with prior work ([Omar et al. Nat Commun Med 2025](https://www.nature.com/articles/s43856-025-01021-3), [Han et al. npj Digit Med 2024](https://www.nature.com/articles/s41746-024-01282-7), [Alber et al. Nat Med 2025](https://www.nature.com/articles/s41591-024-03445-1)). Future work may investigate how the performance of LEON is affected over longer periods of time due to information that slowly becomes outdated over time.

---

> ### Author Response · Authors · 2025-11-18
>
> 10. **Baseline Access to External Knowledge**: The baseline methods evaluated in our work do not explicitly define a way to integrate their method with tool-calling to access external knowledge, and there is no method that is immediately obvious to do this. For these reasons, the baseline methods in Table 1 of the main text did not have access to external knowledge sources in order to remain faithful to the authors' original implementations.
>
> However, for each of LLAMBO, OPRO, and Eureka, we can still first run Algorithm 2 using the backbone LLM (exactly as done in LEON) and append the generated knowledge to the string task context used in the respective user prompts in each of these three methods. We compare each of these methods both with and without the generated knowledge in Table S4 on page 48 of Appendix D.11 (copied below for easy reference):
>
> | **Method** | **Knowledge** | **Warfarin** $\downarrow$ | **HIV** $\downarrow$ | **Breast** $\uparrow$ | **NSCLC** $\uparrow$ | **ADR** $\downarrow$ | **Rank** $\downarrow$ |
> | ---: | :---: | :---: | :---: | :---: | :---: | :---: | :---: |
> | LLAMBO | ❌ | 3.28 $\pm$ 0.10 | 4.52 $\pm$ 0.05 | 48.83 $\pm$ 2.48 | 20.60 $\pm$ 0.31 | 20.6 $\pm$ 1.9 | 6.4 |
> | LLAMBO | ✅ | 2.70 $\pm$ 0.53 | 4.59 $\pm$ 0.04 | 43.18 $\pm$ 2.11 | 27.78 $\pm$ 0.35 | 25.1 $\pm$ 1.6 | 7.0 |
> | OPRO | ❌ | *1.40 $\pm$ 0.13* | 4.55 $\pm$ 0.04 | 55.68 $\pm$ 2.86 | 24.35 $\pm$ 0.43 | 23.8 $\pm$ 1.7 | 5.4 |
> | OPRO | ✅ | 1.87 $\pm$ 0.22 | *4.50 $\pm$ 0.06* | *72.34 $\pm$ 2.45* | *38.90 $\pm$ 0.44* | 14.4 $\pm$ 1.8 | 2.8 |
> | Eureka | ❌ | 1.54 $\pm$ 0.25 | 4.58 $\pm$ 0.04 | 63.48 $\pm$ 3.52 | 25.10 $\pm$ 0.69 | 21.3 $\pm$ 2.0 | 5.0 |
> | Eureka | ✅ | 1.81 $\pm$ 0.17 | **4.48 $\pm$ 0.04** | 71.73 $\pm$ 2.54 | **38.93 $\pm$ 0.38** | 14.4 $\pm$ 1.8 | *2.4* |
> | LEON | ❌ | 2.16 $\pm$ 0.19 | 4.57 $\pm$ 0.09 | 55.90 $\pm$ 2.43 | 24.64 $\pm$ 0.76 | *13.4 $\pm$ 2.9* | 5.0 |
> | LEON | ✅ | **1.36 $\pm$ 0.13** | *4.50 $\pm$ 0.04* | **72.43 $\pm$ 2.86** | 32.71 $\pm$ 0.32 | **12.4 $\pm$ 1.6** | **1.8** |
>
> These results in Appendix D.11 (see *teal* colored-text in the updated manuscript) suggest that knowledge alone is not sufficient to close the performance gap between LEON and other LLM-based optimization methods evaluated herein.
>
> 11. **Extending to Multiobjective Optimization**: We do not explicitly optimize over toxicity or drug-drug interactions, such constraints are implicitly captured by the learned surrogate model (trained on real patient trajectories) and integrated external knowledge sources. Future work might explore explicitly incorporating toxicity, availability, or interaction profiles into LEON as a *multi-objective* optimization task. Standard multi-objective optimization techniques include scalarizing the objectives using a weighting vector or enforcing safety/availability as constraints. Both methods are directly compatible with LEON's optimization framework - see Appendix D.12 on page 48 for an in-depth discussion (in *teal*-colored text).

---

> > ### Author Response · Authors · 2025-11-26
> >
> > Hi, hope you're doing well! Thank you again for your helpful feedback and review of our manuscript. We hope that (1) the clarification on our source critic in LEON; and (2) our additional experimental results (included in our revised manuscript) have helped address your initial concerns. We wanted to check if there are any remaining questions we can help address during this discussion period?

---

### Official Review · Reviewer_6DUi · 2025-11-05

**Soundness:** 3
**Presentation:** 4
**Contribution:** 3
**Rating:** 8
**Confidence:** 3

**Summary:**

The paper proposes using LLM as a black box optimizer to propose stochastic treatment designs in the context of personalized medicine. The paper experiments on real-world optimization tasks to show that the method outperforms other existing methods.

**Strengths:**

I like the paper a lot. Even though I am not quite familiar with the relevant literature, it is a pleasure to read the paper and learn what it is trying to convey.

Originality: the use of LLM as black box in solving optimization problems in personalized medicine is quite new.
Quality: The paper has good theoretical derivations as well as real-world examples.
Clarity: It is easy to follow for someone like me who does not know relevant literature.

**Weaknesses:**

Some minor weaknesses:

1. The idea of using LLM as a black box tool in optimization is studied before. The being said, the paper is still studying a quite original and significant problem.
2. Some of the really interesting experiments are in the appendix.

**Questions:**

None

---

> ### Author Response · Authors · 2025-11-18
>
> We thank Reviewer 6DUi for their thoughtful review of our work, and are grateful that they found our work interesting and easy to follow. Please find our responses below - we would be happy to discuss further and answer any follow-up questions. Thank you!
>
> 1. **Prior work on LLMs as black-box optimizers**: We agree with the Reviewer that LLMs have been explored as optimizers in prior work, and include these prior methods as both baselines (OPRO and Eureka) with additional discussion in Appendix B and C.4. We build on these works to introduce a principled approach to (1) **integrate domain knowledge** to (2) **overcome real-world distribution shifts** in personalized medicine, which have not been previously explored in the relevant literature to the best of our knowledge.
> 2. **Experiments in the Appendix**: Given the extended page limit for the camera-ready version, we have added additional discussion (in *red* text) to our Results section highlighting the additional experiments in the Appendix. We hope this will better point readership to our additional results that the Reviewer (and our team too!) found interesting.

---

> > ### Author Response · Authors · 2025-11-26
> >
> > Hi, hope you're doing well! Thank you again for your thoughtful review and support of our work. If you have any additional questions or comments about our revised manuscript, please feel free to let us know and we would be happy to discuss!

---

### Author Response · Authors · 2025-11-30
**Summary of Manuscript Updates**

Dear AC and readers,

We thank the Reviewers, AC, and the rest of the ICLR organizing team for their thoughtful feedback and consideration of our work. In light of the recent updates to this year's peer review process, we wanted to provide a summary of the discussion period and the improvements made to our manuscript.

### Reviewer Score Updates

To the best of our knowledge, our post-discussion average score immediately prior to the reversion of reviews was **6.5** with a confidence of **3**. The initial and updated reviewer recommendations were as follows:

| Reviewer | Initial Score/Confidence $\rightarrow$ Updated Score/Confidence |
| ---: | :--- |
| 6DUi | 8/3 (Unchanged) |
| iXHF | 4/3 (Unchanged) |
| cRy4 | 6/2 $\rightarrow$ 8/3 (**Score Increase**, see [reviewer comment](https://openreview.net/forum?id=w025bYRVkO&noteId=40XHdeFVlC)) |
| qPrg | 6/3 (Unchanged) |

We are grateful that the reviewers generally found our method (LEON) **performant** (Reviewers iXHF, cRy4) and work **theoretically grounded** (Reviewers 6DUi, qPrg), **easy to follow** (Reviewers 6DUi, iXHF, cRy4), and **well-motivated** for precision medicine applications (Reviewers 6DUi, cRy4).

### Summary of Manuscript Revisions

1. We included additional discussion on interesting experiments in the main paper (Reviewer 6DUi).
2. We updated our source code in our Supplementary Material to include a **code tutorial** (`notebooks/tutorial.ipynb`) featuring a new **`sklearn`-style API** for LEON (Reviewer qPrg).
3. We discussed how to extend LEON to **multi-objective optimization** in Appendix D.12 (Reviewer iXHF).
4. We included **new fairness subgroup analyses** in Appendix D.10 to demonstrate that the performance of LEON is not biased against specific patient races or genders on our tasks (Reviewer iXHF).
5. We included **additional ablation experiments** (Reviewers iXHF, cRy4, qPrg) outlined below:

| Ablation Experimental Question | Summary of Results |
| :--- | :--- |
| [iXHF, qPrg] Can we further boost the performance of LEON by using better design clustering algorithms? | **Yes**, we explored options like using medical embedding models and using a fixed number of clusters $N$. LEON can be used with many different clustering methods (Appendix E.11-E.13). |
| [iXHF] Can we improve the performance of baseline LLM optimizers by using domain knowledge? | Sometimes. However, **our method LEON still outperforms LLM baselines** (OPRO, Eureka, and LLAMBO) that have access to external knowledge (Appendix D.11). |
| [cRy4] Is iterative optimization against the surrogate model even necessary? | **Yes**, our iterative optimization in LEON consistently outperforms simple LLM prompting (Appendix E.14). |

Other questions - such as how the (1) quality of knowledge, (2) hyperparameters, and (3) severity distribution shift impact LEON performance - were answered in our original manuscript submission; see our previous responses to Reviewers below for detailed discussion.

Importantly, **there were no changes to the main findings and conclusions reported in our manuscript as a result of any additional experiments**.

Thank you again for your time and consideration of our work! We would be happy to answer any follow-up questions prior to the end of the discussion period.

---

### Meta-Review · Area_Chair_jK3L · 2026-01-06

**Summary:**

The reviewers identified several critical areas of concern：
​
- Heuristic Components and Stability: Reviewers questioned the use of heuristic clustering (k-means) and the choice of l_x clipping for Lipschitz constraints, citing potential instability or loss of model capacity
- Evaluation Bias: Reviewer was concerned that using learned oracle functions (surrogates) rather than real clinical endpoints could bake evaluation bias into the results.
- Fairness and Safety: Reviewer noted a lack of subgroup analysis (e.g., race and gender) and questioned the safeguards against unsafe treatment suggestions.
- Baseline Parity and Knowledge Necessity: Reviewers questioned whether the performance gain was simply due to the LLM's access to external knowledge rather than the LEON optimization framework itself.
- Practicality and Cost: Concerns were raised regarding the computational and monetary costs of an iterative agent-based pipeline, as well as the reproducibility of such a complex system.

**Reviewer Concerns:**

Concerns Addressed:
- Baseline Parity: The authors added new experiments giving baseline methods (LLAMBO, OPRO, Eureka) access to the same medical knowledge. The results showed that LEON still outperformed these knowledge-augmented baselines, proving the framework’s optimization logic is a primary driver of success.
- Fairness Analysis: The authors included a new subgroup analysis in Appendix D.10, showing no statistically significant performance difference across different patient races or genders on the tested tasks.
- Reproducibility: A minimal working example and an sklearn-style API were added to the supplementary material to lower the barrier for implementation.
- Clustering Sensitivity: The authors clarified the use of the "Elbow method" for cluster counts and demonstrated that LEON's performance is robust across various clustering counts and embedding models.
- Necessity of Iteration: An ablation study confirmed that LEON's iterative process significantly outperforms simple zero-iteration LLM prompting.

Outstanding Concerns:
- Learned Oracles: The reliance on learned oracles remains a limitation. While the authors correctly argue that administering arbitrary treatments to real patients for testing is impossible, the external validity of these simulated outcomes is inherently limited.
- Clinical Safety/Human-in-the-Loop: The authors explicitly stated that rigorous clinician-in-the-loop studies and hard safety constraints are outside the scope of this methodological contribution.
- Lipschitz Enforcement: The authors defended l_x  clipping as standard and efficient, leaving more sophisticated methods like spectral normalization for future work

**Reviewer Scores:**

- Reviewer cRy4 (Initial 6): Change to 8. This reviewer explicitly increased their score after the rebuttal, stating that the cost analysis and iterative ablations addressed their questions well.

- Reviewer 6DUi (Initial 8): No Change. Already a strong advocate
- Reviewer qPrg (Initial 6): Remain 6. Their requests for cost reporting, baseline parity, and a code tutorial were partially addressed. While the "learned oracle" concern persistst.
• Reviewer iXHF (Initial 4): upgrade to 5. The authors provided the requested subgroup analysis and trajectories. However, their fundamental skepticism regarding heuristic clipping and the conservative nature of the source-critic remains.

---

### Decision · Program_Chairs · 2026-01-26

Accept (Poster)